# ADAPTIVE GRADIENT CLIPPING FOR ROBUST FEDERATED LEARNING

**Youssef Allouah**[1][*]     **Rachid Guerraoui**[1]     **Nirupam Gupta**[2]

**Ahmed Jellouli**[1]     **Geovani Rizk**[1]     **John Stephan**[1][†]

[1]EPFL, Switzerland     [2]University of Copenhagen, Denmark
[†] Correspondence to: `john.stephan@epfl.ch`

## ABSTRACT

Robust federated learning aims to maintain reliable performance despite the presence of adversarial or misbehaving workers. While state-of-the-art (SOTA) robust distributed gradient descent (Robust-DGD) methods were proven theoretically optimal, their empirical success has often relied on pre-aggregation *gradient clipping*. However, existing *static* clipping strategies yield inconsistent results: enhancing robustness against some attacks while being ineffective or even detrimental against others. To address this limitation, we propose a principled *adaptive* clipping strategy, Adaptive Robust Clipping (ARC), which dynamically adjusts clipping thresholds based on the input gradients. We prove that ARC not only preserves the theoretical robustness guarantees of SOTA Robust-DGD methods but also provably improves asymptotic convergence when the model is well-initialized. Extensive experiments on benchmark image classification tasks confirm these theoretical insights, demonstrating that ARC significantly enhances robustness, particularly in highly heterogeneous and adversarial settings.

## 1 INTRODUCTION

Distributed machine learning, a.k.a. federated learning, has emerged as a dominant paradigm to cope with the increasing computational cost of learning tasks, mainly due to growing model sizes and datasets (Kairouz et al., 2021). *Worker* machines, holding each a fraction of the training dataset, collaborate over a network to learn an optimal common model over the collection of their datasets. Workers typically collaborate with the help of a central coordinator, that we call *server* (McMahan et al., 2017). Besides scalability, distributed learning is also helpful in preserving data ownership and sovereignty, since the workers do not have to share their local datasets during the learning.

Conventional distributed learning algorithms are known to be vulnerable to misbehaving workers that could behave unpredictably (Blanchard et al., 2017; Kairouz et al., 2021; Guerraoui et al., 2023). Misbehavior may result from software and hardware bugs, data poisoning, or malicious players controlling part of the network. In the parlance of distributed computing, misbehaving workers are referred to as *Byzantine* (Lamport et al., 1982). Due to the growing influence of distributed learning in critical public-domain applications such as healthcare (Nguyen et al., 2022) and finance (Long et al., 2020), the problem of robustness to misbehaving workers, a.k.a. robust distributed learning, has received significant attention (Yin et al., 2018; Farhadkhani et al., 2022; Karimireddy et al., 2022; Gorbunov et al., 2023; Allouah et al., 2023a; Farhadkhani et al., 2023; El-Mhamdi et al., 2023).

Robust distributed learning algorithms primarily rely on robust aggregation, such as coordinate-wise trimmed mean (CWTM) (Yin et al., 2018), geometric median (GM) (Chen et al., 2017) and Multi-Krum (MK) (Blanchard et al., 2017). Specifically, in robust distributed gradient descent (*Robust-DGD*), the server aggregates the workers' local gradients using a robust aggregation method, instead of simply averaging them. This protects the learning from erroneous gradients sent by misbehaving workers. Recent work has made significant improvements over these aggregation techniques by incorporating a pre-aggregation step such as bucketing (Karimireddy et al., 2022; Gorbunov et al., 2023) and nearest-neighbor mixing (NNM) (Allouah et al., 2023a), to tackle gradient

---

[*]Authors are listed in alphabetical order.

dissimilarity resulting from data heterogeneity. The learning guarantee of the resulting Robust-DGD has been proven optimal (Allouah et al., 2023b), i.e., it cannot be improved without additional assumptions under the standard heterogeneity model of $(G, B)$-gradient dissimilarity (Karimireddy et al., 2020).

Despite its theoretical tightness, the empirical success of Robust-DGD has unknowingly relied on pre-aggregation *gradient clipping* (Mhamdi et al., 2021; Farhadkhani et al., 2022; Allouah et al., 2023a). Specifically, clipping the gradients of the workers prior to aggregation has been observed to sometimes enhance the algorithm's empirical performance in the presence of adversarial workers (as evidenced in Figure 7a). Yet, this improvement lacks a concrete explanation, raising the question of whether the observed benefits of clipping are merely anecdotal. This leads to the natural inquiry: *Why, and when, does pre-aggregation clipping improve robustness?*

In particular, using a constant clipping threshold, referred to as static clipping, has exhibited mixed results. Figure 7 shows that while static clipping effectively mitigates sign-flipping (SF) attacks, it completely fails under label-flipping (LF). This indicates an inherent fragility of static clipping. Indeed, we prove in our work that static clipping breaks the standard $(f, \kappa)$-robustness property of an aggregation method (Allouah et al., 2023a). This highlights a key shortcoming of existing empirical results that rely on static clipping (Mhamdi et al., 2021; Farhadkhani et al., 2022; Allouah et al., 2023a). To overcome the limitations of static clipping but preserve its empirical benefits at the same time, we introduce a novel adaptive clipping scheme, termed Adaptive Robust Clipping (ARC).

ARC dynamically adjusts the clipping threshold as per the gradients sent by the workers and the fraction of adversarial workers to be tolerated. We demonstrate that integrating ARC into Robust-DGD consistently improves its empirical performance (see Figures 7 and 1a), while also preserving the convergence guarantee of the original Robust-DGD algorithm. Moreover, we show that when the model initialization is good, ARC provably improves the robustness of Robust-DGD. The benefits of ARC are more pronounced as the fraction of misbehaving workers approaches the system's breakdown point[1] and when the data across the workers is highly heterogeneous. Our key results are summarized below. Critical comparisons to prior work are deferred to Section 6.

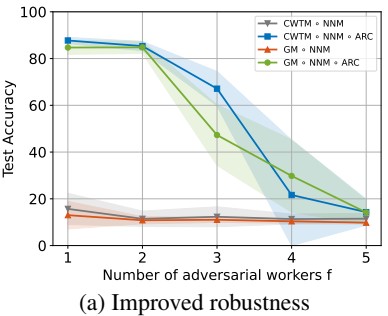
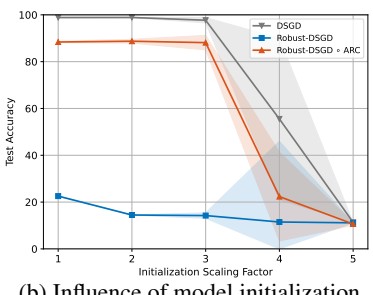

(a) Improved robustness         (b) Influence of model initialization

Figure 1: *Worst-case maximal accuracies* of Robust-DSGD, with and without ARC, across several types of misbehavior for distributed MNIST with 10 honest workers and under *extreme* heterogeneity. On the left, we vary the number of adversarial workers. On the right, we vary the initialization conditions by scaling a well-chosen set of initial parameters (CWTM ∘ NNM is used, and $f = 1$). More details on the experimental setup can be found in Sections 4 and 5.2, and Appendix D.

**Main results & contributions.** We consider a system comprising $n$ workers and a server. The goal is to tolerate up to $f$ adversarial workers.

*(1) Adaptive robust clipping (ARC).* We propose ARC, wherein prior to aggregating the gradients, the server clips the largest $k := \lfloor 2(f/n)(n - f) \rfloor$ gradients using a clipping parameter given by the (Euclidean) norm of the $(k + 1)$-th largest gradient. It is important to note that in contrast to existing adaptive clipping schemes (Diakonikolas et al., 2020; Abdalla & Zhivotovskiy, 2024), ARC does not require additional a priori information on honest workers' gradients. We prove that ARC preserves the robustness guarantee of the original robust aggregation method.

*(2) Improved empirical robustness.* We conduct experiments on MNIST (Deng, 2012), Fashion-MNIST (Xiao et al., 2017), and CIFAR-10 (Krizhevsky et al., 2014), across various data heterogeneity

---

[1]Breakdown point refers to the minimum fraction of adversarial workers that can break the system, making it impossible to guarantee a bound on the learning error (Allouah et al., 2023b).

settings and adversarial regimes. Our results demonstrate that ARC significantly enhances the performance of state-of-the-art Robust-DGD methods, particularly in scenarios with high data heterogeneity (Figure 1a) and a large number of adversarial workers (Figure 4b).

*(3) Improved learning guarantee.* We demonstrate that ARC possesses an additional property that is not satisfied by classical robust aggregation methods. Specifically, ARC constrains the norm of an adversarial gradient by that of an honest (non-adversarial) gradient. Leveraging this property, we show that ARC circumvents the lower bound established under data heterogeneity in Allouah et al. (2023b), provided the honest gradients are bounded at model initialization. An empirical validation of this insight is shown in Figure 1b. Such model initialization is often satisfiable in practice (Glorot & Bengio, 2010), highlighting the practical relevance of ARC. When the model is arbitrarily initialized, ARC recovers the original convergence guarantee of Robust-DGD in the worst case.

## 2 PROBLEM STATEMENT AND RELEVANT BACKGROUND

We consider the problem of distributed learning in the server-based architecture. The system comprises $n$ workers represented by $w_1, \ldots, w_n$, that collaborate with the help of a trusted server. The workers hold local datasets $\mathcal{D}_1, \ldots, \mathcal{D}_n$ respectively, each composed of $m$ data points from an input space $\mathcal{Z}$. Specifically, for any $i \in [n]$, $\mathcal{D}_i := \{z_1^{(i)}, \ldots, z_m^{(i)}\} \subset \mathcal{Z}^m$. For a given model parameterized by vector $\theta \in \mathbb{R}^d$, $d$ being the number of trainable parameters, each worker $w_i$ incurs a loss given by $\mathcal{L}_i(\theta) := \frac{1}{m} \sum_{k=1}^m \ell(\theta, z_k^{(i)})$, where $\ell : \mathbb{R}^d \times \mathcal{Z} \to \mathbb{R}$ is the point-wise loss. We make the following standard assumptions (Bottou et al., 2018): (i) the point-wise loss function $\ell$ is differentiable with respect to $\theta$. (ii) For all $i \in [n]$, $\mathcal{L}_i$ is $L$-Lipschitz smooth, i.e., there exists $L \in \mathbb{R}^+$ such that for all $\theta, \theta' \in \mathbb{R}^d$, $\|\nabla \mathcal{L}_i(\theta) - \nabla \mathcal{L}_i(\theta')\| \leq L \|\theta - \theta'\|$, where $\|\cdot\|$ refers to the Euclidean norm. In the ideal setting where all workers are *honest*, i.e., follow the prescribed algorithm correctly, the server aims to minimize the global average loss function given by $\mathcal{L}(\theta) := \frac{1}{n} \sum_{i=1}^n \mathcal{L}_i(\theta)$. However, this objective is rendered vacuous when some workers could be adversarial, described in the following.

A robust distributed learning algorithm aims to output a good model despite the presence of adversarial (a.k.a., *Byzantine*) workers in the system (Su & Vaidya, 2016; Yin et al., 2018; Allouah et al., 2023a). Specifically, the goal is to design a distributed learning algorithm that can tolerate up to $f$ adversarial workers, of a priori unknown identity, out of $n$ workers. Adversarial workers need not follow a prescribed algorithm, and can send arbitrary information to the server. In the context of distributed gradient descent, adversarial workers can send incorrect gradients to the server (Baruch et al., 2019; Xie et al., 2020). We let $\mathcal{H} \subseteq [n]$, with $|\mathcal{H}| = n - f$, denote the set of honest workers that do not deviate from the algorithm. The objective of the server is to minimize the average loss function of honest workers given by $\mathcal{L}_{\mathcal{H}}(\theta) := \frac{1}{|\mathcal{H}|} \sum_{i \in \mathcal{H}} \mathcal{L}_i(\theta)$, $\forall \theta \in \mathbb{R}^d$. While we can find a minimum of $\mathcal{L}_{\mathcal{H}}(\theta)$ when $\mathcal{L}_{\mathcal{H}}$ is convex, in general however the loss function is non-convex, and the optimization problem is NP-hard (Boyd & Vandenberghe, 2004). Therefore, we aim to find a *stationary point* of $\mathcal{L}_{\mathcal{H}}$ instead, i.e., $\theta^*$ such that $\|\nabla \mathcal{L}_{\mathcal{H}}(\theta^*)\| = 0$. Formally, we define robustness to adversarial workers by $(f, \epsilon)$-*resilience* (Allouah et al., 2023a).

**Definition 2.1.** A distributed learning algorithm $\mathcal{A}$ is said to be $(f, \varepsilon)$-*resilient* if, despite the presence of $f$ adversarial workers, $\mathcal{A}$ enables the server to output a model $\hat{\theta}$ such that $\mathbb{E}\left[\left\|\nabla \mathcal{L}_{\mathcal{H}}\left(\hat{\theta}\right)\right\|^2\right] \leq \varepsilon$, where the expectation $\mathbb{E}\left[\cdot\right]$ is over the randomness of the algorithm.

If a distributed learning algorithm $\mathcal{A}$ is $(f, \varepsilon)$-resilient, then it can tolerate up to $f$ adversarial workers. It is standard in robust distributed learning to design robust algorithms using $f$ as a parameter (Blanchard et al., 2017; Yin et al., 2018; Gorbunov et al., 2023; Allouah et al., 2023a).

In the context of distributed learning, data heterogeneity is characterized by the following standard notion of $(G, B)$-gradient dissimilarity (Vaswani et al., 2019; Karimireddy et al., 2020; 2022; Gorbunov et al., 2023; Allouah et al., 2023b).

**Definition 2.2.** Loss functions $\mathcal{L}_i$, $i \in \mathcal{H}$, are said to satisfy $(G, B)$-*gradient dissimilarity* if,

$$\frac{1}{|\mathcal{H}|} \sum_{i \in \mathcal{H}} \|\nabla \mathcal{L}_i(\theta) - \nabla \mathcal{L}_{\mathcal{H}}(\theta)\|^2 \leq G^2 + B^2 \|\nabla \mathcal{L}_{\mathcal{H}}(\theta)\|^2, \quad \forall \theta \in \mathbb{R}^d.$$

**General limitations on robustness.** Note that $(f, \varepsilon)$-resilience is impossible (for any $\varepsilon$) when $f/n \geq 1/2$ (Gupta & Vaidya, 2020). Moreover, even for $f/n < 1/2$, it is generally impossible

to achieve $(f, \varepsilon)$-resilience for arbitrarily small $\varepsilon$ due to the disparity amongst the local datasets $\mathcal{D}_i$, $i \in \mathcal{H}$ (a.k.a., data heterogeneity) (Liu et al., 2021). Henceforth, we assume that $n > 2f$. Under $(G, B)$-gradient dissimilarity, we have the following lower bound on the achievable resilience.

**Lemma 2.3** (Non-convex extension of Theorem 1 Allouah et al.). *Under $(G, B)$-gradient dissimilarity, a distributed learning algorithm is $(f, \varepsilon)$-resilient only if $\frac{f}{n} < \frac{1}{2+B^2}$ and $\varepsilon \geq \frac{1}{4} \cdot \frac{f}{n-(2+B^2)f} G^2$.*

For a given distributed learning problem, the minimum fraction of adversarial workers that renders any distributed learning algorithm ineffective is called the *breakdown point* (Guerraoui et al., 2023; Allouah et al., 2023b). Thus, according to Lemma 2.3, there exists a distributed learning problem satisfying $(G, B)$-gradient dissimilarity whose breakdown point is given by $1/(2 + B^2)$.

**Robust distributed gradient descent.** To tolerate adversarial workers, we replace the averaging operation in the classic distributed gradient descent (DGD) method with a robust aggregation, e.g., coordinate-wise trimmed mean (CWTM) and median (CWMed) (Yin et al., 2018), geometric median (GM) (Chen et al., 2017), and Multi-Krum (MK) (Blanchard et al., 2017). This yields Robust-DGD, presented in Algorithm 1, where robust aggregation protects the learning from incorrect gradients sent by the adversarial workers. The robustness of an aggregation method can be quantified by the following property of $(f, \kappa)$-robustness (Allouah et al., 2023a).

---

**Algorithm 1** Robust Distributed Gradient Descent (Robust-DGD)

---

**Initialization: Server** chooses a model $\theta_1 \in \mathbb{R}^d$, a learning rate $\gamma \in \mathbb{R}^+$ and a robust aggregation method $\mathbf{F} : \mathbb{R}^{n \times d} \to \mathbb{R}^d$.

**for** $t = 1$ to $T$ **do**

    **Server** broadcasts $\theta_t$ to all workers.

    **for each honest worker** $w_i$ **in parallel do**

        Compute local gradient $g_t^{(i)} \coloneqq \nabla \mathcal{L}_i(\theta_t)$, and send $g_t^{(i)}$ to **Server**.

        *// An adversarial worker $w_j$ can send an arbitrary vector in $\mathbb{R}^d$ for $g_t^{(j)}$*

    **end for**

    **Server** computes $R_t \coloneqq \mathbf{F}\left(g_t^{(1)}, \ldots, g_t^{(n)}\right)$.

    **Server** computes the updated model $\theta_{t+1} \coloneqq \theta_t - \gamma R_t$ .

**end for**

**Output: Server** outputs $\hat{\theta}$ chosen uniformly at random from $\{\theta_1, \ldots, \theta_T\}$.

---

**Definition 2.4.** $\mathbf{F} : \mathbb{R}^{n \times d} \to \mathbb{R}^d$ is said to be $(f, \kappa)$-*robust* if there exists a robustness coefficient $\kappa \in \mathbb{R}$ such that for all $x_1, \ldots, x_n \in \mathbb{R}^d$ and any set $S \subseteq [n]$, $|S| = n - f$, the following holds:

$$\|\mathbf{F}(x_1, \ldots, x_n) - \overline{x}_S\|^2 \leq \frac{\kappa}{|S|} \sum_{i \in S} \|x_i - \overline{x}_S\|^2 \quad , \quad \text{where } \overline{x}_S = \frac{1}{|S|} \sum_{i \in S} x_i.$$

$(f, \kappa)$-robustness encompasses several robust aggregations, including the ones mentioned above and more (e.g., see Allouah et al. (2023a)). It has been shown that an aggregation method is $(f, \kappa)$-robust only if $\kappa \geq \frac{f}{n-2f}$ (Allouah et al., 2023a). Importantly, the asymptotic error of Robust-DGD with an $(f, \kappa)$-robust aggregation, where $\kappa \in \mathcal{O}(f/(n-2f))$, matches the lower bound (recalled in Lemma 2.3) (Allouah et al., 2023b). This testifies to the tightness of the $(f, \kappa)$-robustness property. Note that the aforementioned aggregation methods (CWTM, CWMed, GM and MK) attain optimal robustness when composed with the pre-aggregation scheme: nearest neighbor mixing (NNM) (Allouah et al., 2023a). Specifically, we recall the following result.

**Lemma 2.5** (Lemma 1 in Allouah et al. (2023a)). *For $\mathbf{F} \in \{\mathbf{CWTM}, \mathbf{CWMed}, \mathbf{GM}, \mathbf{MK}\}$, the composition $\mathbf{F} \circ \mathbf{NNM}$ is $(f, \kappa)$-robust with $\kappa \leq \frac{8f}{n-f}\left(1 + \left(1 + \frac{f}{n-2f}\right)^2\right)$.*

Thus, if $n \geq (2+\delta)f$ for $\delta > 0$, then $\mathbf{F} \circ \mathbf{NNM}$ is $(f, \kappa)$-robust with $\kappa \leq \frac{16f}{n-f}\left(\frac{\delta+1}{\delta}\right)^2 \in \mathcal{O}\left(\frac{f}{n-2f}\right)$.

**Convergence of Robust-DGD.** Lastly, we recall the convergence result for Robust-DGD with an $(f, \kappa)$-robust aggregation $\mathbf{F}$. We let $\mathcal{L}_{\mathcal{H}}^*$ denote the minimum value of $\mathcal{L}_{\mathcal{H}}(\theta)$.

**Lemma 2.6** (Theorem 2 in Allouah et al. (2023b)). *Consider Algorithm 1 with $\gamma \leq 1/L$. Let $\Delta_o \in \mathbb{R}^+$ such that $\mathcal{L}_{\mathcal{H}}(\theta_1) - \mathcal{L}_{\mathcal{H}}^* \leq \Delta_o$. If $\mathbf{F}$ is $(f, \kappa)$-robust with $\kappa B^2 < 1$, then*

$$\frac{1}{T}\sum_{t=1}^{T}\|\nabla\mathcal{L}_{\mathcal{H}}(\theta_t)\|^2 \leq \frac{2\Delta_o}{(1-\kappa B^2)\gamma T} + \frac{\kappa G^2}{1-\kappa B^2}.$$

Thus, when $\kappa \in \mathcal{O}\left(f/(n-2f)\right)$ (and smaller than $1/B^2$), Robust-DGD is optimal, i.e., its error matches the lower bound recalled in Lemma 2.3, as soon as $T \geq \Delta_o/\gamma\kappa G^2$. For $f = 0$, Lemma 2.6 recovers the convergence of DGD, provided that $\kappa$ is tight, i.e., $\kappa \in \mathcal{O}(\frac{f}{n-2f})$.

## 3 ADAPTIVE ROBUST CLIPPING (ARC) AND ITS PROPERTIES

In this section, we first present a preliminary observation on the fragility of *static* clipping, and then introduce *adaptive* robust clipping (i.e., ARC) along with its robustness guarantees.

For a clipping parameter $C \in \mathbb{R}^+$ and a vector $x \in \mathbb{R}^d$, we denote $\text{clip}_C(x) := \min\left(1, \frac{C}{\|x\|}\right)x$. For a set of $n$ vectors $x_1, \ldots, x_n \in \mathbb{R}^d$, we denote $\mathbf{Clip}_C(x_1, \ldots, x_n) := (\text{clip}_C(x_1), \ldots, \text{clip}_C(x_n))$ .

Let $\mathbf{F}$ be an $(f, \kappa)$-robust aggregation. Given a set of $n$ vectors $x_1, \ldots, x_n$, let $\mathbf{F} \circ \mathbf{Clip}_C(x_1, \ldots, x_n) := \mathbf{F}(\mathbf{Clip}_C(x_1, \ldots, x_n))$. We make the following observation.

**Lemma 3.1.** *For any fixed $C \in \mathbb{R}^+$ and $\kappa' \geq 0$, $\mathbf{F} \circ \mathbf{Clip}_C$ is not $(f, \kappa')$-robust.*

Thus, if the clipping threshold is fixed, i.e., independent from the input vectors, pre-aggregation clipping does not preserve the robustness of the original aggregation. This fragility of such *static* clipping is also apparent in practice, as shown by our experimental study in Section 4 and Appendix E.

**Description of ARC.** ARC is an adaptive clipping scheme that only makes use of the standard robustness parameter $f$, i.e., the tolerable number of adversarial workers. ARC is *adaptive* in the sense that the clipping threshold $C$ is not fixed but depends on the input vectors. Specifically, ARC clips the largest $k = \lfloor 2(f/n)(n-f) \rfloor$ vectors using a clipping parameter given by the norm of the $(k+1)$-th largest input vector. The overall scheme is formally presented in Algorithm 2, and its computational complexity is $\mathcal{O}(nd + n\log(n))$ (see Appendix A.2). Thus, pre-composing more computationally expensive schemes like NNM and Multi-Krum, which have a complexity of $\mathcal{O}(dn^2)$, with ARC does not introduce a significant overhead (see Appendix F.4). For a detailed explanation of the underlying intuition behind our adaptive clipping strategy, ARC, we refer the reader to Appendix A.1.

---

**Algorithm 2** Adaptive Robust Clipping (ARC)

---

**Input:** $f$ and $x_1, \ldots, x_n \in \mathbb{R}^d$.
Find a permutation $\pi : [n] \to [n]$ such that $\|x_{\pi(1)}\| \geq \|x_{\pi(2)}\| \ldots \geq \|x_{\pi(n)}\|$.
Set $k = \left\lfloor 2\frac{f}{n}(n-f) \right\rfloor$ and $C = \|x_{\pi(k+1)}\|$.
**Output:** $\mathbf{Clip}_C(x_1, \ldots, x_n)$ .

---

**Robustness Guarantee.** We present below the preservation of robustness guaranteed by ARC. Let $\mathbf{F}$ be an $(f, \kappa)$-robust aggregation method and $\mathbf{F} \circ \mathbf{ARC}(x_1, \ldots, x_n) := \mathbf{F}(\mathbf{ARC}(x_1, \ldots, x_n))$ .

**Theorem 3.2.** *If $\mathbf{F}$ is $(f, \kappa)$-robust, then $\mathbf{F} \circ \mathbf{ARC}$ is $\left(f, \kappa + \frac{2f}{n-2f}\right)$-robust.*

Proofs for the results presented in this section are deferred to Appendix B. Since $\kappa \geq \frac{f}{n-2f}$ (recalled in Section 2), Theorem 3.2 implies that $\mathbf{F} \circ \mathbf{ARC}$ is $(f, 3\kappa)$-robust, i.e., ARC preserves the robustness of the original aggregation scheme. Therefore, a convergence result for Robust-DGD with ARC follows verbatim from Lemma 2.6, replacing $\kappa$ with $3\kappa$. Despite this multiplicative factor of 3, we observe in Section 4 that incorporating ARC consistently improves the empirical performance of classical aggregation methods. A more detailed theoretical explanation is provided later in Section 5.1.

## 4 EMPIRICAL EVALUATION

In this section, we delve into the practical performance of ARC when incorporated in Robust-D*S*GD (Algorithm 3 in Appendix D), an order-optimal *stochastic* variant of Robust-DGD (Allouah et al., 2023a). We conduct experiments on standard image classification tasks, covering different adversarial scenarios. We empirically test four aggregation methods when pre-composed with ARC. We contrast

these outcomes against the performance of these aggregations under static clipping and when no gradient clipping is used. We show in Table 1, and Figures 2 and 4, the metric of *worst-case maximal accuracy*. For each of the five Byzantine attacks executed (see Appendix D.3), we record the maximal accuracy achieved by Robust-DSGD during the learning procedure under that attack. The worst-case maximal accuracy is thus the smallest maximal accuracy reached across the five attacks. Furthermore, we use the Dirichlet (Hsu et al., 2019a) distribution of parameter $\alpha$ to simulate data heterogeneity. The comprehensive experimental setup can be found in Appendix D. In this section, we focus on presenting our results for the CWTM and GM aggregation methods applied to MNIST and CIFAR-10. As benchmark, we also execute the standard DSGD algorithm, in the absence of adversarial workers (i.e., without attack and $f = 0$). Results for Fashion-MNIST, as well as for CWMed and MK, are provided in Appendices E and F. All aggregation methods in our experiments are pre-composed with NNM (Allouah et al., 2023a), but we omit explicit mention of NNM in their names for simplicity.

## 4.1 BRITTLENESS OF STATIC CLIPPING AND SUPERIORITY OF ARC

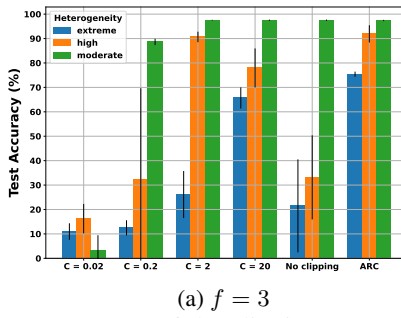
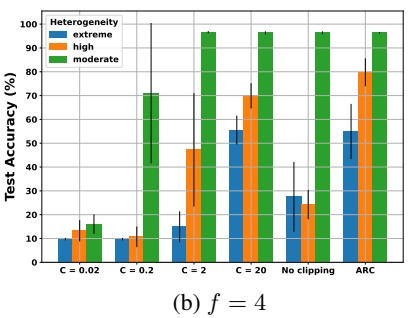

(a) $f = 3$                               (b) $f = 4$

Figure 2: Impact of the clipping strategy, under varying heterogeneity levels, on the worst-case maximal accuracy achieved by Robust-DSGD on MNIST with $n = 15$ workers. CWTM ∘ NNM is used as aggregation. DSGD reaches at least 98.5% in accuracy in all heterogeneity regimes.

We empirically demonstrate the brittleness of static clipping in robust distributed learning. While an exhaustive empirical study can be found in Appendix E, we present findings on MNIST using CWTM, considering three heterogeneity levels: *moderate* ($\alpha = 1$), *high* ($\alpha = 0.1$), and *extreme*. We execute Robust-DSGD with $n = 15$ workers, among which $f \in \{3, 4\}$ are adversarial, and test static clipping thresholds $C \in \{0.02, 0.2, 2, 20\}$. First, we observe that the optimal $C$ depends heavily on data heterogeneity. As seen in Figure 2, $C = 2$ is ideal in high heterogeneity, whereas $C = 20$ performs much better under extreme heterogeneity. This dependency requires fine-tuning, which is impractical in distributed settings. Moreover, static clipping's effectiveness varies significantly across Byzantine attacks. While $C = 2$ performs well under SF (Allen-Zhu et al., 2020), it fails under LF, causing training collapse (see Figure 7 in Appendix E). This unpredictability makes static clipping unreliable in practice. The number of adversarial workers further affects static clipping's performance. In Figure 2, $C = 2$ is optimal for $f = 3$ under high heterogeneity, yet its accuracy drops below 50% when $f = 4$, making it ineffective. Unlike static clipping, ARC consistently matches or outperforms the best static threshold across all configurations, as shown in Figure 2. These results underscore the superiority of ARC over static clipping, eliminating the need for manual tuning and ensuring robustness regardless of unpredictable parameters like data heterogeneity and Byzantine attacks.

## 4.2 PERFORMANCE GAINS OF ARC OVER ROBUST-DSGD

To quantify the improvement induced by ARC over Robust-DSGD without clipping, we evaluate a distributed system with $n - f = 10$ honest workers while varying the number of adversarial workers $f$. We compare the performance of Robust-DSGD with and without ARC on the MNIST dataset, highlighting the improvements achieved by our method. Additionally, we extend our analysis to larger distributed systems with 30 honest workers and $f \in \{3, 6, 9\}$, as detailed in Appendix F.1.2.

**ARC boosts robustness in high heterogeneity.** Figure 3 shows the performance of Robust-DSGD against the FOE attack when using ARC opposed to no clipping. In low heterogeneity (i.e., $\alpha = 0.5$), the performances of ARC and no clipping are comparable. When the heterogeneity increases ($\alpha = 0.1$), CWTM and GM significantly struggle to learn, while the same aggregations composed

with ARC almost match the final accuracy of DSGD. In extreme heterogeneity, the improvement induced by ARC is the most pronounced, enabling both aggregations to reach a final accuracy close to 90%. Contrastingly, the same aggregations without clipping stagnate at around 10% throughout the training. Interestingly, only $f = 1$ adversarial worker among $n = 11$ (i.e., less than 10%) suffices to completely deteriorate the learning when no clipping is applied, highlighting the necessity of ARC in extreme heterogeneity. These observations are also conveyed in Figure 4a which compares the worst-case maximal accuracies achieved by Robust-DSGD when $f = 1$, with and without ARC, for varying levels of heterogeneity. While CWTM and GM yield comparable accuracies with ARC in low heterogeneity ($\alpha \geq 0.5$), their performance significantly degrades when $\alpha$ drops below that threshold. Indeed, when $\alpha = 0.1$, their accuracies drop below 65%, while ARC enables the same aggregations to maintain their accuracy at just below 98%. In extreme heterogeneity, the performance of the aggregations without clipping completely deteriorates with accuracies close to 15%. Contrastingly, ARC efficiently mitigates the Byzantine attacks, resulting in accuracies above 85% in the worst case for both aggregations. Similar plots for $f \in \{3, 5, 7\}$ convey similar observations (see Appendix F.1).

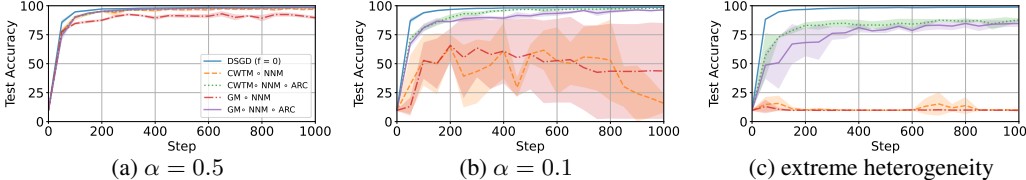

(a) $\alpha = 0.5$       (b) $\alpha = 0.1$       (c) extreme heterogeneity

Figure 3: Performance of Robust-DSGD when using ARC and without clipping on MNIST. There are 10 honest workers and $f = 1$ adversarial worker executing FOE (Xie et al., 2020).

**ARC increases the breakdown point in adverse settings.** Figure 1a of Section 1 shows that for $f \in \{1, ..., 5\}$, Robust-DSGD with CWTM and GM completely fails to learn, consistently yielding worst-case maximal accuracies close to 15%. This suggests that $f = 1$ constitutes the breakdown point for these aggregations in extreme heterogeneity. However, composing them with ARC increases the breakdown point of these aggregations to $f = 3$. Indeed, for $f = 1$ and 2, ARC enables Robust-DSGD to achieve accuracies greater than 85% in the worst case. However, when $f \geq 3$, the performance degrades, although CWTM with ARC is still able to reach a satisfactory accuracy close to 70%. Moreover, even when the heterogeneity is not large ($\alpha = 0.5$), ARC still produces a significant improvement when the fraction of adversarial workers increases in the system. Indeed, in Figure 4b, the performances of ARC and no clipping are comparable for $f \leq 3$. However, the improvement induced by ARC is much more visible when $f \geq 5$. Particularly when $f = 7$, ARC enables CWTM and GM to reach accuracies greater than 80% in the worst case, whereas the same aggregations yield accuracies below 40% without clipping, indicating the raise of the breakdown point due to ARC. Plots for $\alpha = 0.1$ and 1 convey the same observations in Appendix F.1.

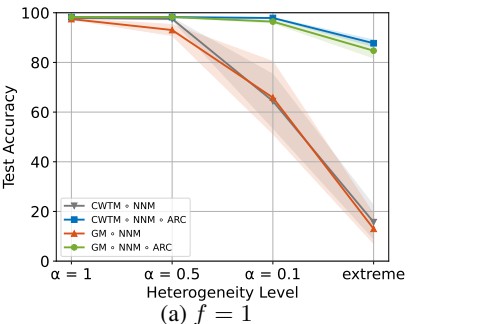 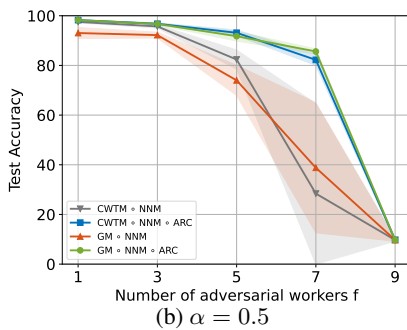

(a) $f = 1$       (b) $\alpha = 0.5$

Figure 4: *Worst-case maximal accuracies* achieved by Robust-DSGD, with and without ARC, on heterogeneously-distributed MNIST with 10 honest workers. *Left:* $f = 1$ adversarial worker among $n = 11$ for varying levels of heterogeneity. *Right:* $\alpha = 0.5$ for varying $f$.

**Improved robustness on CIFAR-10.** We also conduct experiments on CIFAR-10 with $n = 17$ and $f = 1$. Table 1 shows the worst-case maximal accuracies achieved by ARC and no clipping for CWTM and GM. For $\alpha = 0.2$, ARC consistently outputs accuracies greater than 67% for both aggregations, while no clipping yields lower accuracies (with a larger variance). For instance, GM achieves 41.2% on average, i.e., 26% less than its counterpart with ARC. In the more heterogeneous setting $\alpha = 0.075$, ARC enables all aggregations to reach worst-case maximal accuracies close to

60% (with a small variance). However, executing the same methods without clipping significantly deteriorates the performance of Robust-DSGD, with GM achieving 16% in worst-case accuracy. This can also be seen in Figure 23a, where FOE completely degrades the learning when ARC is not used. We further extend this experiment to a larger distributed system with 33 honest workers, as detailed in Appendix F.3, where we observe the same trends as reported here.

| Aggregation | $\alpha = 0.2$ | | $\alpha = 0.075$ | |
| | No Clipping | ARC | No Clipping | ARC |
| --- | --- | --- | --- | --- |
| CWTM | $51.6 \pm 5.1$ | $\mathbf{67.8 \pm 0.9}$ | $40.7 \pm 0.5$ | $\mathbf{60.5 \pm 1.2}$ |
| GM | $41.2 \pm 3.5$ | $\mathbf{67.0 \pm 1.0}$ | $16.0 \pm 2.3$ | $\mathbf{60.0 \pm 2.0}$ |

Table 1: *Worst-case maximal accuracies* (%) achieved by Robust-DSGD on heterogeneously-distributed CIFAR-10 with ARC and without. There is $f = 1$ adversarial worker among $n = 17$. As a baseline, DSGD ($f = 0$) reaches 76.5% and 70% when $\alpha = 0.2$ and 0.075, respectively.

## 5 IMPROVED GUARANTEE OF ROBUST-DGD WITH ARC

In Section 4, we empirically demonstrated that ARC outperforms SOTA aggregation methods without clipping, despite these methods being theoretically proven to be optimal. This naturally raises the question: why does ARC, which shares similar convergence guarantees, significantly outperform plain Robust-DSGD? To address this, we now focus on the influence of model initialization on the robustness of aggregation methods, considering both theoretical results and empirical insights.

### 5.1 IMPROVEMENT OF CONVERGENCE GUARANTEES

In this section, we characterize the improvement induced by ARC over the lower bound recalled in Lemma 2.3. Specifically, we consider Algorithm 1 with aggregation $\mathbf{F} \circ \mathbf{ARC}$, i.e., in each learning step $t$, $R_t := \mathbf{F} \circ \mathbf{ARC}\left(g_t^{(1)}, \ldots, g_t^{(n)}\right)$. We establish in Lemma 5.1 a key property of ARC, crucial to derive the improved stationarity error of Robust-DGD with ARC in Theorem 5.2.

**Lemma 5.1** (**Bounded Output**). *Let* $\mathbf{F}$ *be an* $(f, \kappa)$-*robust aggregation method. For any vectors* $x_1, \ldots, x_n \in \mathbb{R}^d$ *and set* $S \subset [n]$ *such that* $|S| = n - f$, $\|\mathbf{F} \circ \mathbf{ARC}(x_1, \ldots, x_n)\| \leq \max_{i \in S} \|x_i\|$.

This result indicates that incorporating ARC bounds the norm of the aggregated output by the largest norm of the honest gradients, i.e., $\max_{i \in \mathcal{H}} \|x_i\|$. In Appendix C, we provide the proof of Lemma 5.1 and further demonstrate in Lemma C.1 that this property is specific to ARC, i.e., classic $(f, \kappa)$-robust aggregation methods do not inherently exhibit this behavior. This property enables us to obtain the following improvement on the asymptotic convergence guarantee of Robust-DGD with ARC.

In the following, we show that when the local gradients of the honest workers at the initial model $\theta_1$ are sufficiently small, then Robust-DGD with ARC overcomes the lower bound recalled in Lemma 2.3. We denote $\mathsf{BP} := \frac{1}{2+B^2}$, $\varepsilon_o := \frac{1}{4} \cdot \frac{G^2(f/n)}{1-(2+B^2)(f/n)}$, and $\Psi(G, B, \rho) := 640 \left(1 + \frac{1}{B^2}\right)^2 \left(1 + \frac{B^2\rho^2}{G^2}\right)$,

where $\rho$ denotes a real value. Recall from Lemma 2.3 that $\mathsf{BP}$ and $\varepsilon_o$ are the breakdown point and the lower bound on the stationarity error when $f/n < \mathsf{BP}$, respectively, under $(G, B)$-gradient dissimilarity. We obtain the following theorem for Robust-DGD with ARC. Let $\Delta_o$ be a real value such that $\mathcal{L}_\mathcal{H}(\theta_1) - \mathcal{L}_\mathcal{H}^* \leq \Delta_o$. The proof of Theorem 5.2 is deferred to Appendix C.

**Theorem 5.2.** *Suppose* $B > 0$ *and there exists* $\zeta \in \mathbb{R}^+$ *such that* $\max_{i \in \mathcal{H}} \|\nabla \mathcal{L}_i(\theta_1)\| \leq \zeta$. *Let* $\mathbf{F} \in \{\mathbf{CWTM}, \mathbf{CWMed}, \mathbf{GM}, \mathbf{MK}\} \circ \mathbf{NNM}$, $\gamma = \min\left\{\left(\frac{\Delta_o}{\kappa G^2}\right)\frac{1}{T}, \frac{1}{L}\right\}$ *and* $T \geq \frac{\Delta_o L}{\kappa G^2}$. *Consider an arbitrary real value* $\xi_o \in (0, 1)$. *Let* $\rho := \exp\left(\frac{(2+B^2)\Delta_o}{(1-\xi_o)G^2}L\right)\zeta$. *For any* $\upsilon \in (0, 1)$, *if* $\frac{f}{n} := (1-\xi)\mathsf{BP}$, *where* $0 < \xi \leq \min\left\{\frac{\upsilon}{\Psi(G,B,\rho)}, \xi_o\right\}$, *then* $\mathbb{E}\left[\left\|\nabla \mathcal{L}_\mathcal{H}\left(\hat{\theta}\right)\right\|^2\right] \leq \upsilon \varepsilon_o$.

Theorem 5.2 demonstrates that Robust-DGD with ARC improves over the lower bound $\varepsilon_o$ when the ratio $f/n$ is sufficiently close to the breakdown point (BP), provided the local gradients at model initialization are bounded. This, in turn, leads to an improvement over the original learning guarantee of Robust-DGD (as stated in Lemma 2.6), which applies for arbitrary model initialization.

**Influence of model initialization and data heterogeneity.** Note that the smaller the bound on the initial gradients (i.e., the better the model initialization), the greater the improvement induced by ARC. Specifically, a decrease in $\zeta$ leads to a reduction in $\rho$, which subsequently lowers $\Psi(G, B, \rho)$.

As a result, for a fixed value of $\xi$ (i.e., for a fixed ratio of adversarial workers), the condition $\xi \leq \boldsymbol{v}/\Psi(G, B, \rho)$ is satisfied for a smaller $\boldsymbol{v}$, thereby yielding a lower stationarity error. This theoretical deduction is empirically validated in Section 5.2. A similar improvement occurs when increasing data heterogeneity, while keeping all other factors unchanged. Specifically, as $G$ increases, $\Psi(G, B, \rho)$ decreases, allowing for a smaller reduction factor $\boldsymbol{v}$ and thus a larger improvement in performance. This trend is also validated empirically in Figures 3 and 4a.

**Influence of $f/n$.** Additionally, for a fixed model initialization, an increase of $f/n$ towards the BP (i.e., as $\xi \to 0$) allows the reduction factor $\boldsymbol{v}$ to become smaller, leading to a greater improvement, as evidenced in Figures 1a and 4b. Indeed, we show in Theorem C.4 (a more complete version of Theorem 5.2) that ARC effectively increases the BP of $(f, \kappa)$-robust methods from $\frac{1}{2+B^2}$ to $\frac{1}{2}$, provided the initial honest gradients are bounded in norm. Lastly, we would like to recall that when the workers' gradients at model initialization are arbitrarily large, the convergence guarantee of Robust-DGD with ARC reduces to that of classic Robust-DGD (see Section 3).

**Practical scope of Theorem 5.2.** Since $\xi = \frac{\text{BP} - f/n}{\text{BP}}$ and $\boldsymbol{v} \geq \Psi(G, B, \rho)\xi$, it follows that the improvement factor $\boldsymbol{v}$ is at least $\Psi(G, B, \rho)\frac{\text{BP} - f/n}{\text{BP}}$. For ARC to induce an improvement, we require $\boldsymbol{v} < 1$, which reduces to having $f > n\text{BP}\left(1 - \frac{1}{\Psi(G,B,\rho)}\right)$. Since Theorem 5.2 assumes $f < n\text{BP}$, the improvement occurs when $f \in \left(n\text{BP}\left(1 - \frac{1}{\Psi(G,B,\rho)}\right), n\text{BP}\right)$. The length of this interval, i.e., $n\frac{\text{BP}}{\Psi(G,B,\rho)}$, depends on the size of the system $n$. In small systems where $n < \Psi(G, B, \rho)$, the length of the interval is smaller than 1 since $\text{BP} < 1/2$. Therefore, at most one value of $f$, i.e., the largest integer smaller than $n\text{BP}$, can lie in the improvement interval. Conversely, in large-scale systems where $n \in \Omega(\Psi(G, B, \rho)\log\Psi(G, B, \rho))$, the improvement interval expands significantly and can be much larger than 1, thereby demonstrating ARC's effectiveness in large-scale distributed learning.

## 5.2 INFLUENCE OF MODEL INITIALIZATION ON EMPIRICAL ROBUSTNESS

Figures 1b and 5 illustrate the effect of model initialization on the performance of Robust-DSGD with and without ARC, evaluated on MNIST with 10 honest workers. We investigate two regimes of heterogeneity: extreme heterogeneity and $\alpha = 0.1$, and consider $f = 1$ adversarial worker. The experiment proceeds as follows: we begin with the default model parameters initialized by PyTorch (i.e., the same ones used in Section 4), which represent a well-chosen set of initial parameters. These parameters are then scaled multiplicatively by a factor $\mu$ where larger values of $\mu$ correspond to progressively worse initialization, and vary $\mu \in \{1, ..., 5\}$. The results show that, under well-initialized conditions ($\mu = 1$), ARC significantly enhances the performance of Robust-DSGD, achieving a substantial improvement in worst-case maximal accuracy, particularly under extreme heterogeneity. In this regime, ARC boosts accuracy by about 70% compared to plain Robust-DSGD (Figure 1b). This ($\mu = 1$) corresponds to the initialization conditions of the empirical results presented in Section 4. As $\mu$ increases (i.e., the initialization worsens), the perfor-

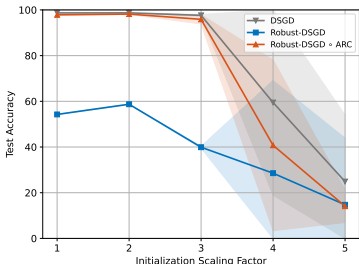

Figure 5: Worst-case maximal accuracies achieved by Robust-DSGD with CWTM, on MNIST ($\alpha = 0.1$), with 10 honest workers and 1 Byzantine worker. The x-axis represents worsening model initialization.

mance of ARC-enhanced Robust-DSGD gradually declines, with noticeable degradation starting from $\mu = 4$. Nevertheless, even at this point, ARC still offers a performance advantage over Robust-DSGD without clipping. By $\mu = 5$, the performance of Robust-DSGD with ARC converges to that of plain Robust-DSGD, both achieving an accuracy of around 10%. Another key observation from Figures 1b and 5 is that the behavior of Robust-DSGD with ARC closely mirrors that of Byzantine-free DSGD. Both exhibit similarly poor performance when $\mu = 5$, and their accuracies are comparable when $\mu \leq 3$, especially when $\alpha = 0.1$. This suggests that ARC is particularly effective at exploiting good model initialization, similar to the performance of DSGD in the absence of Byzantine workers. In contrast, plain Robust-DSGD struggles to fully leverage well-initialized models, as evidenced by its consistently lower accuracy (around 20%) in Figure 1b. These findings highlight the important influence of model initialization on the robustness of aggregation methods, and empirically validate our theoretical findings in Section 5.1.

## 6 RELATED WORK

Gradient clipping is a well-known technique for tackling exploding gradients in deep learning (Goodfellow et al., 2016). It has been extensively analyzed in the centralized setting (Zhang et al., 2020b;a; Koloskova et al., 2023), with applications to differential privacy (Pichapati et al., 2019). Moreover, we would like to note that gradient clipping has also been shown useful to obtain tighter convergence guarantees for stochastic gradient-descent methods in the case where the gradients have heavy-tailed noise (Gorbunov et al., 2020; 2022; Danilova, 2023). The considered proof techniques, however, do not apply directly to our adversarial distributed learning setting. A similar study on the impact of clipping also exists for distributed settings (Zhang et al., 2022; Khirirat et al., 2023).

In the context of robust distributed learning, prior work has proposed *iterative* clipping for robust aggregation (Karimireddy et al., 2021). The clipping scheme, however, has not been shown to induce any improvement over other SOTA robustness schemes. Recent work (Malinovsky et al., 2023) has proposed pre-aggregation clipping using temporal gradient differences, in conjunction with variance reduction, to tackle partial worker participation when adversarial workers can form a majority. We, however, propose and study pre-aggregation clipping as a tool to improve the robustness obtained by a general class of aggregation rules. Other prior work (Mhamdi et al., 2021; Farhadkhani et al., 2022; Allouah et al., 2023a) that used static pre-aggregation clipping to enhance the empirical robustness of Robust-DGD, did not provide any formal explanation.

While adaptive clipping schemes, similar to ARC, have been studied in the context of robust aggregation (Gupta & Vaidya, 2020; Liu et al., 2021; Diakonikolas et al., 2020; Abdalla & Zhivotovskiy, 2024), critical differences should be noted. The robustness guarantees in Gupta & Vaidya (2020); Liu et al. (2021) only apply to strongly convex loss functions, under the specific $2f$-*redundancy* condition. We consider non-convex loss functions, as well as a generic heterogeneous setting of $(G, B)$-gradient dissimilarity. In contrast to the clipping scheme proposed in Diakonikolas et al. (2020); Abdalla & Zhivotovskiy (2024), ARC only uses the $f$ parameter to tune the clipping threshold and does not rely on any additional a priori knowledge of the distribution of honest workers' gradients. Moreover, we prove a deterministic robustness property of ARC (see Theorem 3.2), whereas Diakonikolas et al. (2020); Abdalla & Zhivotovskiy (2024) only provides probability guarantees assuming the non-adversarial inputs to be i.i.d. with the distribution satisfying certain special properties.

Lastly, prior work (Guerraoui et al., 2021; Zhu & Ling, 2022; Allouah et al., 2023c; Choffrut et al., 2024; Allouah et al., 2024) has considered the challenge of privacy preservation alongside robustness in distributed learning. In this context, the gradients are assumed to have bounded norms in order to control the sensitivity of the algorithm, as is common in differential privacy (DP) (Abadi et al., 2016). This is often enforced in practice through static gradient clipping. Our findings suggest that an adaptive approach, such as ARC, may provide a more reliable alternative. However, a rigorous analysis of ARC in the context of DP remains an important avenue for future research.

## 7 CONCLUSION & DISCUSSION

We introduced Adaptive Robust Clipping (ARC), a pre-aggregation clipping scheme designed to harness the empirical benefits of gradient clipping, while preserving the worst-case optimal convergence guarantees of Robust-DGD. Unlike existing adaptive clipping schemes, ARC does not require additional tuning since it only relies on the standard robustness parameter, i.e., the tolerable fraction of adversarial workers. Through theoretical analysis, we explained ARC's ability to enhance the robustness of Robust-DGD, particularly when the model is well-initialized. This phenomenon was validated through comprehensive experiments on standard image classification datasets. In short, our experiments showed that ARC consistently boosts the performance of Robust-DGD, especially in scenarios with high heterogeneity and large fraction of adversarial workers.

**Future direction.** Our work also reveals a gap between theory and practice in Byzantine machine learning (ML). While Robust-DGD and ARC-enhanced Robust-DGD share the same worst-case convergence guarantees, their empirical performances are drastically different. In fact, and unsurprisingly, Byzantine ML theory focuses on worst-case guarantees, which, although essential, may not fully capture the practical realities of many learning settings. In practice, we often operate under favorable conditions (e.g., well-initialized models), where worst-case guarantees have limited relevance. This gap opens the door for future work that prioritizes practically-driven research in Byzantine ML, under conditions that are realizable in real-world scenarios. It encourages the development of robust distributed learning methods, like ARC, that can take full advantage of favorable practical conditions, thereby yielding relevant theoretical guarantees and superior empirical performance.

**Acknowledgment.** This work was supported in part by the FNS Project *Controlling the Spread of Epidemics: A Computing Perspective* (200021_200477) and the FNS Project *TruBrain* (20CH21_218778 / 1). We also extend our gratitude to the anonymous reviewers of the ICLR 2025 conference for their valuable insights and feedback.

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

# A    ADDITIONAL DETAILS ON ARC

This section provides further insights into the design and computational complexity of ARC, explaining its adaptive clipping strategy and its theoretical foundations. We also analyze its efficiency compared to static clipping.

## A.1    DESIGN OF ARC

Lemma B.1 in Appendix B shows that $\mathbf{F} \circ \mathbf{Clip}_C$ is $(f, \tilde{\kappa})$-robust, provided that $|S \setminus S_c| \geq 1$ for all subsets $S$ of size $n - f$. Note that this condition is impossible to guarantee when using a fixed clipping threshold that does not depend on the input vectors. In order to ensure that $|S \setminus S_c| \geq 1$ for all subsets $S$ of size $n - f$, the clipping threshold $C$ should be large enough such that less than $n - f$ input vectors are clipped. This motivates the design of an adaptive clipping strategy. Accordingly, we propose to choose a clipping threshold such that the total number of clipped vectors is of the form $\lfloor \lambda(n - f) \rfloor$, where $\lambda < 1$. Note that it is natural to clip more vectors as the fraction of adversarial workers $f/n$ increases, to control the impact of Byzantine behavior. Therefore, we set $\lambda := \zeta \frac{f}{n}$ where $0 \leq \zeta \leq 2$. Since $\frac{f}{n} < \frac{1}{2}$ and $\lambda < 1$, the total number of clipped vectors $\left\lfloor \zeta \frac{f}{n}(n - f) \right\rfloor < n - f$ for all $\zeta \in [0, 2]$. This constitutes the underlying idea behind our adaptive clipping strategy ARC, presented in Algorithm 2. Notably, our theoretical results hold for all $\zeta \in [0, 2]$. However, empirical evaluations show that setting $\zeta = 2$ consistently provides the best performance, leading us to adopt this value in Algorithm 2 and throughout the paper.

## A.2    COMPUTATIONAL COMPLEXITY OF ARC

The computational complexity of ARC is comparable to static clipping, with the primary difference being an additional sorting step. The overall complexity can be broken down as follows:

1. Norm computation: Computing the norms of all input vectors requires $\mathcal{O}(nd)$ time. (This step is also required in static clipping.)

2. Sorting step: Sorting the norms incurs an additional $\mathcal{O}(n \log n)$ time. If $n$ is large relative to $k = \lfloor 2(f/n)(n - f) \rfloor$, an efficient selection algorithm such as quick-select (Hoare, 1961) can be used instead, reducing the complexity to $\mathcal{O}(n)$ in the average case.

3. Gradient clipping: Clipping vectors requires $\mathcal{O}(nd)$ time.

Thus, the total time complexity of ARC is:

$$\mathcal{O}(nd + n \log n).$$

Crucially, this complexity remains *dimension-independent* in the sorting step, making it scalable while maintaining robustness.

# B    PROOFS FOR RESULTS IN SECTION 3

Proofs for Lemma 3.1 and Theorem 3.2 are presented in B.1 and B.2, respectively.

**Notation**  Let $n \in \mathbb{N}^*$. Given vectors $x_1, \ldots, x_n \in \mathbb{R}^d$ and a set $S \subseteq [n]$, we denote by $\overline{x}_S$ the mean of the vectors in $S$ $\overline{x}_S = \frac{1}{|S|} \sum_{i \in S} x_i$. Given a clipping parameter $C \geq 0$, let

$$y_i := \mathrm{clip}_C(x_i) = \min \left(1, \frac{C}{\|x_i\|}\right) x_i,$$

and

$$\overline{y}_S := \frac{1}{|S|} \sum_{i \in S} y_i.$$

We denote by $S_c$ the set of clipped vectors in $S$,

$$S_c := \{i \in S, \|x_i\| > C\}. \tag{1}$$

Recall that we denote by $\mathbf{Clip}_C$ be the operator such that, for $x_1, \ldots, x_n \in \mathbb{R}^d$

$$\mathbf{Clip}_C(x_1, \ldots, x_n) := (\mathrm{clip}_C(x_1), \ldots, \mathrm{clip}_C(x_n)) .$$

Further, let $\mathbf{F} : \mathbb{R}^{n \times d} \to \mathbb{R}^d$. We denote by $\mathbf{F} \circ \mathbf{Clip}_C$ the aggregation rule that first clips the input vectors using parameter $C$, and then aggregates them using $\mathbf{F}$

$$\mathbf{F} \circ \mathbf{Clip}_C(x_1, \ldots, x_n) := \mathbf{F}(\mathbf{Clip}_C(x_1, \ldots, x_n)) .$$

## B.1 PROOF OF LEMMA 3.1

**Lemma 3.1.** *For any fixed $C \in \mathbb{R}^+$ and $\kappa' \geq 0$, $\mathbf{F} \circ \mathbf{Clip}_C$ is not $(f, \kappa')$-robust.*

*Proof.* We use reasoning by contradiction to prove the lemma.

We first consider the case when $C > 0$. Let $S := \{1, \ldots, n - f\}$. Consider an arbitrary set of $n$ vectors $x_1, \ldots, x_n$ in $\mathbb{R}^d$ such that $x_1 = \ldots = x_{n-f} = \overline{x}_S$ and $\|\overline{x}_S\| = 2C$. We assume that that $\mathbf{F} \circ \mathbf{Clip}_C$ is $(f, \kappa')$-robust for some $\kappa' \geq 0$. This assumption implies that

$$\|\mathbf{F} \circ \mathbf{Clip}_C(x_1, \ldots, x_n) - \overline{x}_S\|^2 \leq \frac{\kappa'}{|S|} \sum_{i \in S} \|x_i - \overline{x}_S\|^2 = 0. \tag{2}$$

Therefore,

$$\mathbf{F} \circ \mathbf{Clip}_C(x_1, \ldots, x_n) = \overline{x}_S.$$

However, the clipping operation results in $\mathrm{clip}_C(x_1) = \ldots = \mathrm{clip}_C(x_{n-f}) = \frac{1}{2}\overline{x}_S$. Therefore, by $(f, \kappa)$-robustness of $\mathbf{F}$,

$$\mathbf{F} \circ \mathbf{Clip}_C(x_1, \ldots, x_n) = \mathbf{F}\left(\frac{1}{2}\overline{x}_S, \ldots, \frac{1}{2}\overline{x}_S, \mathrm{clip}_C(x_{n-f+1}), \ldots, \mathrm{clip}_C(x_n)\right) = \frac{1}{2}\overline{x}_S.$$

This contradicts (2). As this contradiction holds for any value of $\kappa'$, $\mathbf{F} \circ \mathbf{Clip}_C$ cannot be $(f, \kappa')$-robust for any $\kappa'$.

The proof for the case when $C = 0$ is similar to the above, where we choose $x_1 = \ldots = x_{n-f} = \overline{x}_S$ such that $\overline{x}_S$ is any vector with strictly positive norm. $\square$

## B.2 PROOF OF THEOREM 3.2

Our proof relies on the following lemma, proof of which is deferred to B.2.1.

**Lemma B.1.** *Let $C \in \mathbb{R}^+$ and $\mathbf{F}$ be an $(f, \kappa)$-robust aggregation rule. Consider an arbitrary set of $n$ vectors $x_1, x_2, \ldots, x_n \in \mathbb{R}^d$ and an arbitrary $S \subseteq [n]$ with $|S| = n - f$. Let $S_c$ denote the set of indices of clipped vectors in $S$, i.e., $S_c := \{i \in S, \|x_i\| > C\}$. If $|S \setminus S_c| \geq 1$, then*

$$\|\mathbf{F} \circ \mathbf{Clip}_C(x_1, \ldots, x_n) - \overline{x}_S\|^2 \leq \frac{\tilde{\kappa}}{|S|} \sum_{i \in S} \|x_i - \overline{x}_S\|^2 ,$$

*where $\tilde{\kappa} = \kappa + \frac{|S_c|}{|S \setminus S_c|}$.*

Lemma B.1 shows that $\mathbf{F} \circ \mathbf{Clip}_C$ is $(f, \tilde{\kappa})$-robust, provided that $|S \setminus S_c| \geq 1$ for all subsets $S$ of size $n - f$. Note that this condition is impossible to guarantee when using a fixed clipping threshold that does not depend on the input vectors. In order to ensure that $|S \setminus S_c| \geq 1$ for all subsets $S$ of size $n - f$, the clipping threshold $C$ should be large enough such that less than $n - f$ input vectors are clipped. By construction, ARC satisfies the condition of Lemma B.1. This brings us to the proof of Theorem 3.2, which we recall below for convenience.

**Theorem 3.2.** *If $\mathbf{F}$ is $(f, \kappa)$-robust, then $\mathbf{F} \circ \mathbf{ARC}$ is $\left(f, \kappa + \frac{2f}{n-2f}\right)$-robust.*

*Proof.* Since we clip the largest $\lfloor 2(f/n)(n - f) \rfloor$ gradients, for a given $S \subseteq [n]$ with $|S| = n - f$ we have

$$|S_c| \leq \lfloor 2(f/n)(n - f) \rfloor \leq 2(f/n)(n - f).$$

Therefore,

$$|S \setminus S_c| = |S| - |S_c| \geq (n - f) - \frac{2f}{n}(n - f) = (n - f)\frac{n - 2f}{n}.$$

Since it is assumed that $f < n/2$, we have

$$|S \setminus S_c| > \left(n - \frac{n}{2}\right)\frac{n - n}{n} = 0.$$

Thus, the condition $|S \setminus S_c| \geq 1$ is always verified. Hence, from Lemma B.1 we obtain that $\mathbf{F} \circ \mathbf{ARC}$ is $\left(f, \left(\kappa + \frac{|S_c|}{|S \setminus S_c|}\right)\right)$-robust where

$$\frac{|S_c|}{|S \setminus S_c|} \leq \frac{2(f/n)(n - f)}{(n - f)\frac{n - 2f}{n}} = \frac{2f}{n - 2f}.$$

This concludes the proof. □

### B.2.1  PROOF OF LEMMA B.1

We start by giving bounds on the **variance** (Lemma B.2) and the **bias** (Lemma B.3 and Lemma B.4) due to the clipping operation. We combine these bounds to prove the theorem.

#### VARIANCE REDUCTION DUE TO CLIPPING

We start by giving a bound on the variance of the clipped vectors. Recall from (1) that $S_c := \{i \in S, \|x_i\| > C\}$.

**Lemma B.2.** *Let $C \geq 0$, $n > 0$ and $f < \frac{n}{2}$. For all $S \subseteq [n]$ with $|S| = n - f$, the following holds true:*

1. *If $\|\overline{x}_S\| \leq C$ then*

$$\frac{1}{|S|}\sum_{i \in S}\|y_i - \overline{y}_S\|^2 \leq \frac{1}{|S|}\sum_{i \in S}\|x_i - \overline{x}_S\|^2 - \frac{1}{|S|}\sum_{i \in S_c}(\|x_i\| - C)^2.$$

2. *If $\|\overline{x}_S\| > C$ then*

$$\frac{1}{|S|}\sum_{i \in S}\|y_i - \overline{y}_S\|^2 \leq \frac{1}{|S|}\sum_{i \in S}\|x_i - \overline{x}_S\|^2 - \frac{|S \setminus S_c|}{|S|}(\|\overline{x}_S\| - C)^2 - \frac{1}{|S|}\sum_{i \in S_c}(\|x_i\| - \|\overline{x}_S\|)^2.$$

*Proof.* Note that

$$\frac{1}{|S|}\sum_{i \in S}\|y_i - \overline{y}_S\|^2 = \frac{1}{|S|}\sum_{i \in S}\|y_i - \overline{x}_S + \overline{x}_S - \overline{y}_S\|^2$$

$$= \frac{1}{|S|}\sum_{i \in S}\left(\|y_i - \overline{x}_S\|^2 + \|\overline{x}_S - \overline{y}_S\|^2 + 2\langle y_i - \overline{x}_S, \overline{x}_S - \overline{y}_S\rangle\right)$$

$$= \frac{1}{|S|}\sum_{i \in S}\|y_i - \overline{x}_S\|^2 + \|\overline{x}_S - \overline{y}_S\|^2 + 2\left\langle \underbrace{\frac{1}{|S|}\sum_{i \in S}y_i}_{\overline{y}_S} - \overline{x}_S, \overline{x}_S - \overline{y}_S\right\rangle$$

$$= \frac{1}{|S|}\sum_{i \in S}\|y_i - \overline{x}_S\|^2 + \|\overline{x}_S - \overline{y}_S\|^2 - 2\langle \overline{x}_S - \overline{y}_S, \overline{x}_S - \overline{y}_S\rangle$$

$$= \frac{1}{|S|}\sum_{i \in S}\|y_i - \overline{x}_S\|^2 - \|\overline{x}_S - \overline{y}_S\|^2. \tag{3}$$

By the definition of $S_c$ (in (1)), for all $i \in S \setminus S_c$, $y_i = x_i$. Therefore,

$$\frac{1}{|S|} \sum_{i \in S} \|y_i - \overline{x}_S\|^2 = \frac{1}{|S|} \sum_{i \in S \setminus S_c} \|y_i - \overline{x}_S\|^2 + \frac{1}{|S|} \sum_{i \in S_c} \|y_i - \overline{x}_S\|^2$$

$$= \frac{1}{|S|} \sum_{i \in S \setminus S_c} \|x_i - \overline{x}_S\|^2 + \frac{1}{|S|} \sum_{i \in S_c} \|y_i - \overline{x}_S\|^2.$$

The above can be written as

$$\frac{1}{|S|} \sum_{i \in S} \|y_i - \overline{x}_S\|^2 = \frac{1}{|S|} \sum_{i \in S} \|x_i - \overline{x}_S\|^2 + \frac{1}{|S|} \sum_{i \in S_c} (\|y_i - \overline{x}_S\|^2 - \|x_i - \overline{x}_S\|^2). \qquad (4)$$

For $i \in S_c$, we have $y_i = \frac{C}{\|x_i\|} x_i$. Thus, for all $i \in S_c$, we obtain that

$$\|y_i - \overline{x}_S\|^2 - \|x_i - \overline{x}_S\|^2 = \underbrace{\|y_i\|^2}_{=C^2} + \|\overline{x}_S\|^2 - 2\langle y_i, \overline{x}_S \rangle - \|x_i\|^2 - \|\overline{x}_S\|^2 + 2\langle x_i, \overline{x}_S \rangle$$

$$= C^2 - \|x_i\|^2 + 2 \left( 1 - \frac{C}{\|x_i\|} \right) \langle x_i, \overline{x}_S \rangle$$

$$= -(\|x_i\| - C)(\|x_i\| + C) + 2(\|x_i\| - C) \frac{\langle x_i, \overline{x}_S \rangle}{\|x_i\|}$$

$$= (\|x_i\| - C) \left( 2 \frac{\langle x_i, \overline{x}_S \rangle}{\|x_i\|} - \|x_i\| - C \right).$$

By Cauchy-Schwarz inequality, we have $\langle x_i, \overline{x}_S \rangle \leq \|x_i\| \|\overline{x}_S\|$ Therefore,

$$\|y_i - \overline{x}_S\|^2 - \|x_i - \overline{x}_S\|^2 \leq (\|x_i\| - C) (2\|\overline{x}_S\| - \|x_i\| - C).$$

Substituting from the above in (4), we obtain that

$$\frac{1}{|S|} \sum_{i \in S} \|y_i - \overline{x}_S\|^2 \leq \frac{1}{|S|} \sum_{i \in S} \|x_i - \overline{x}_S\|^2 + \frac{1}{|S|} \sum_{i \in S_c} (\|x_i\| - C) (2\|\overline{x}_S\| - \|x_i\| - C).$$

Substituting from the above in (3), we obtain that

$$\frac{1}{|S|} \sum_{i \in S} \|y_i - \overline{y}_S\|^2 \leq \frac{1}{|S|} \sum_{i \in S} \|x_i - \overline{x}_S\|^2 + \frac{1}{|S|} \sum_{i \in S_c} (\|x_i\| - C) (2\|\overline{x}_S\| - \|x_i\| - C) - \|\overline{x}_S - \overline{y}_S\|^2.$$

$$(5)$$

We now consider below the two cases: $\|\overline{x}_S\| \leq C$ and $\|\overline{x}_S\| > C$.

In the first case, i.e,. when $\|\overline{x}_S\| \leq C$, we have

$$\frac{1}{|S|} \sum_{i \in S_c} (\|x_i\| - C) (2\|\overline{x}_S\| - \|x_i\| - C) \leq \frac{1}{|S|} \sum_{i \in S_c} (\|x_i\| - C) (2C - \|x_i\| - C)$$

$$\leq -\frac{1}{|S|} \sum_{i \in S_c} (\|x_i\| - C)^2.$$

Substituting from the above in (5) yields the following

$$\frac{1}{|S|} \sum_{i \in S} \|y_i - \overline{y}_S\|^2 \leq \frac{1}{|S|} \sum_{i \in S} \|x_i - \overline{x}_S\|^2 - \frac{1}{|S|} \sum_{i \in S_c} (\|x_i\| - C)^2 - \|\overline{x}_S - \overline{y}_S\|^2$$

$$\leq \frac{1}{|S|} \sum_{i \in S} \|x_i - \overline{x}_S\|^2 - \frac{1}{|S|} \sum_{i \in S_c} (\|x_i\| - C)^2.$$

This proves the first part of the lemma.

Consider the second case, i.e., $\|\overline{x}_S\| > C$. Note that

$$\frac{1}{|S|}\sum_{i \in S_c}(\|x_i\| - C)\,(2\|\overline{x}_S\| - \|x_i\| - C) = \frac{1}{|S|}\sum_{i \in S_c}\left((\|\overline{x}_S\| - C)^2 - (\|x_i\| - \|\overline{x}_S\|)^2\right)$$

$$= \frac{|S_c|}{|S|}(\|\overline{x}_S\| - C)^2 - \frac{1}{|S|}\sum_{i \in S_c}(\|x_i\| - \|\overline{x}_S\|)^2. \quad (6)$$

Since $\|\overline{x}_S\| > C$, and $\|\overline{y}_S\| \leq C$, we have $\|\overline{x}_S\| - \|\overline{y}_S\| \geq \|\overline{x}_S\| - C \geq 0$. This, in conjunction with the reverse triangle inequality, implies that

$$\|\overline{x}_S - \overline{y}_S\|^2 \geq (\|\overline{x}_S\| - \|\overline{y}_S\|)^2 \geq (\|\overline{x}_S\| - C)^2. \quad (7)$$

Substituting from (6) and (7) in (5) yields the following:

$$\frac{1}{|S|}\sum_{i \in S}\|y_i - \overline{y}_S\|^2 \leq \frac{1}{|S|}\sum_{i \in S}\|x_i - \overline{x}_S\|^2 + \frac{|S_c|}{|S|}(\|\overline{x}_S\| - C)^2 - \frac{1}{|S|}\sum_{i \in S_c}(\|x_i\| - \|\overline{x}_S\|)^2 - (\|\overline{x}_S\| - C)^2$$

$$\leq \frac{1}{|S|}\sum_{i \in S}\|x_i - \overline{x}_S\|^2 + \left(\frac{|S_c|}{|S|} - 1\right)(\|\overline{x}_S\| - C)^2 - \frac{1}{|S|}\sum_{i \in S_c}(\|x_i\| - \|\overline{x}_S\|)^2$$

$$= \frac{1}{|S|}\sum_{i \in S}\|x_i - \overline{x}_S\|^2 - \frac{|S \setminus S_c|}{|S|}(\|\overline{x}_S\| - C)^2 - \frac{1}{|S|}\sum_{i \in S_c}(\|x_i\| - \|\overline{x}_S\|)^2.$$

This proves the second part, which concludes the proof of the lemma. $\qquad\square$

BIAS DUE TO CLIPPING

We now bound the bias induced by clipping the input vectors.

**Lemma B.3.** *Let* $C \geq 0$, $n > 0$, $f < \frac{n}{2}$, *and* $S \subseteq [n]$, $|S| = n - f$. *Then,*

$$\|\overline{x}_S - \overline{y}_S\|^2 \leq \frac{|S_c|}{|S|^2}\sum_{i \in S_c}(\|x_i\| - C)^2.$$

*Proof.* Note that

$$\|\overline{x}_S - \overline{y}_S\|^2 = \left\|\frac{1}{|S|}\sum_{i \in S}x_i - \frac{1}{|S|}\sum_{i \in S}y_i\right\|^2 = \left\|\frac{1}{|S|}\sum_{i \in S}(x_i - y_i)\right\|^2.$$

As $x_i = y_i$ for all $i \in S \setminus S_c$, we have

$$\|\overline{x}_S - \overline{y}_S\|^2 = \frac{1}{|S|^2}\left\|\sum_{i \in S_c}(x_i - y_i)\right\|^2.$$

Due to Jensen's inequality,

$$\|\overline{x}_S - \overline{y}_S\|^2 = \frac{|S_c|^2}{|S|^2}\left\|\frac{1}{|S_c|}\sum_{i \in S_c}(x_i - y_i)\right\|^2 \leq \frac{|S_c|}{|S|^2}\sum_{i \in S_c}\|x_i - y_i\|^2.$$

As $y_i = \frac{C}{\|x_i\|}x_i$ for all $i \in S_c$, substituting this in the above proves the lemma. $\qquad\square$

We now show that the bias is upper bounded by a multiplicative factor of the variance of the input vectors, as long as there is at least one unclipped honest vector.

**Lemma B.4** (Bias due to clipping)**.** $C \geq 0$, $n > 0$, $f < {}^n/2$, *and* $S \subseteq [n]$, $|S| = n - f$, *if* $|S \setminus S_c| \geq 1$ *then*

$$\|\overline{x}_S - \overline{y}_S\|^2 \leq \frac{|S_c|}{|S \setminus S_c|}\frac{1}{|S|}\sum_{i \in S}\|x_i - \overline{x}_S\|^2.$$

*Proof.* We assume throughout the proof that $|S_c| > 0$. Otherwise, if $|S_c| = 0$, the bias is $0$ and the statement is trivially true.

By Lemma B.3, we have

$$\|\overline{x}_S - \overline{y}_S\|^2 \leq \frac{|S_c|}{|S|^2} \sum_{i \in S_c} (\|x_i\| - C)^2. \tag{8}$$

We distinguish two cases: $\|\overline{x}_S\| \leq C$ and $\|\overline{x}_S\| > C$. In the first case we have that

$$0 \leq \|x_i\| - C \leq \|x_i\| - \|\overline{x}_S\|.$$

Substituting the above in (8), we find

$$\|\overline{x}_S - \overline{y}_S\|^2 \leq \frac{|S_c|}{|S|^2} \sum_{i \in S_c} (\|x_i\| - \|\overline{x}_S\|)^2 \leq \frac{|S_c|}{|S|^2} \sum_{i \in S} (\|x_i\| - \|\overline{x}_S\|)^2.$$

Using the reverse triangle inequality, we have that $(\|x_i\| - \|\overline{x}_S\|)^2 \leq \|x_i - \overline{x}_S\|^2$. This implies that

$$\|\overline{x}_S - \overline{y}_S\|^2 \leq \frac{|S_c|}{|S|^2} \sum_{i \in S} (\|x_i\| - \|\overline{x}_S\|)^2 \leq \frac{|S_c|}{|S \setminus S_c|} \frac{1}{|S|} \sum_{i \in S} \|x_i - \overline{x}_S\|^2.$$

This proves the result for the first case.

Consider now the second case, i.e $\|\overline{x}_S\| > C$. Noting that we assume that $|S \setminus S_c| \geq 1$ and using Young's inequality with $c = \frac{|S_c|}{|S \setminus S_c|}$, we find, for any $i \in S_c$,

$$(\|x_i\| - C)^2 = (\|x_i\| - \|\overline{x}_S\| + \|\overline{x}_S\| - C)^2$$
$$\leq (1 + c)(\|x_i\| - \|\overline{x}_S\|)^2 + (1 + 1/c)(\|\overline{x}_S\| - C)^2)$$
$$= \frac{|S_c| + |S \setminus S_c|}{|S \setminus S_c|} (\|x_i\| - \|\overline{x}_S\|)^2 + \frac{|S_c| + |S \setminus S_c|}{|S_c|} (\|\overline{x}_S\| - C)^2.$$

Recall from (1) that $S_c := \{i \in S, \|x_i\| > C\}$. This implies that $|S_c| + |S \setminus S_c| = |S|$. Therefore

$$(\|x_i\| - C)^2 \leq |S| \left( \frac{1}{|S \setminus S_c|} (\|x_i\| - \|\overline{x}_S\|)^2 + \frac{1}{|S_c|} (\|\overline{x}_S\| - C)^2 \right). \tag{9}$$

Substituting (9) in (8), we find

$$\|\overline{x}_S - \overline{y}_S\|^2 \leq \frac{|S_c|}{|S|^2} \sum_{i \in S_c} |S| \left( \frac{1}{|S \setminus S_c|} (\|x_i\| - \|\overline{x}_S\|)^2 + \frac{1}{|S_c|} (\|\overline{x}_S\| - C)^2 \right)$$
$$= \frac{|S_c|}{|S|} \left( \frac{1}{|S \setminus S_c|} \sum_{i \in S_c} (\|x_i\| - \|\overline{x}_S\|)^2 + (\|\overline{x}_S\| - C)^2 \right). \tag{10}$$

Since $\|\overline{x}_S\| > C$, we have for any $i \in S \setminus S_c$

$$0 \leq \|\overline{x}_S\| - C \leq \|\overline{x}_S\| - \|x_i\|.$$

Therefore

$$|S \setminus S_c| (\|\overline{x}_S\| - C)^2 \leq \sum_{i \in S \setminus S_c} (\|\overline{x}_S\| - \|x_i\|)^2$$

Which gives the bound

$$(\|\overline{x}_S\| - C)^2 \leq \frac{1}{|S \setminus S_c|} \sum_{i \in S \setminus S_c} (\|\overline{x}_S\| - \|x_i\|)^2. \tag{11}$$

Substituting (11) in (10), we find

$$\|\overline{x}_S - \overline{y}_S\|^2 \leq \frac{|S_c|}{|S|} \left( \frac{1}{|S \setminus S_c|} \sum_{i \in S_c} (\|x_i\| - \|\overline{x}_S\|)^2 + \frac{1}{|S \setminus S_c|} \sum_{i \in S \setminus S_c} (\|x_i\| - \|\overline{x}_S\|)^2 \right)$$
$$= \frac{|S_c|}{|S \setminus S_c|} \frac{1}{|S|} \sum_{i \in S} (\|x_i\| - \|\overline{x}_S\|)^2 \leq \frac{|S_c|}{|S \setminus S_c|} \frac{1}{|S|} \sum_{i \in S} \|x_i - \overline{x}_S\|^2.$$

This proves the result for the second case and concludes the proof. $\square$

PROOF OF LEMMA B.1

Let us recall the lemma below.

**Lemma B.1.** *Let $C \in \mathbb{R}^+$ and $\mathbf{F}$ be an $(f, \kappa)$-robust aggregation rule. Consider an arbitrary set of $n$ vectors $x_1, x_2, \ldots, x_n \in \mathbb{R}^d$ and an arbitrary $S \subseteq [n]$ with $|S| = n - f$. Let $S_c$ denote the set of indices of clipped vectors in $S$, i.e., $S_c := \{i \in S, \|x_i\| > C\}$. If $|S \setminus S_c| \geq 1$, then*

$$\|\mathbf{F} \circ \mathbf{Clip}_C(x_1, \ldots, x_n) - \overline{x}_S\|^2 \leq \frac{\tilde{\kappa}}{|S|} \sum_{i \in S} \|x_i - \overline{x}_S\|^2 \quad,$$

*where $\tilde{\kappa} = \kappa + \frac{|S_c|}{|S \setminus S_c|}$.*

*Proof.* We assume throughout the proof that $|S_c| > 0$. Otherwise, the statement is trivially true by $(f, \kappa)$-robustness of $\mathbf{F}$ and the fact that the vectors in $S$ remain unchanged after the clipping operation.

Consider first the case $\kappa = 0$. Since $\kappa \geq \frac{f}{n - 2f}$ [2], we have that $f = 0$. Recall that we denote $y_i := \text{clip}_C(x_i)$ and $\overline{y}_S := \frac{1}{|S|} \sum_{i \in S} y_i$. Since $\kappa = 0$, the output of $\mathbf{F}$ will correspond to the average of its input, which implies that

$$\|\mathbf{F} \circ \mathbf{Clip}_C(x_1, \ldots, x_n) - \overline{y}_S\|^2 = \|\mathbf{F}(y_1, \ldots, y_n) - \overline{y}_S\|^2 = 0.$$

Therefore

$$\mathbf{F} \circ \mathbf{Clip}_C(x_1, \ldots, x_n) = \overline{y}_S.$$

We find then

$$\|\mathbf{F} \circ \mathbf{Clip}_C(x_1, \ldots, x_n) - \overline{x}_S\|^2 = \|\overline{y}_S - \overline{x}_S\|^2.$$

The result for the case $\kappa = 0$ follows from lemma B.4.

Suppose now that $\kappa > 0$. Using Young's inequality with $c = \frac{|S_c|}{\kappa |S \setminus S_c|}$ we find

$$\|\mathbf{F} \circ \mathbf{Clip}_C(x_1, \ldots, x_n) - \overline{x}_S\|^2 = \|\mathbf{F} \circ \mathbf{Clip}_C(x_1, \ldots, x_n) - \overline{y}_S + \overline{y}_S - \overline{x}_S\|^2$$

$$\leq (1 + c) \|\mathbf{F} \circ \mathbf{Clip}_C(x_1, \ldots, x_n) - \overline{y}_S\|^2 + \left(1 + \frac{1}{c}\right) \|\overline{x}_S - \overline{y}_S\|^2. \tag{12}$$

On the one hand, by $(f, \kappa)$-robustness of $\mathbf{F}$, we have

$$(1 + c) \|\mathbf{F} \circ \mathbf{Clip}_C(x_1, \ldots, x_n) - \overline{y}_S\|^2 = (1 + c) \|\mathbf{F}(y_1, \ldots, y_n) - \overline{y}_S\|^2$$

$$\leq (1 + c) \frac{\kappa}{|S|} \sum_{i \in S} \|y_i - \overline{y}_S\|^2$$

$$= \left(\kappa + \frac{|S_c|}{|S \setminus S_c|}\right) \frac{1}{|S|} \sum_{i \in S} \|y_i - \overline{y}_S\|^2. \tag{13}$$

On the other hand, we have

$$(1 + \text{1/}c) \|\overline{x}_S - \overline{y}_S\|^2 = \left(1 + \frac{\kappa |S \setminus S_c|}{|S_c|}\right) \|\overline{x}_S - \overline{y}_S\|^2 = \left(\kappa + \frac{|S_c|}{|S \setminus S_c|}\right) \frac{|S \setminus S_c|}{|S_c|} \|\overline{x}_S - \overline{y}_S\|^2. \tag{14}$$

Substituting (13) and (14) in (12) we obtain that

$$\|\mathbf{F} \circ \mathbf{Clip}_C(x_1, \ldots, x_n) - \overline{x}_S\|^2 \leq \left(\kappa + \frac{|S_c|}{|S \setminus S_c|}\right) \left(\frac{1}{|S|} \sum_{i \in S} \|y_i - \overline{y}_S\|^2 + \frac{|S \setminus S_c|}{|S_c|} \|\overline{x}_S - \overline{y}_S\|^2\right). \tag{15}$$

---

[2] Proposition 6 (Allouah et al., 2023a)

We consider below the two cases $\|\overline{x}_S\| \le C$ and $\|\overline{x}_S\| > C$ separately.

In the first case, we use the the first part of lemma B.2 and lemma B.3 to obtain that

$$\frac{1}{|S|} \sum_{i \in S} \|y_i - \overline{y}_S\|^2 + \frac{|S \setminus S_c|}{|S|} \|\overline{x}_S - \overline{y}_S\|^2$$

$$\le \frac{1}{|S|} \sum_{i \in S} \|x_i - \overline{x}_S\|^2 - \frac{1}{|S|} \sum_{i \in S_c} (\|x_i\| - C)^2 + \frac{|S \setminus S_c|}{|S|} \frac{1}{|S|} \sum_{i \in S_c} (\|x_i\| - C)^2$$

$$= \frac{1}{|S|} \sum_{i \in S} \|x_i - \overline{x}_S\|^2 - \frac{|S_c|}{|S|} \frac{1}{|S|} \sum_{i \in S} (\|x_i\| - C)^2 \le \frac{1}{|S|} \sum_{i \in S} \|x_i - \overline{x}_S\|^2.$$

This proves the result for this case.

Consider the second case, i.e $\|\overline{x}_S\| > C$. Following the proof of corollary B.4 until (10) we have

$$\|\overline{x}_S - \overline{y}_S\|^2 \le \frac{|S_c|}{|S|} \left( \frac{1}{|S \setminus S_c|} \sum_{i \in S_c} (\|x_i\| - \|\overline{x}_S\|)^2 + (\|\overline{x}_S\| - C)^2 \right).$$

Therefore,

$$\frac{|S \setminus S_c|}{|S_c|} \|\overline{x}_S - \overline{y}_S\|^2 \le \frac{|S \setminus S_c|}{|S|} \left( \frac{1}{|S \setminus S_c|} \sum_{i \in S_c} (\|x_i\| - \|\overline{x}_S\|)^2 + (\|\overline{x}_S\| - C)^2 \right)$$

$$\le \frac{1}{|S|} \sum_{i \in S_c} (\|x_i\| - \|\overline{x}_S\|)^2 + \frac{|S \setminus S_c|}{|S|} (\|\overline{x}_S\| - C)^2.$$

Using the above in conjunction with lemma B.2 for the case $\|\overline{x}_S\| > C$ in (15) proves the result for the second case, which concludes the proof. □

## C PROOFS OF THE RESULTS IN SECTION 5

In this section, we first prove the *Bounded Aggregation Output* property of ARC in Lemma 5.1, and show that the other SOTA aggregation rules do not satisfy it in Lemma C.1. Then, to prove Theorem 5.2, we first prove Lemma C.2. We assume throughout this appendix that for any set of vectors $x_1, \ldots, x_n \in \mathbb{R}^d$, $\|\mathbf{F}(x_1, \ldots, x_n)\| \le \max_{i \in [n]} \|x_i\|$. This assumption can be made without loss of generality, see Appendix C.1.

**Lemma 5.1 (Bounded Output).** *Let* $\mathbf{F}$ *be an* $(f, \kappa)$-*robust aggregation method. For any vectors* $x_1, \ldots, x_n \in \mathbb{R}^d$ *and set* $S \subset [n]$ *such that* $|S| = n - f$, $\|\mathbf{F} \circ \mathbf{ARC}(x_1, \ldots, x_n)\| \le \max_{i \in S} \|x_i\|$.

*Proof.* Without loss of generality and for the sake of simplicity, let us suppose that the vectors $x_1, \ldots, x_n$ are indexed such that $\|x_1\| \ge \|x_2\| \ge \cdots \ge \|x_n\|$. Let $(\tilde{x}_1, \ldots, \tilde{x}_n) = ARC(x_1, \ldots, x_n)$ be the clipped version of these vectors after applying ARC.

**Case 1:** When $f = 0$, $S = [n]$ and we clip all the vectors by $\|x_1\| = \max_{i \in S} \|x_i\|$, hence for any $j$, $\|\tilde{x}_j\| \le \max_{i \in S} \|x_i\|$. Using the fact that for any set of vectors $x_1, \ldots, x_n \in \mathbb{R}^d$, $\|\mathbf{F}(x_1, \ldots, x_n)\| \le \max_{i \in [n]} \|x_i\|$, we have

$$\|\mathbf{F} \circ ARC(x_1, \ldots, x_n)\| = \|\mathbf{F}(\tilde{x}_1, \ldots, \tilde{x}_n)\| \le \max_{j \in [n]} \|\tilde{x}_j\| \le \max_{i \in S} \|x_i\| . \tag{16}$$

**Case 2:** When $0 < f < \frac{n}{2}$, as presented in Algorithm 2, ARC clips all the vectors $x_1, \ldots, x_n$ to $\|x_{k+1}\|$ where $k = \lfloor 2\frac{f}{n}(n-f) \rfloor$. Hence, for any $j \in [n]$,

$$\|\tilde{x}_j\| \le \|x_{k+1}\| . \tag{17}$$

We have that $k + 1 = \lfloor 2\frac{f}{n}(n-f) \rfloor + 1 \ge 2\frac{f}{n}(n-f)$.

One can show that

$$\frac{2f}{n}(n-f) - f = f(1 - \frac{2f}{n})$$
$$> 0, \quad \forall f \in \left] 0, \frac{n}{2} \right[ \tag{18}$$

Hence, $\frac{2f}{n}(n-f) > f$, which gives $k+1 > f$, and since $k+1$ is an integer, we have $k+1 \geq f+1$. Hence we know that at least $f+1$ vectors have their norm greater or equal to the clipping threshold $C = \|x_{k+1}\|$,

$$\underbrace{\|x_1\| \geq \cdots \geq \|x_{k+1}\|}_{\text{at least } f+1 \text{ vectors}} \geq \underbrace{\|x_{k+2}\| \geq \cdots \geq \|x_n\|}_{\text{at most } n-f-1 \text{ vectors}}$$

Hence, at most $n-f-1$ vectors will not be clipped. For any subset of $S \subset [n]$ of size $n-f$, at least one of the vectors has its norm equal to the clipping threshold (i.e. $\max_{i \in S} \|x_i\| = \|x_{k+1}\|$) or will be clipped (i.e. $\max_{i \in S} \|x_i\| > \|x_{k+1}\|$). Hence, we have that

$$\max_{i \in S} \|x_i\| \geq \|x_{k+1}\| \quad . \tag{19}$$

Combining (17) and (19), we have that for all $j \in [n]$,

$$\|\tilde{x}_j\| \leq \max_{i \in S} \|x_i\| \quad .$$

Using the fact that for any set of vectors $x_1, \ldots, x_n \in \mathbb{R}^d$, $\|\mathbf{F}(x_1, \ldots, x_n)\| \leq \max_{i \in [n]} \|x_i\|$, we have

$$\|\mathbf{F} \circ ARC(x_1, \ldots, x_n)\| = \|\mathbf{F}(\tilde{x}_1, \ldots, \tilde{x}_n)\| \leq \max_{j \in [n]} \|\tilde{x}_j\| \leq \max_{i \in S} \|x_i\| \quad . \tag{20}$$

$\square$

**Lemma C.1.** *For any aggregation rule* $\mathbf{F} \in \{\mathbf{CWTM}, \mathbf{CWMed}, \mathbf{GM}, \mathbf{MK}, \mathbf{CWTM} \circ \mathbf{NNM},$ $\mathbf{CWMed} \circ \mathbf{NNM}, \mathbf{GM} \circ \mathbf{NNM}, \mathbf{MK} \circ \mathbf{NNM}\}$, *there exists* $f \in [0, \frac{n}{2})$, *a set of vectors* $x_1, \ldots, x_n$ *and a subset* $S \subset [n]$, $|S| = n - f$ *such that*

$$\mathbf{F}(x_1, \ldots, x_n) \nleq \max_{i \in S} \|x_i\| \tag{21}$$

*Proof.* Let us consider $(x_1, x_2, x_3) = ((0, 1), (1, 0), (1, 1))$, and $S = \{x_1, x_2\}$,

For $\mathbf{F} \in \{\mathbf{CWTM}, \mathbf{CWMed}, \mathbf{GM}\}$, we have $\|\mathbf{F}(x_1, x_2, x_3)\| = \|(1, 1)\| = \sqrt{2} > \max_{i \in S} \|x_i\| = \|x_1\| = 1$.

For $\mathbf{F} \in \{\mathbf{MK}, \mathbf{CWTM} \circ \mathbf{NNM}, \mathbf{CWMed} \circ \mathbf{NNM}, \mathbf{GM} \circ \mathbf{NNM}, \mathbf{MK} \circ \mathbf{NNM}\}$, we have $\|\mathbf{F}(x_1, x_2, x_3)\| = \|(0.5, 1)\| = \sqrt{1.25} > \max_{i \in S} \|x_i\| = \|x_1\| = 1$. $\square$

**Lemma C.2.** *Let* $\mathbf{F}$ *be* $(f, \kappa)$-*robust. Let* $\Delta_o$ *and* $\rho$ *be real values such that* $\mathcal{L}_{\mathcal{H}}(\theta_1) - \mathcal{L}_{\mathcal{H}}^* \leq \Delta_o$ *and* $\max_{i \in \mathcal{H}} \|\nabla \mathcal{L}_i(\theta_1)\| \leq \exp\left(-\frac{\Delta_o}{\kappa G^2} L\right) \rho$. *If* $\gamma = \min\left\{\left(\frac{\Delta_o}{\kappa G^2}\right) \frac{1}{T}, \frac{1}{L}\right\}$, *then*

$$\frac{1}{T} \sum_{t=1}^{T} \|\nabla \mathcal{L}_{\mathcal{H}}(\theta_t)\|^2 \leq \frac{2 \Delta_o L}{T} + 5\kappa \left(G^2 + B^2 \rho^2\right).$$

Our proof for Lemma C.2 relies on the following sub-result.

**Lemma C.3.** *For all* $t \in [T]$, *we obtain that*

$$\max_{i \in \mathcal{H}} \|\nabla \mathcal{L}_i(\theta_t)\| \leq \exp\left(\gamma L T\right) \max_{i \in \mathcal{H}} \|\nabla \mathcal{L}_i(\theta_1)\|.$$

*Proof.* Consider an arbitrary $t \in [T]$. Let $(x_1, \ldots, x_n) := \mathbf{ARC}\left(g_t^{(1)}, \ldots, g_t^{(n)}\right)$. As $n > 2f$ and $|\mathcal{H}| = n - f$, the clipping threshold $C$ used in ARC (see Algorithm 2) is bounded by $\max_{i \in \mathcal{H}} \left\| g_t^{(i)} \right\|$ (see (19) in Lemma 5.1). Therefore,

$$\max_{i \in [n]} \|x_i\| \leq \max_{i \in \mathcal{H}} \left\| g_t^{(i)} \right\| = \max_{i \in \mathcal{H}} \|\nabla \mathcal{L}_i(\theta_t)\|. \tag{22}$$

Since for any set of $n$ vectors $z_1, \ldots, z_n \in \mathbb{R}^d$, $\|\mathbf{F}(z_1, \ldots, z_n)\| \leq \max_{i \in [n]} \|z_i\|$, from (22) we obtain that

$$\|R_t\| = \left\| \mathbf{F} \circ \mathbf{ARC}\left(g_t^{(1)}, \ldots, g_t^{(n)}\right) \right\| \leq \max_{i \in [n]} \|x_i\| \leq \max_{i \in \mathcal{H}} \|\nabla \mathcal{L}_i(\theta_t)\|.$$

Recall that $\theta_{t+1} := \theta_t - \gamma R_t$. Thus, from above we obtain that

$$\left\| \theta_{t+1} - \theta_t \right\| \leq \gamma \max_{i \in \mathcal{H}} \|\nabla \mathcal{L}_i(\theta_t)\|. \tag{23}$$

Due to $L$-Lipschitz smoothness of $\mathcal{L}_i(\theta)$ for all $i \in \mathcal{H}$, the above implies that

$$\left\| \nabla \mathcal{L}_i(\theta_{t+1}) - \nabla \mathcal{L}_i(\theta_t) \right\| \leq \gamma L \max_{i \in \mathcal{H}} \|\nabla \mathcal{L}_i(\theta_t)\|, \quad \forall i \in \mathcal{H}.$$

This, in conjunction with the triangle inequality, implies that

$$\left\| \nabla \mathcal{L}_i(\theta_{t+1}) \right\| \leq \|\nabla \mathcal{L}_i(\theta_t)\| + \gamma L \max_{i \in \mathcal{H}} \|\nabla \mathcal{L}_i(\theta_t)\|, \quad \forall i \in \mathcal{H}.$$

Therefore,

$$\max_{i \in \mathcal{H}} \left\| \nabla \mathcal{L}_i(\theta_{t+1}) \right\| \leq \max_{i \in \mathcal{H}} \|\nabla \mathcal{L}_i(\theta_t)\| + \gamma L \max_{i \in \mathcal{H}} \|\nabla \mathcal{L}_i(\theta_t)\| = (1 + \gamma L) \max_{i \in \mathcal{H}} \|\nabla \mathcal{L}_i(\theta_t)\|.$$

As $t$ was chosen arbitrarily from $[T]$, the above holds true for all $t \in [T]$. For an arbitrary $\tau \in [T]$, using the inequality recursively for $t = \tau, \ldots, 1$ we obtain that

$$\max_{i \in \mathcal{H}} \left\| \nabla \mathcal{L}_i(\theta_{\tau+1}) \right\| \leq (1 + \gamma L)^\tau \max_{i \in \mathcal{H}} \|\nabla \mathcal{L}_i(\theta_1)\| \leq (1 + \gamma L)^T \max_{i \in \mathcal{H}} \|\nabla \mathcal{L}_i(\theta_1)\|.$$

Since $(1 + \gamma L)^T \leq \exp(\gamma L T)$, the above concludes the proof. $\qquad \square$

We are now ready to present our **proof of Lemma C.2**.

*Proof of Lemma C.2.* For simplicity, we write $\mathcal{L}_{\mathcal{H}}$ as $\mathcal{L}$ throughout the proof.

Consider an arbitrary $t \in [T]$. Due to $L$-Lipschitz smoothness of $\mathcal{L}(\theta)$, we obtain that

$$\mathcal{L}(\theta_{t+1}) \leq \mathcal{L}(\theta_t) + \langle \theta_{t+1} - \theta_t, \nabla \mathcal{L}(\theta_t) \rangle + \frac{L}{2} \left\| \theta_{t+1} - \theta_t \right\|^2.$$

Substituting $\theta_{t+1} = \theta_t - \gamma R_t$, and using the identity: $2 \langle a, b \rangle = \|a\|^2 + \|b\|^2 - \|a - b\|^2$, we obtain that

$$\mathcal{L}(\theta_{t+1}) \leq \mathcal{L}(\theta_t) - \frac{\gamma}{2} \|\nabla \mathcal{L}(\theta_t)\|^2 + \frac{\gamma}{2} \|R_t - \nabla \mathcal{L}(\theta_t)\|^2 - \frac{\gamma}{2} (1 - \gamma L) \|R_t\|^2.$$

Since $\gamma \leq \frac{1}{L}$, $(1 - \gamma L) \geq 0$. Therefore, from above we obtain that

$$\mathcal{L}(\theta_{t+1}) \leq \mathcal{L}(\theta_t) - \frac{\gamma}{2} \|\nabla \mathcal{L}(\theta_t)\|^2 + \frac{\gamma}{2} \|R_t - \nabla \mathcal{L}(\theta_t)\|^2.$$

Recall that $t$ is chosen arbitrarily from $[T]$. Thus, the above holds true for all $t \in [T]$. Taking summation on both sides from $t = 1$ to $t = T$ we obtain that

$$\frac{\gamma}{2} \sum_{t=1}^{T} \|\nabla \mathcal{L}(\theta_t)\|^2 \leq \mathcal{L}(\theta_t) - \mathcal{L}(\theta_{T+1}) + \frac{\gamma}{2} \sum_{t=1}^{T} \|R_t - \nabla \mathcal{L}(\theta_t)\|^2.$$

Multiplying both sides by $2/(\gamma T)$ we obtain that

$$\frac{1}{T} \sum_{t=1}^{T} \|\nabla \mathcal{L}(\theta_t)\|^2 \leq \frac{2 \left(\mathcal{L}(\theta_1) - \mathcal{L}(\theta_{T+1})\right)}{\gamma T} + \frac{1}{T} \sum_{t=1}^{T} \|R_t - \nabla \mathcal{L}(\theta_t)\|^2 .$$

Note that $\mathcal{L}(\theta_1) - \mathcal{L}(\theta_{T+1}) = \mathcal{L}(\theta_1) - \mathcal{L}^* - \left(\mathcal{L}(\theta_{T+1}) - \mathcal{L}^*\right) \leq \mathcal{L}(\theta_1) - \mathcal{L}^* \leq \Delta_o$. Using this above we obtain that

$$\frac{1}{T} \sum_{t=1}^{T} \|\nabla \mathcal{L}(\theta_t)\|^2 \leq \frac{2\Delta_o}{\gamma T} + \frac{1}{T} \sum_{t=1}^{T} \|R_t - \nabla \mathcal{L}(\theta_t)\|^2 .$$

Recall, by Theorem 3.2, that $\mathbf{F} \circ \mathbf{ARC}$ is $(f, 3\kappa)$-robust. Therefore, by Definition 2.4, $\|R_t - \nabla \mathcal{L}(\theta_t)\|^2 \leq \frac{3\kappa}{|\mathcal{H}|} \sum_{i \in \mathcal{H}} \|\nabla \mathcal{L}_i(\theta_t) - \nabla \mathcal{L}(\theta_t)\|^2$ for all $t$. Using this above we obtain that

$$\frac{1}{T} \sum_{t=1}^{T} \|\nabla \mathcal{L}(\theta_t)\|^2 \leq \frac{2\Delta_o}{\gamma T} + \frac{3\kappa}{T} \sum_{t=1}^{T} \frac{1}{|\mathcal{H}|} \|\nabla \mathcal{L}_i(\theta_t) - \nabla \mathcal{L}(\theta_t)\|^2 . \tag{24}$$

Note that, since $\gamma \leq \frac{\Delta_o}{\kappa G^2 T}$ and $\|\nabla \mathcal{L}_i(\theta_1)\| \leq \exp\left(-\frac{\Delta_o L}{\kappa G^2}\right) \rho$, by Lemma C.3 we have

$$\max_{i \in \mathcal{H}} \|\nabla \mathcal{L}_i(\theta_t)\| \leq \exp\left(\frac{\Delta_o L}{\kappa G^2}\right) \max_{i \in \mathcal{H}} \|\nabla \mathcal{L}_i(\theta_1)\| \leq \rho, \quad \forall t \in [T].$$

This, in conjunction with triangle inequality, implies that

$$\|\nabla \mathcal{L}(\theta_t)\| \leq \frac{1}{|\mathcal{H}|} \sum_{i \in \mathcal{H}} \|\nabla \mathcal{L}_i(\theta_t)\| \leq \rho, \quad \forall t \in [T].$$

Therefore, under $(G, B)$-gradient dissimilarity, for all $t \in [T]$,

$$\frac{1}{|\mathcal{H}|} \|\nabla \mathcal{L}_i(\theta_t) - \nabla \mathcal{L}(\theta_t)\|^2 \leq G^2 + B^2 \, \rho^2.$$

Using this in (24) we obtain that

$$\frac{1}{T} \sum_{t=1}^{T} \|\nabla \mathcal{L}(\theta_t)\|^2 \leq \frac{2\Delta_o}{\gamma T} + 3\kappa \left(G^2 + B^2 \rho^2\right). \tag{25}$$

Consider the two cases: (i) $T \geq \frac{\Delta_o L}{\kappa G^2}$ and (ii) $T < \frac{\Delta_o L}{\kappa G^2}$. In case (i), $\gamma = \frac{\Delta_o}{\kappa G^2 T}$. Using this in (25) implies that

$$\frac{1}{T} \sum_{t=1}^{T} \|\nabla \mathcal{L}(\theta_t)\|^2 \leq 2\kappa G^2 + 3\kappa \left(G^2 + B^2 \rho^2\right) \leq 5\kappa \left(G^2 + B^2 \rho^2\right). \tag{26}$$

In case (ii), $\gamma = \frac{1}{L}$. Using this in (25) implies that

$$\frac{1}{T} \sum_{t=1}^{T} \|\nabla \mathcal{L}(\theta_t)\|^2 \leq \frac{2\Delta_o L}{T} + 3\kappa \left(G^2 + B^2 \rho^2\right). \tag{27}$$

Combining (26) and (27) concludes the proof. $\qquad \square$

We now prove Theorem C.4, stated below, which immediately implies Theorem 5.2. Specifically, **Theorem 5.2 follows immediately from Part 1 of Theorem C.4, upon substituting $\xi \leq \frac{1}{\Psi(G,B,\rho)} \upsilon$.**

**Theorem C.4.** *Suppose $B > 0$ and there exists $\zeta \in \mathbb{R}^+$ such that $\max_{i \in \mathcal{H}} \|\nabla \mathcal{L}_i(\theta_1)\| \leq \zeta$. Let $\mathbf{F} \in \{\mathbf{CWTM}, \mathbf{CWMed}, \mathbf{GM}, \mathbf{MK}\}$, $\gamma = \min\left\{\left(\frac{\Delta_o}{\kappa G^2}\right) \frac{1}{T}, \frac{1}{L}\right\}$ and $T \geq \frac{\Delta_o L}{\kappa G^2}$. Consider an arbitrary real value $\xi_o \in (0, 1)$. Let $\rho := \exp\left(\frac{(2+B^2)\Delta_o}{(1-\xi_o)G^2} L\right) \zeta$. Then, the following holds true:*

> *1. If $\frac{f}{n} := (1 - \xi)\mathsf{BP}$, where $\xi \in (0, \xi_o]$, then, $\mathbb{E}\left[\left\|\nabla \mathcal{L}_{\mathcal{H}}\left(\hat{\theta}\right)\right\|^2\right] \leq \xi \, \Psi(G, B, \rho) \, \varepsilon_o.$*

2. *If* $\mathsf{BP} \leq f/n < 1/2$, *then* $\mathbb{E}\left[\left\|\nabla\mathcal{L}_{\mathcal{H}}\left(\hat{\theta}\right)\right\|^2\right] \leq \Psi(G, B, \rho) \cdot \frac{G^2}{8}$.

In order to prove Theorem C.4, we first obtain the following corollary of Lemma C.2 and Lemma 2.6.

**Corollary C.5.** *For the parameters given in Lemma C.2, if* $T \geq \frac{\Delta_o L}{\kappa G^2}$, *then Robust-DGD* $\circ$ *ARC achieves*

$$\mathbb{E}\left[\left\|\nabla\mathcal{L}_{\mathcal{H}}\left(\hat{\theta}\right)\right\|^2\right] \leq 5\kappa \, \min\left\{G^2 + B^2\rho^2, \, \frac{G^2}{\max\{(1 - \kappa B^2), 0\}}\right\},$$

*where* $\mathbb{E}\left[\cdot\right]$ *denotes the expectation over the choice of* $\hat{\theta}$.

*Proof of Corollary C.5.* For simplicity, we write $\mathcal{L}_{\mathcal{H}}$ as $\mathcal{L}$ throughout the proof.

Since $T \geq \frac{\Delta_o L}{\kappa G^2}$, $\gamma = \frac{\Delta_o}{\kappa G^2 T} \leq \frac{1}{L}$. Therefore, from Lemma C.2 we obtain that

$$\frac{1}{T}\sum_{t=1}^{T}\|\nabla\mathcal{L}(\theta_t)\|^2 \leq 5\kappa\left(G^2 + B^2\rho^2\right). \tag{28}$$

In the specific case when $\kappa B^2 < 1$, as $\gamma = \frac{\Delta_o}{\kappa G^2 T}$, from Lemma 2.6 we obtain that

$$\frac{1}{T}\sum_{t=1}^{T}\|\nabla\mathcal{L}(\theta_t)\|^2 \leq \frac{2\Delta_o}{(1 - \kappa B^2)\gamma T} + \frac{\kappa G^2}{1 - \kappa B^2} = \frac{3\kappa G^2}{1 - \kappa B^2}. \tag{29}$$

Combining (28) and (29) we obtain that

$$\frac{1}{T}\sum_{t=1}^{T}\|\nabla\mathcal{L}(\theta_t)\|^2 \leq 5\kappa \min\left\{G^2 + B^2\rho^2, \, \frac{G^2}{1 - \kappa B^2}\right\}.$$

Since $\mathbb{E}\left[\|\nabla\mathcal{L}(\theta_t)\|^2\right] = \frac{1}{T}\sum_{t=1}^{T}\|\nabla\mathcal{L}(\theta_t)\|^2$ (see Algorithm 1), the above concludes the proof. $\quad\square$

We are now ready to prove Theorem C.4. Recall that

$$\mathsf{BP} := \frac{1}{2 + B^2}, \quad \varepsilon_o := \frac{1}{4} \cdot \frac{G^2(f/n)}{1 - (2 + B^2)(f/n)}, \quad \text{and}$$

$$\Psi(G, B, \rho) := 640\left(1 + \frac{1}{B^2}\right)^2\left(1 + \frac{B^2\rho^2}{G^2}\right).$$

*Proof of Theorem C.4.* For simplicity, we write $\mathcal{L}_{\mathcal{H}}$ as $\mathcal{L}$ throughout the proof.

In the proof, we substitute $\mathsf{BP} = \frac{1}{2 + B^2}$.

We first consider the case when $\frac{f}{n} = \frac{1 - \xi}{2 + B^2}$. In this particular case,

$$n = \left(\frac{2}{1 - \xi} + \frac{B^2}{1 - \xi}\right)f \geq \left(2 + \frac{B^2}{1 - \xi}\right)f.$$

Therefore, thanks to Lemma 2.5, $\mathbf{F}$ is $(f, \kappa)$-robust with

$$\kappa \leq \frac{16f}{n - f}\left(1 + \frac{1 - \xi}{B^2}\right)^2 = \frac{16(1 - \xi)}{1 + B^2 + \xi}\left(1 + \frac{1 - \xi}{B^2}\right)^2 \leq \frac{16(1 - \xi)}{1 + B^2}\left(1 + \frac{1}{B^2}\right)^2 \leq \kappa_o(1 - \xi),$$

where $\kappa_o := \frac{16}{1 + B^2}\left(1 + \frac{1}{B^2}\right)^2$. Also, recall from the limitations of $(f, \kappa)$-robustness in Section 2 that

$$\kappa \geq \frac{f}{n - 2f} \geq \frac{f}{n} \geq \frac{1 - \xi_o}{2 + B^2}.$$

Summarizing from above, we have

$$\frac{1 - \xi_o}{2 + B^2} \leq \kappa \leq \kappa_o (1 - \xi). \tag{30}$$

This implies that

$$\rho := \exp\left(\frac{(2 + B^2)\Delta_o}{(1 - \xi_o)G^2} L\right) \zeta \geq \exp\left(\frac{\Delta_o}{\kappa G^2} L\right) \zeta.$$

Therefore, since $\|\nabla \mathcal{L}_i(\theta_1)\| \leq \zeta$ for all $i \in \mathcal{H}$, the condition: $\max_{i \in \mathcal{H}} \|\nabla \mathcal{L}_i(\theta_1)\| \leq \exp\left(-\frac{\Delta_o}{\kappa G^2} L\right) \rho$ in Lemma C.2 is satisfied. Thus, by Corollary C.5 and (30) we obtain that

$$\mathbb{E}\left[\|\nabla \mathcal{L}(\theta_t)\|^2\right] \leq 5\kappa_o (1 - \xi)(G^2 + B^2 \rho^2) = \frac{5\kappa_o (1 - \xi)(G^2 + B^2 \rho^2)}{\frac{fG^2}{n - (2 + B^2)f}} \frac{fG^2}{n - (2 + B^2)f}. \tag{31}$$

Since $\frac{f}{n} = \frac{1 - \xi}{2 + B^2}$, we obtain that

$$\frac{5\kappa_o (1 - \xi)(G^2 + B^2 \rho^2)}{\frac{fG^2}{n - (2 + B^2)f}} = 5\kappa_o (1 - \xi)(G^2 + B^2 \rho^2) \frac{\xi(2 + B^2)}{G^2(1 - \xi)} = 5\kappa_o \left(1 + \frac{B^2 \rho^2}{G^2}\right)(2 + B^2)\xi.$$

Substituting $\kappa_o$ from above, we obtain that

$$\frac{5\kappa_o (1 - \xi)(G^2 + B^2 \rho^2)}{\frac{fG^2}{n - (2 + B^2)f}} = 80 \left(\frac{2 + B^2}{1 + B^2}\right)\left(1 + \frac{1}{B^2}\right)^2 \left(1 + \frac{B^2 \rho^2}{G^2}\right)\xi$$

$$\leq 160 \left(1 + \frac{1}{B^2}\right)^2 \left(1 + \frac{B^2 \rho^2}{G^2}\right)\xi.$$

Substituting from above in (31) concludes the proof for the first part of Theorem C.4.

The second part of Theorem C.4 follows immediately from the first inequality in (31), using the fact that $\kappa_o(1 - \xi) \leq \kappa_o$. This concludes the proof. $\qquad \square$

## C.1 ASSUMING $\|\mathbf{F}(x_1, \ldots, x_n)\| \leq \max_{i \in [n]} \|x_i\|$ IS WITHOUT LOSS OF GENERALITY

Recall that we assume $\|\mathbf{F}(x_1, \ldots, x_n)\| \leq \max_{i \in [n]} \|x_i\|$ for all set of $n$ vectors $x_1, \ldots, x_n \in \mathbb{R}^d$. In case this is not true, we can instead use the aggregation rule $\mathbf{F}^\dagger$ given by

$$\mathbf{F}^\dagger(x_1, \ldots, x_n) := \mathrm{clip}_C(\mathbf{F}(x_1, \ldots, x_n)), \text{ where } C = \max_{i \in [n]} \|x_i\|.$$

This modification to the aggregation rule does not affect the learning guarantee. Specifically, due to the non-expansion property of $\mathrm{clip}_C(\cdot)$, for any non-empty set $S \subseteq [n]$, we have

$$\left\|\mathbf{F}^\dagger(x_1, \ldots, x_n) - \overline{x}_S\right\| \leq \left\|\mathbf{F}(x_1, \ldots, x_n) - \overline{x}_S\right\|,$$

where $\overline{x}_S := \frac{1}{|S|} \sum_{i \in S} x_i$. Thus, if $\mathbf{F}$ is $(f, \kappa)$-robust, then $\mathbf{F}^\dagger$ is also $(f, \kappa)$-robust. Hence, we can make the above assumption on $\|\mathbf{F}(x_1, \ldots, x_n)\|$ without loss of generality.

## D COMPREHENSIVE EXPERIMENTAL SETUP

In this section, we present the comprehensive experimental setup considered in our paper.

### D.1 DATASETS AND HETEROGENEITY

In our experiments, we consider three standard image classification datasets, namely MNIST (Deng, 2012), Fashion-MNIST (Xiao et al., 2017), and CIFAR-10 (Krizhevsky et al., 2014). To simulate data heterogeneity in our experiments, we make the honest workers sample from the datasets using a Dirichlet distribution of parameter $\alpha$ (Hsu et al., 2019a), as done in (Hsu et al., 2019b; Allouah et al., 2023a; Farhadkhani et al., 2023). The smaller the $\alpha$, the more heterogeneous the setting. In our empirical evaluation, we set $\alpha \in \{0.1, 0.5, 1\}$ on (Fashion-)MNIST and $\alpha \in \{0.05, 0.075, 0.1, 0.2, 0.5\}$ on CIFAR-10 (refer to Figure 6). Furthermore, on MNIST and Fashion-MNIST, we also consider an *extreme* heterogeneity setting where the datapoints are sorted by increasing labels (0 to 9) and sequentially split equally among the honest workers.

The input images of MNIST are normalized with mean $0.1307$ and standard deviation $0.3081$, while the images of Fashion-MNIST are horizontally flipped. Moreover, CIFAR-10 is expanded with horizontally flipped images, followed by a per channel normalization with means $0.4914, 0.4822, 0.4465$ and standard deviations $0.2023, 0.1994, 0.2010$.

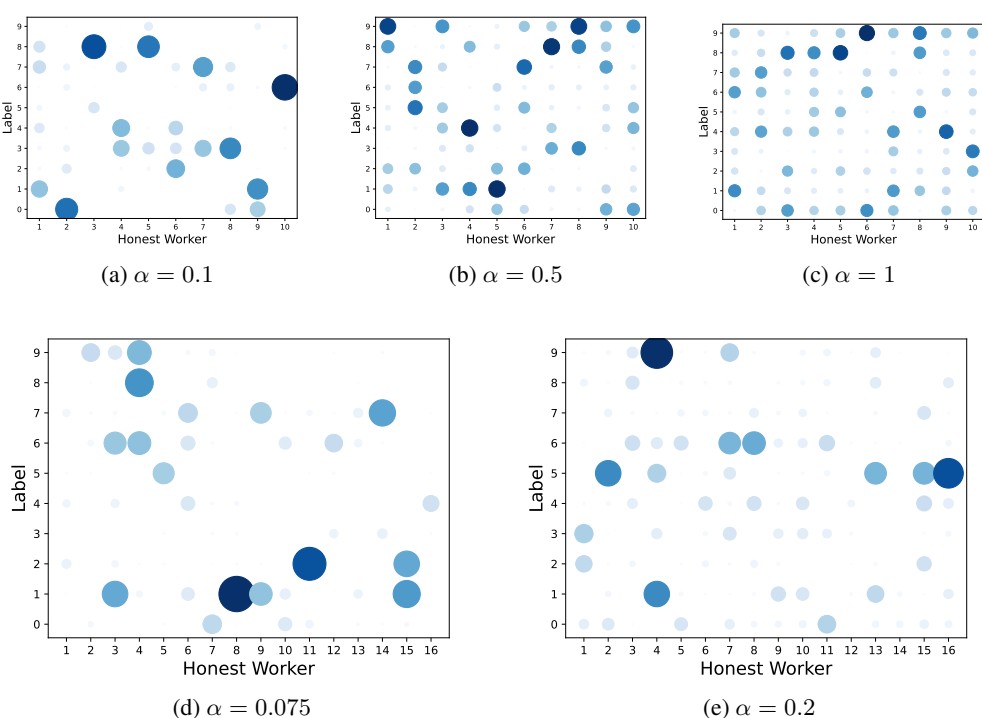

Figure 6: Distribution of labels across honest workers on MNIST (row 1) and CIFAR-10 (row 2).

### D.2 ALGORITHM, DISTRIBUTED SYSTEM, ML MODELS, AND HYPERPARAMETERS

We perform our experiments using Robust-DSGD, an order-optimal robust variant of Robust-DGD (Allouah et al., 2023a) (see Algorithm 3), in distributed systems of varying sizes. On MNIST and Fashion-MNIST, we train a convolutional neural network (CNN) of 431,080 parameters with batch size $b = 25$, $T = 1000$, $\gamma = 0.1$, and momentum parameter $\beta = 0.9$. Moreover, the negative log likelihood (NLL) loss function is used, along with an $\ell_2$-regularization of $10^{-4}$. On CIFAR-10, we train a CNN of 1,310,922 parameters. We set $b = 50$, $T = 2000$, $\beta = 0.9$, and $\gamma = 0.05$ decaying once at step 1500. Finally, we use the NLL loss function with an $\ell_2$

regularization of $10^{-2}$. In order to present the architectures of the ML models used in our experiments, we adopt the following compact terminology introduced in Choffrut et al. (2024).

L(#outputs) represents a **fully-connected linear layer**, R stands for **ReLU activation**, S stands for **log-softmax**, C(#channels) represents a *fully-connected 2D-convolutional layer* (kernel size 5, padding 0, stride 1), M stands for **2D-maxpool** (kernel size 2), B stands for **batch-normalization**, and D represents **dropout** (with fixed probability 0.25).

The comprehensive experimental setup and the architecture of the models are presented in Table 2.

---

**Algorithm 3** Robust Distributed Stochastic Gradient Descent (Robust-DSGD)

---

**Input: Server** chooses an initial model $\theta_1 \in \mathbb{R}^d$, *learning rate* $\gamma \in \mathbb{R}^+$, robust aggregation $\mathbf{F} : \mathbb{R}^{n \times d} \to \mathbb{R}^d$, *momentum parameter* $\beta \in \mathbb{R}^+$. Each **honest worker** $w_i$ sets an initial *momentum* $m_0^{(i)} = 0 \in \mathbb{R}^d$.

**for** $t = 1$ to $T$ **do**

    **Server** broadcasts $\theta_t$ to all workers.

    **for each honest worker** $w_i$ **in parallel do**

        Sample a random data point $z^{(i)}$ uniformly from $\mathcal{D}_i$.

        Compute stochastic gradient $g_t^{(i)} := \nabla \ell(\theta_t, z^{(i)})$.

        Update momentum: $m_t^{(i)} := (1 - \beta) g_t^{(i)} + \beta m_{t-1}^{(i)}$.

        Send $m_t^{(i)}$ to **Server**.

        *// An adversarial worker $w_j$ can send an arbitrary vector in $\mathbb{R}^d$ for $m_t^{(j)}$*

    **end for**

    **Server** computes $R_t := \mathbf{F}\left( m_t^{(1)}, \ldots, m_t^{(n)} \right)$.

    **Server** updates the model: $\theta_{t+1} := \theta_t - \gamma R_t$.

**end for**

**Output: Server** outputs $\hat{\theta}$ chosen uniformly at random from $\{\theta_1, \ldots, \theta_T\}$.

---

| *Dataset* | (Fashion-)MNIST | CIFAR-10 |
|---|---|---|
| *Data heterogeneity* | $\alpha \in \{0.1, 0.5, 1\}$ and extreme | $\alpha \in \{0.05, 0.075, 0.1, 0.2, 0.5\}$ |
| *Model type* | CNN | CNN |
| *Model architecture* | C(20)-R-M-C(20)-R-M-L(500)-R-L(10)-S | (3,32×32)-C(64)-R-B-C(64)-R-B-M-D-C(128)-R-B-C(128)-R-B-M-D-L(128)-R-D-L(10)-S |
| *Number of parameters* | 431,080 | 1,310,922 |
| *Loss* | NLL | NLL |
| $\ell_2$*-regularization* | $10^{-4}$ | $10^{-2}$ |
| *Number of steps* | $T = 1000$ | $T = 2000$ |
| *Learning rate* | $\gamma = 0.1$ | $\gamma_t = \begin{cases} 0.05 & t \leq 1500 \\ 0.00082 & 1500 < t \leq 2000 \end{cases}$ |
| *Momentum parameter* | $\beta = 0.9$ | $\beta = 0.9$ |
| *Batch size* | $b = 25$ | $b = 50$ |

Table 2: Comprehensive experimental setup.

### D.3 BYZANTINE ATTACKS

In our experiments, the adversarial workers execute five state-of-the-art adversarial attacks from the Byzantine ML literature, namely *sign-flipping* (SF) (Allen-Zhu et al., 2020), *label-flipping* (LF) (Allen-Zhu et al., 2020), *mimic* (Karimireddy et al., 2022), *fall of empires* (FOE) (Xie et al., 2020), and *a little is enough* (ALIE) (Baruch et al., 2019). The exact functionality of the attacks is detailed next. In every step $t$, let $\overline{m}_t$ be an estimation of the true honest momentum at step $t$. In our experiments, we estimate $\overline{m}_t$ by averaging the momentums sent by the honest workers in step $t$ of Robust-DSGD. In other words, $\overline{m}_t = \frac{1}{|\mathcal{H}|} \sum_{i \in \mathcal{H}} m_t^{(i)}$, where $m_t^{(i)}$ is the momentum computed by honest worker $w_i$ in step $t$.

- SF: the adversarial workers send the vector $-\overline{m}_t$ to the server.
- LF: the adversarial workers compute their gradients on flipped labels, and send the flipped gradients to the server. Since the original labels $l$ for (Fashion-)MNIST and CIFAR-10 are in $\{0, ..., 9\}$, the adversarial workers execute a label flip/rotation by computing their gradients on the modified labels $l' = 9 - l$.
- Mimic: the adversarial workers *mimic* a certain honest worker by sending its gradient to the server. In order to determine the optimal honest worker for the adversarial workers to mimic, we use the heuristic in Karimireddy et al. (2022).
- FOE: the adversarial workers send $(1 - \tau)\overline{m}_t$ in step $t$ to the server, where $\tau \geq 0$ is a fixed real number representing the attack factor. When $\tau = 2$, this attack is equivalent to SF.
- ALIE: the adversarial workers send $\overline{m}_t + \tau \sigma_t$ in step $t$ to the server, where $\tau \geq 0$ is a fixed real number representing the attack factor, and $\sigma_t$ is the coordinate-wise standard deviation of $\overline{m}_t$.

Since FOE and ALIE are parametrized by $\tau \in \mathbb{R}$, we implement in our experiments enhanced and adaptive versions of these attacks, where the attack factor is not constant and may be different in every iteration. In every step $t$, we determine the optimal attack factor $\tau_t$ through a grid search over a predefined range of values. More specifically, in every step $t$, $\tau_t$ takes the value that maximizes the damage inflicted by the adversarial workers, i.e., that maximizes the $\ell_2$-norm of the difference between the average of the honest momentums $\overline{m}_t$ and the output of the aggregation $R_t$ at the server.

### D.4 BENCHMARKING AND REPRODUCIBILITY

We evaluate the performance of ARC compared to no clipping within the context of Robust-DSGD. Accordingly, we choose $\mathbf{F} \circ$ NNM as aggregator in Algorithm 3, where $\mathbf{F} \in \{$CWTM, GM, CWMed, MK$\}$ is an aggregation method proved to be $(f, \kappa)$-robust (Allouah et al., 2023a). Pre-composing these aggregation schemes with NNM provides them with optimal $(f, \kappa)$-robustness (Allouah et al., 2023a). As benchmark, we also execute the standard DSGD algorithm in the same setting, but in the absence of adversarial workers (i.e., without attack and $f = 0$). We use the metric of ***worst-case maximal accuracy*** to evaluate the performance of our algorithm. In other words, for each of the aforementioned five Byzantine attacks, we record the maximal accuracy achieved by Robust-DSGD during the learning under that attack. The worst-case maximal accuracy is thus the *smallest* maximal accuracy encountered across the five attacks. As the attack executed by adversarial workers cannot be known in advance in a practical system, this metric is critical to accurately evaluate the robustness of aggregation methods, as it gives us an estimate of the potential worst-case performance of the algorithm. Finally, all our experiments are run with seeds 1 to 5 for reproducibility purposes. We provide standard deviation measurements for all our results (across the five seeds).

# E ARC vs. Static Clipping

In this section, we present empirical results on static clipping when used as a pre-aggregation technique in Robust-DSGD, and compare its performance to our proposed adaptive algorithm ARC.

## E.1 Comparison with Static Clipping on MNIST

On MNIST, we execute Robust-DSGD in a distributed system composed of $n = 15$ workers, among which $f \in \{3, 4\}$ are adversarial. We consider three different levels of heterogeneity: *moderate* ($\alpha = 1$), *high* ($\alpha = 0, 1$), and *extreme* (as explained in Appendix D). In our experiments, we examine a wide range of static clipping parameters $C \in \{0.02, 0.2, 2, 20\}$.

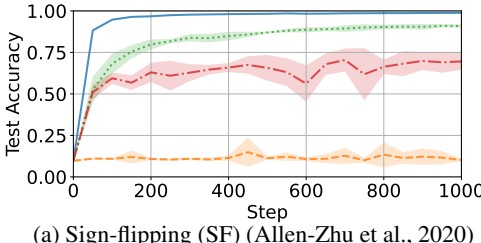 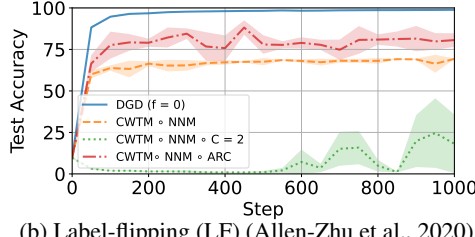

(a) Sign-flipping (SF) (Allen-Zhu et al., 2020)    (b) Label-flipping (LF) (Allen-Zhu et al., 2020)

Figure 7: Impact of pre-aggregation clipping, specifically static clipping with $C = 2$ and our adaptive clipping algorithm ARC, on Robust-DGD with CWTM ∘ NNM, under the SF and LF attacks. We consider the MNIST (Deng, 2012) dataset distributed amongst 10 honest (non-adversarial) workers with *extreme* heterogeneity, and there are three adversarial workers.

**Unreliability due to the attack.** First, the efficacy of static clipping is significantly influenced by the type of Byzantine attack executed during the learning process. To illustrate, while $C = 2$ exhibits the best performance under the SF attack (Allen-Zhu et al., 2020), depicted in Figure 7, it leads to a complete collapse of learning under LF (Allen-Zhu et al., 2020). Consequently, identifying a static clipping threshold $C$ that consistently performs well in practice against any attack is difficult (if not impossible). This highlights the fragility of static clipping, as the unpredictable nature of the Byzantine attack, a parameter that cannot be a priori known, can significantly degrade the performance of the chosen static clipping approach.

**Unreliability due to the heterogeneity model.** Second, the robustness under static clipping is notably influenced by the level of heterogeneity present across the datasets of honest workers. To illustrate this impact, we present in Figure 2 the performances of Robust-DSGD for different levels of heterogeneity (moderate, high, and extreme) and different values of $C$. In Figure 2a, $C = 2$ emerges as the best static clipping threshold when the heterogeneity is high whereas $C = 20$ appears to be sub-optimal. On the other hand, under extreme heterogeneity, the accuracy associated to $C = 2$ diminishes drastically while $C = 20$ becomes a better choice. Intuitively, in heterogeneous scenarios, honest gradients become large in $\ell_2$-norm, due to the increasingly detrimental effect of Byzantine attacks. Consequently, we must increase the static clipping threshold. This highlights the intricate dependence of the performance of static clipping on data heterogeneity, and emphasizes the necessity to fine-tune static clipping strategies prior to the learning.

**Unreliability due to the number of adversarial workers.** Last but not least, the number of adversarial workers $f$ also affects the efficacy of static clipping. As seen in Figure 2, $C = 2$ leads to the highest accuracy among static clipping strategies when $f = 3$ in high heterogeneity, but cannot be used when $f = 4$ as its corresponding accuracy drops below 50% (see Figure 2b).

Overall, our empirical findings reveal that there is no single value of $C$ for which static clipping consistently delivers satisfactory performance across the various settings. We have also shown that the performance of static clipping is greatly influenced on typically uncontrollable parameters, such as data heterogeneity and the nature of the Byzantine attack. Indeed, it should be noted that explicitly estimating these parameters is challenging (if not impossible) in a distributed setting. First, since the server does not have direct access to the data, it cannot estimate the degree of heterogeneity. Second, as (by definition) an adversarial worker can behave in an unpredictable manner, we cannot adjust $C$

to the attack(s) being executed by the adversarial workers. This highlights the necessity for a robust clipping alternative that can naturally adapt to the setting in which it is deployed.

In contrast to static clipping, ARC is adaptive, delivering consistent performance across diverse levels of heterogeneity, Byzantine attacks, and number of adversarial workers.

**Adaptiveness.** ARC dynamically adjusts its clipping parameter $C_t$ based on the norms of honest momentums at step $t$, avoiding static over-clipping or under-clipping. This adaptability is evident in Figure 8, where $C_t$ consistently decreases with time under all attacks, an expected behavior when reaching convergence. Moreover, Figure 8 also illustrates that any surge in the norm of the honest mean corresponds to a direct increase in $C_t$, highlighting the adaptive nature of ARC.

**Robust performance across heterogeneity regimes.** The efficacy of our solution is illustrated in Figure 2, showcasing a consistently robust performance for all considered heterogeneity levels. Specifically, in scenarios of moderate heterogeneity where clipping may not be essential, ARC matches the performance of *No clipping* as well as static strategies $C = 2$ and $C = 20$. Conversely, Figure 2a shows that under high and extreme heterogeneity, ARC surpasses the best static clipping strategy in terms of accuracy, highlighting the effectiveness of our approach.

**Robust performance across Byzantine regimes.** ARC exhibits robust performance across diverse Byzantine scenarios, encompassing variations in both the type of Byzantine attack and the number of adversarial workers $f$. As depicted in Figures 7 and 2, ARC consistently yields robust performance, regardless of the Byzantine attack. In extreme heterogeneity with $f = 3$, ARC maintains an accuracy of 75%, while $C = 2$ demonstrates a subpar worst-case accuracy of 25% (also refer to Figure 7b). Furthermore, despite the increase in the number of adversarial workers to $f = 4$, ARC remains the top-performing clipping approach among all considered strategies.

This analysis underscores the empirical superiority of ARC over static clipping methods, eliminating the reliance on data heterogeneity and the specific Byzantine regime. Similar trends are also observed in Figure 9, when executing Robust-DSGD using other aggregation methods such as CWMed ∘ NNM, GM ∘ NNM, and Multi-Krum ∘ NNM.

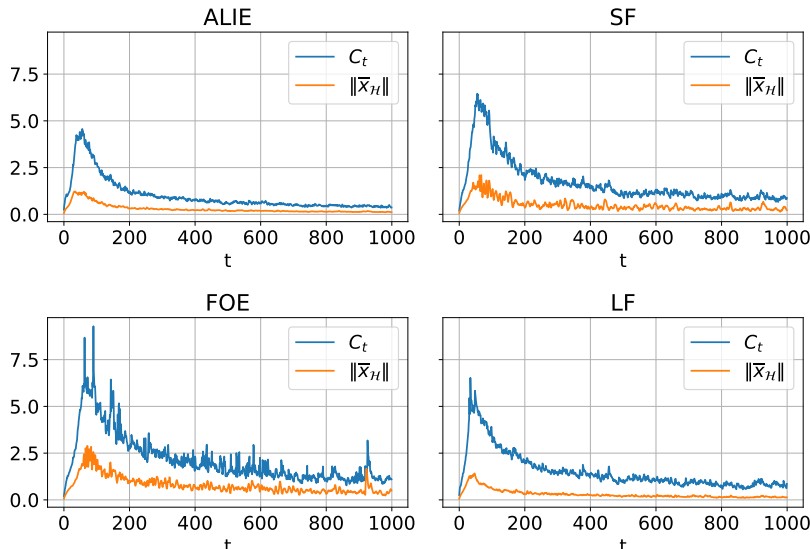

Figure 8: Evolution of the adaptive clipping parameter $C_t$ of ARC compared to the norm of the honest mean during the learning on heterogeneous MNIST ($\alpha = 1$). There are $f = 3$ adversarial workers among $n = 15$. CWTM ∘ NNM is used as aggregation.

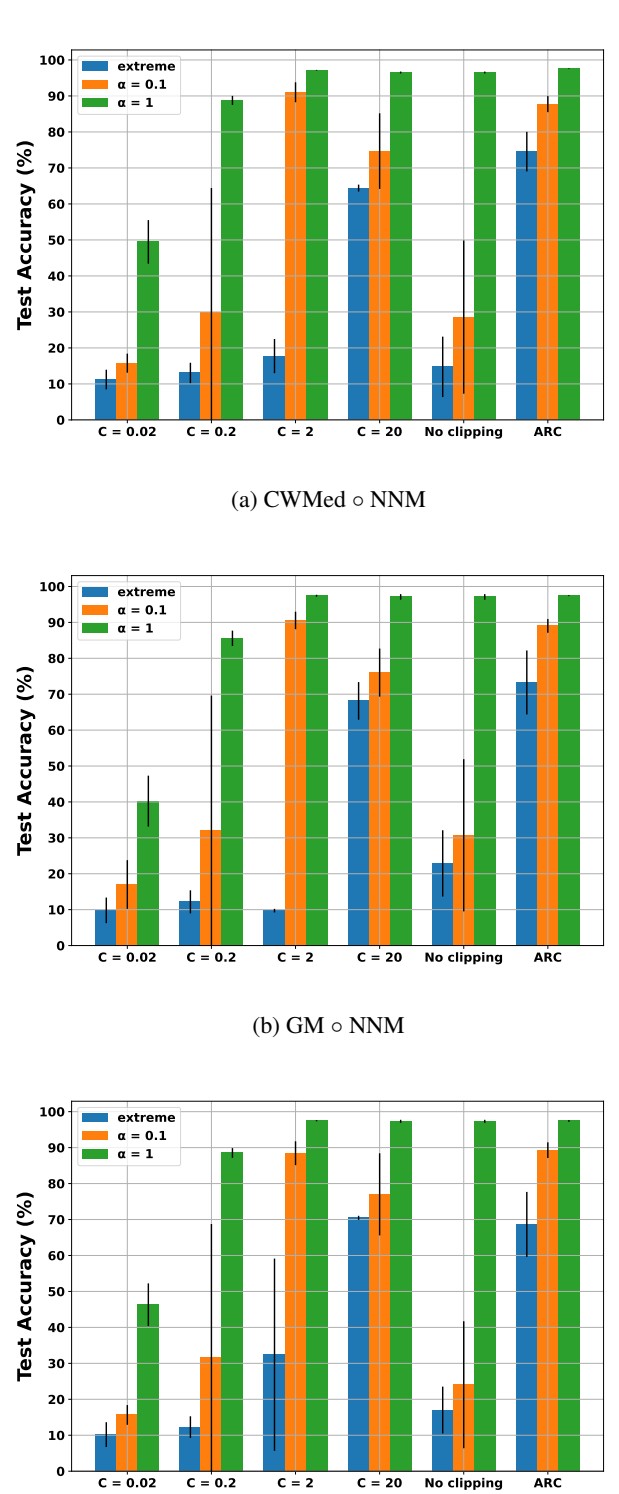

(a) CWMed ∘ NNM

(b) GM ∘ NNM

(c) MK ∘ NNM

Figure 9: Impact of the clipping strategy on the worst-case maximal accuracy achieved during the learning against five Byzantine attacks on heterogeneous MNIST, with $f = 3$ and $n = 15$.

### E.2 COMPARISON WITH STATIC CLIPPING ON FASHION-MNIST

On Fashion-MNIST, we execute Robust-DSGD in a distributed system composed of $n = 15$ workers, among which $f = 3$ are adversarial. See Figure 10.

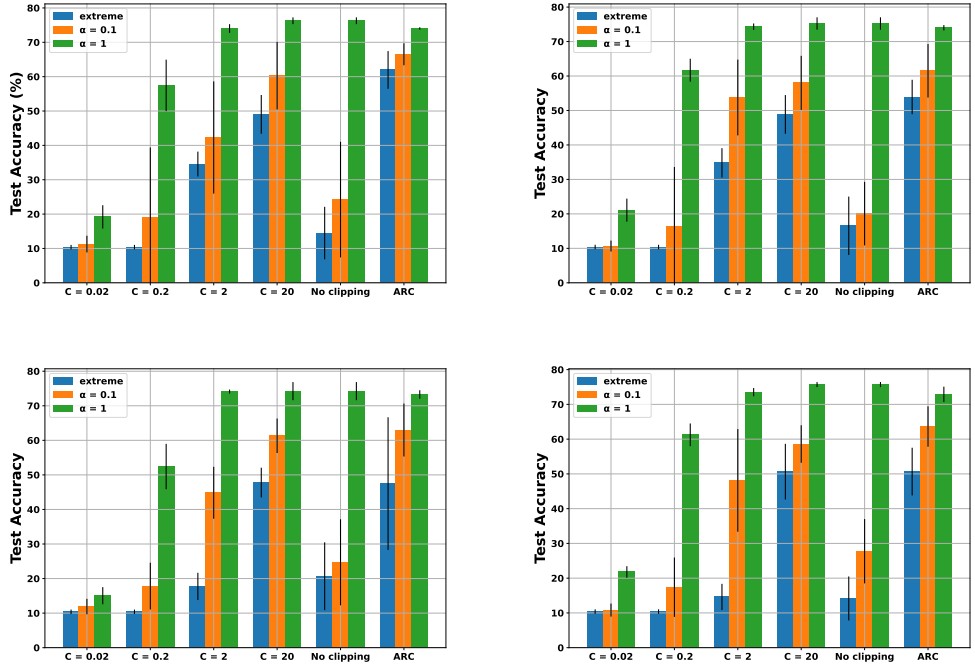

Figure 10: Impact of the clipping strategy and heterogeneity level on the worst-case *maximal* accuracy achieved during the learning against five Byzantine attacks on Fashion-MNIST. $f = 3$ and $n = 15$, and the heterogeneity levels considered are *extreme*, *high* ($\alpha = 0.1$), and *moderate* ($\alpha = 1$). Aggregation methods used: CWTM ∘ NNM (top left), CWMed ∘ NNM (top right), GM ∘ NNM (bottom left), and Multi-Krum ∘ NNM (bottom right). DSGD reaches 85% in accuracy in all settings.

### E.3 COMPARISON WITH STATIC CLIPPING ON CIFAR-10

For this particular experiment, we execute Robust-DSGD using the ResNet-18 (He et al., 2015) model, in a distributed system comprising $n = 9$ workers among which $f \in \{1, 2\}$ are adversarial. We set $b = 128$, $T = 2000$, $\beta = 0.9$, and $\gamma = 0.1$ decaying once by $10\times$ at step 1500. Finally, we use the cross-entropy loss function with an $\ell_2$-regularization of $5 \times 10^{-4}$. See Tables 3 and 4.

| Attack | $C = 0.05$ | $C = 0.5$ | $C = 5$ | $C = 50$ | No clipping | ARC |
|---|---|---|---|---|---|---|
| FOE | $26.4 \pm 1.2$ | $52.6 \pm 2.0$ | $76.4 \pm 5.4$ | $47.0 \pm 8.0$ | $43.0 \pm 6.2$ | $79.1 \pm 1.6$ |
| ALIE | $34.6 \pm 1.3$ | $43.3 \pm 1.1$ | $47.2 \pm 3.4$ | $47.2 \pm 4.2$ | $47.2 \pm 4.2$ | $51.8 \pm 3.5$ |
| LF | $33.7 \pm 1.5$ | $58.4 \pm 1.8$ | $83.6 \pm 1.2$ | $85.7 \pm 0.5$ | $85.9 \pm 0.4$ | $81.5 \pm 1.4$ |
| SF | $32.3 \pm 1.4$ | $57.9 \pm 1.2$ | $77.2 \pm 4.6$ | $68.6 \pm 8.5$ | $66.4 \pm 9.0$ | $82.0 \pm 1.1$ |
| Mimic | $38.2 \pm 1.6$ | $68.3 \pm 1.4$ | $85.6 \pm 0.42$ | $85.7 \pm 0.4$ | $85.7 \pm 0.4$ | $85.4 \pm 0.5$ |
| Worst-case | $26.4 \pm 1.2$ | $43.3 \pm 1.1$ | $47.2 \pm 3.4$ | $47.0 \pm 8.0$ | $43.0 \pm 6.2$ | $51.8 \pm 3.5$ |

Table 3: Maximum accuracy (%) achieved by CWTM ∘ NNM on CIFAR-10 under moderate heterogeneity ($\alpha = 1$), for various clipping strategies and attacks. There are $f = 2$ adversarial workers among $n = 9$. We highlight in blue the highest accuracy achieved per attack (i.e., per row).

| Attack | $C = 0.05$ | $C = 0.5$ | $C = 5$ | $C = 50$ | No clipping | ARC |
|---|---|---|---|---|---|---|
| FOE | $28.9 \pm 1.7$ | $56.8 \pm 1.9$ | $75.6 \pm 2.8$ | $35.5 \pm 11.6$ | $34.9 \pm 12.2$ | $75.9 \pm 3.0$ |
| ALIE | $34.9 \pm 1.3$ | $43.4 \pm 1.0$ | $47.4 \pm 4.1$ | $44.9 \pm 3.8$ | $44.9 \pm 3.8$ | $54.6 \pm 3.6$ |
| LF | $33.7 \pm 1.4$ | $58.4 \pm 1.8$ | $77.4 \pm 12.9$ | $85.8 \pm 0.6$ | $85.6 \pm 0.7$ | $80.5 \pm 1.8$ |
| SF | $32.5 \pm 1.4$ | $57.5 \pm 1.4$ | $77.7 \pm 5.0$ | $63.0 \pm 12.0$ | $55.4 \pm 18.9$ | $82.5 \pm 0.3$ |
| Mimic | $38.1 \pm 1.3$ | $68.5 \pm 1.4$ | $85.5 \pm 0.5$ | $85.9 \pm 0.5$ | $85.9 \pm 0.5$ | $85.4 \pm 0.2$ |
| Worst-case | $28.9 \pm 1.7$ | $43.4 \pm 1.0$ | $47.4 \pm 4.1$ | $35.5 \pm 11.6$ | $34.9 \pm 12.2$ | $54.6 \pm 3.6$ |

Table 4: Maximum accuracy (%) achieved by CWMed ∘ NNM on heterogeneous CIFAR-10 ($\alpha = 1$), for various clipping strategies and attacks. There are $f = 2$ adversarial workers among $n = 9$. We highlight in blue the highest accuracy achieved per attack (i.e., per row).

# F    PERFORMANCE GAINS OF ARC OVER ROBUST-DSGD

In this section, we present additional experimental results demonstrating the performance improvements achieved by ARC in Robust-DSGD on the MNIST, Fashion-MNIST, and CIFAR-10 datasets. These results could not be included in the main paper due to space constraints.

## F.1    MNIST

### F.1.1    INITIAL SYSTEM (MAIN PAPER): 10 HONEST WORKERS

We consider training on MNIST in a system comprised of $n - f = 10$ honest workers. These results complement the ones presented in Section 4 of the main paper.

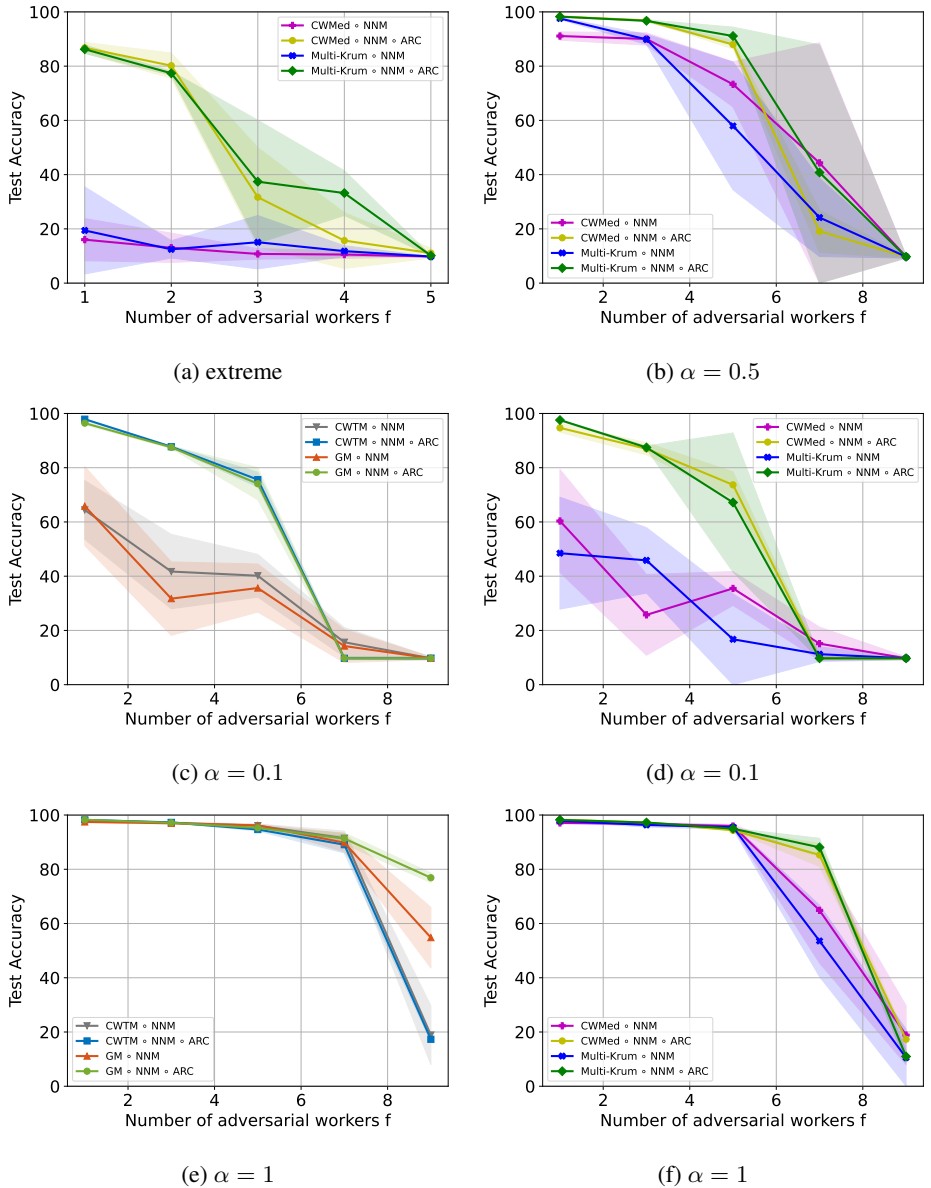

Figure 11: *Worst-case maximal accuracies* achieved by Robust-DSGD when using ARC compared to no clipping, on heterogeneously-distributed MNIST with 10 honest workers. We fix the heterogeneity level, and vary the the number of adversarial workers $f$.

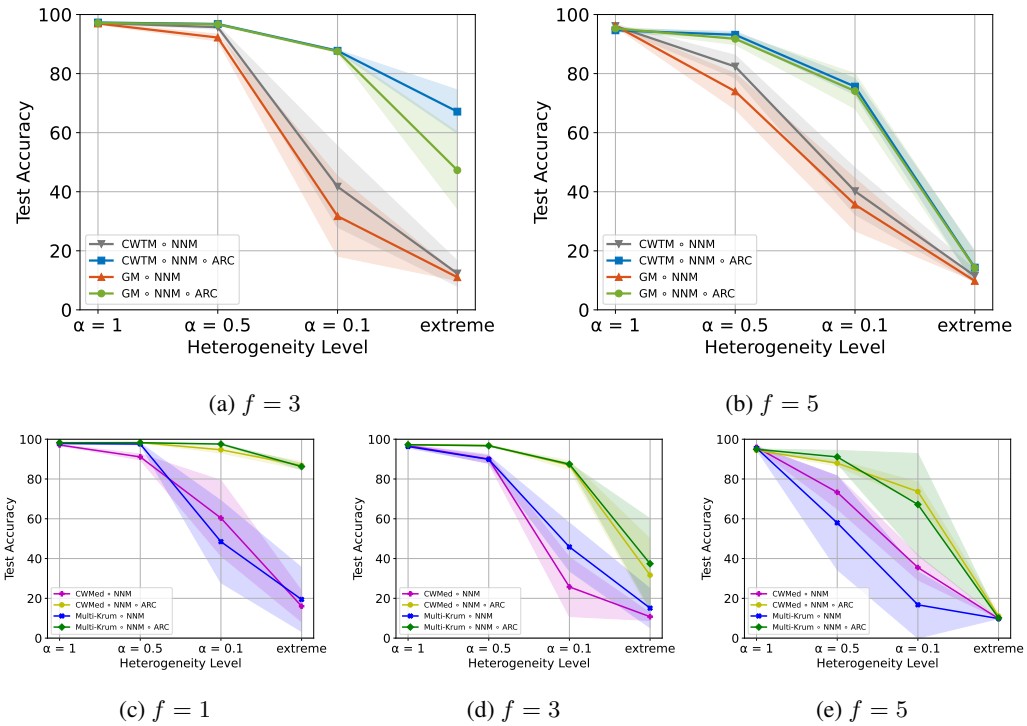

(a) $f = 3$

(b) $f = 5$

(c) $f = 1$

(d) $f = 3$

(e) $f = 5$

Figure 12: *Worst-case maximal accuracies* achieved by Robust-DSGD when using ARC compared to no clipping, on heterogeneously-distributed MNIST with 10 honest workers. We fix the number of adversarial workers $f$, and vary the heterogeneity level.

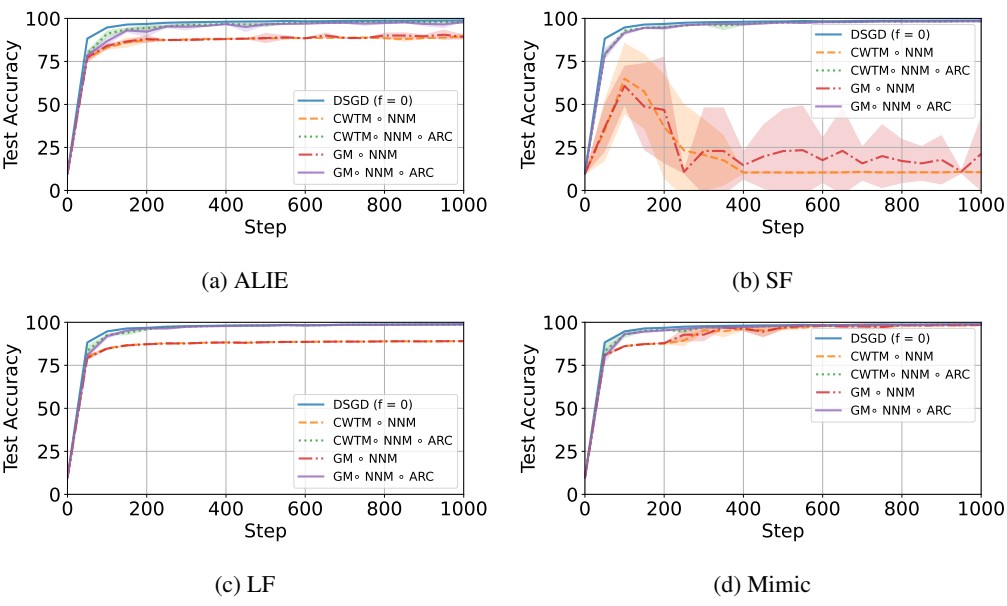

(a) ALIE

(b) SF

(c) LF

(d) Mimic

Figure 13: Performance of Robust-DSGD when using ARC compared to no clipping, on heterogeneously-distributed MNIST (extreme heterogeneity) with 10 honest workers and $f = 1$, under several attacks. This complements Figure 3c of the main paper.

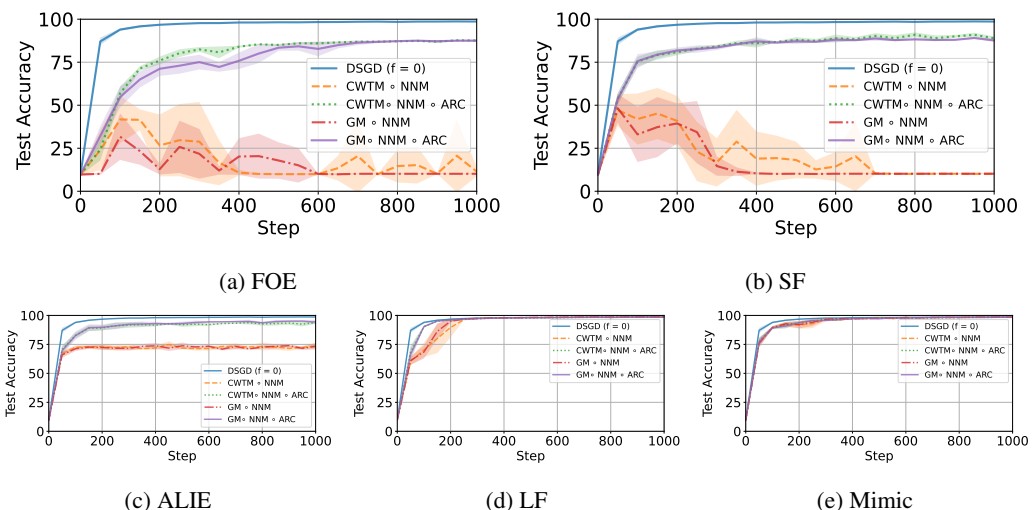

(a) FOE

(b) SF

(c) ALIE

(d) LF

(e) Mimic

Figure 14: Performance of Robust-DSGD when using ARC compared to no clipping, on heterogeneous MNIST ($\alpha = 0.1$) with 10 honest workers and $f = 3$, under several attacks.

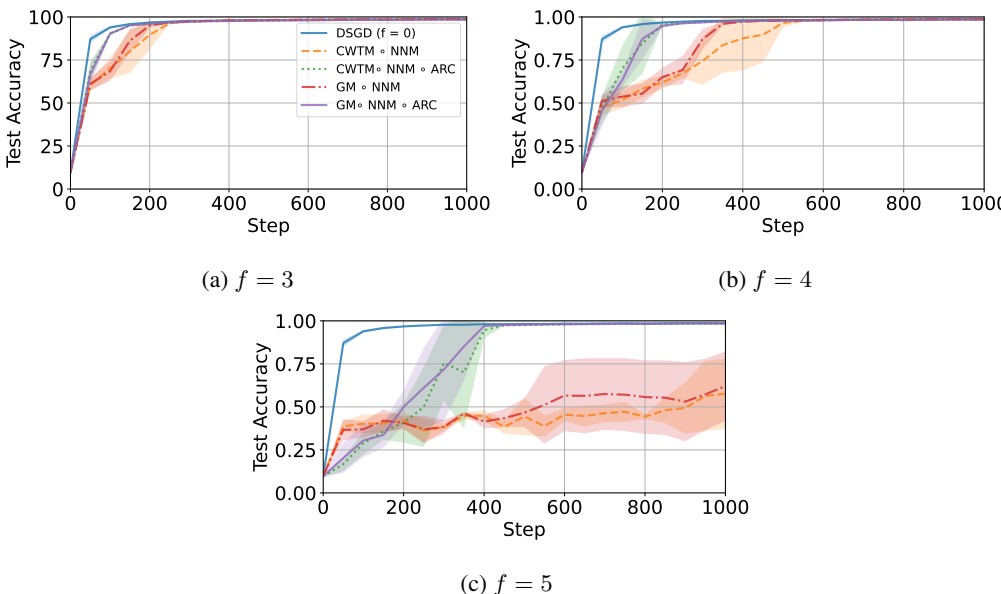

(a) $f = 3$

(b) $f = 4$

(c) $f = 5$

Figure 15: Performance of Robust-DSGD under the LF attack when using ARC compared to no clipping, on heterogeneously-distributed MNIST ($\alpha = 0.1$) with 10 honest workers and varying number of adversarial workers $f$.

### F.1.2 LARGER SYSTEM: 30 HONEST WORKERS

We consider training on the MNIST dataset in a larger system comprised of $n - f = 30$ honest workers, and $f \in \{3, 6, 9\}$ adversarial workers.

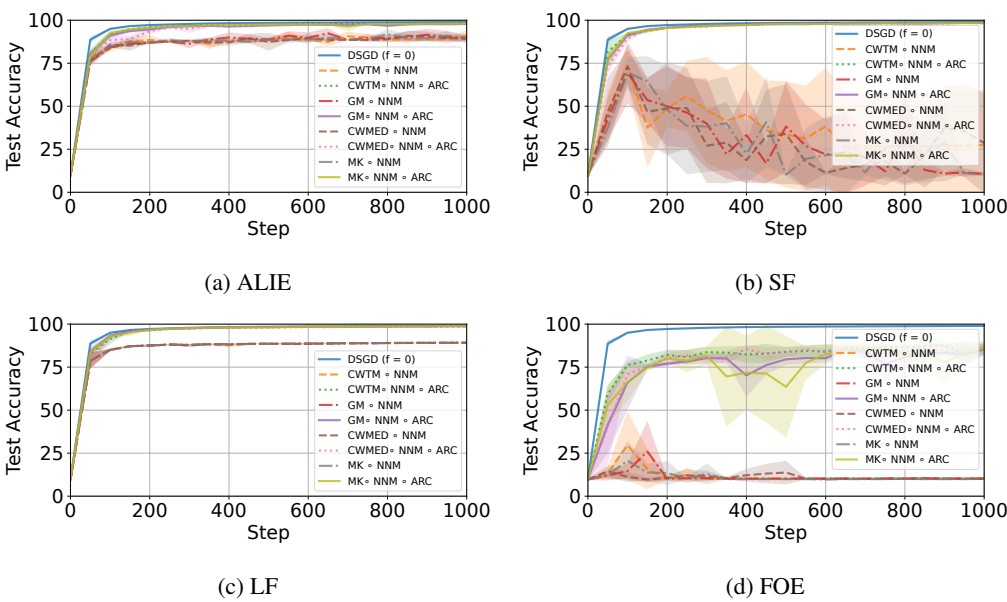

(a) ALIE

(b) SF

(c) LF

(d) FOE

Figure 16: Performance of Robust-DSGD when using ARC and without clipping on distributed MNIST under *extreme* heterogeneity. There are $f = 3$ adversarial workers executing 4 attacks.

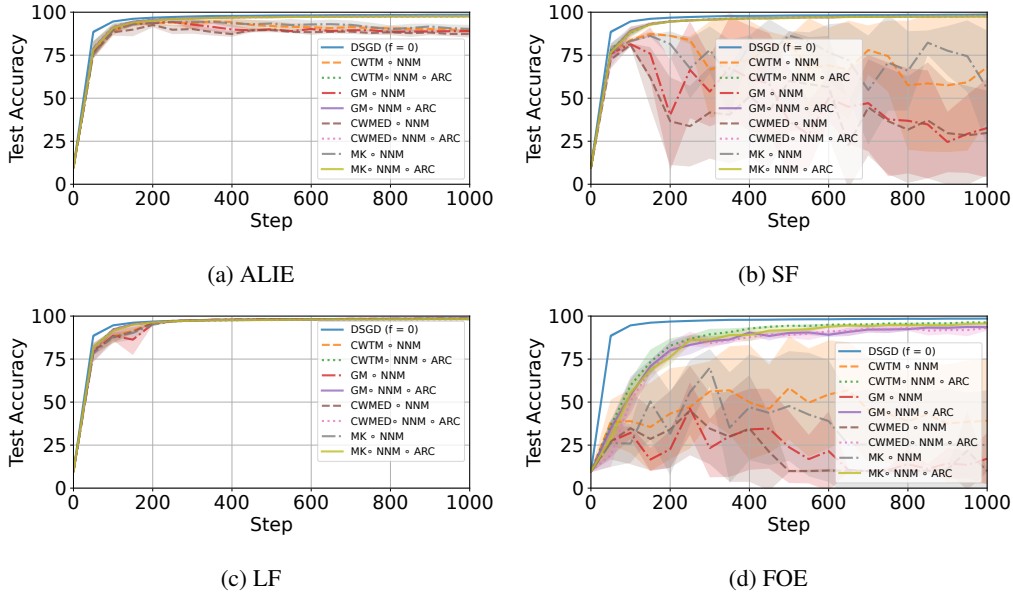

(a) ALIE

(b) SF

(c) LF

(d) FOE

Figure 17: Performance of Robust-DSGD when using ARC and without clipping on heterogeneously-distributed MNIST with $\alpha = 0.1$. There are $f = 6$ adversarial workers executing 4 attacks.

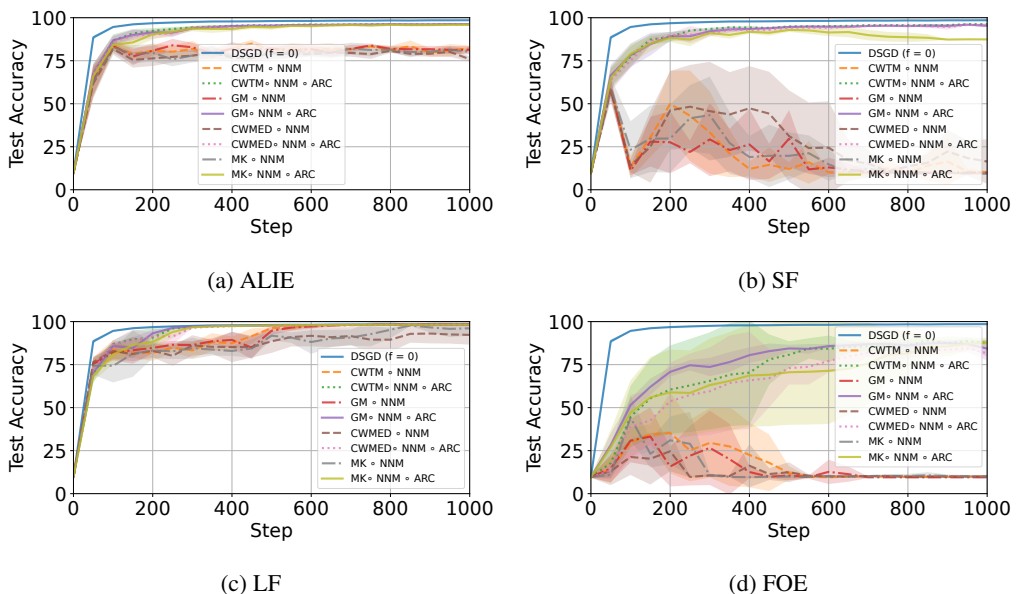

Figure 18: Performance of Robust-DSGD when using ARC and without clipping on heterogeneously-distributed MNIST with $\alpha = 0.1$. There are $f = 9$ adversarial workers executing 4 attacks.

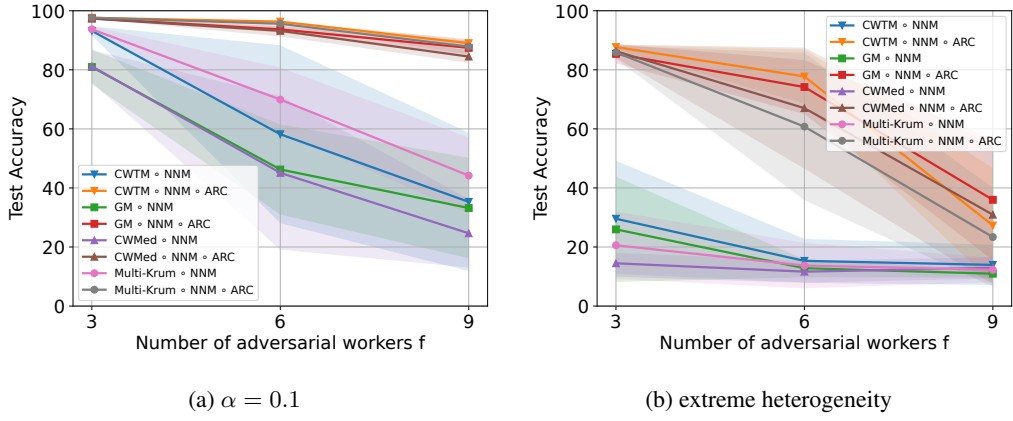

Figure 19: *Worst-case maximal accuracies* achieved by Robust-DSGD, with and without ARC, on heterogeneously-distributed MNIST with $30$ honest workers. We consider a heterogeneous data distribution with $\alpha = 0.1$ (*left*) and extreme heterogeneity (*right*), and vary $f \in \{3, 6, 9\}$.

## F.2   FASHION-MNIST

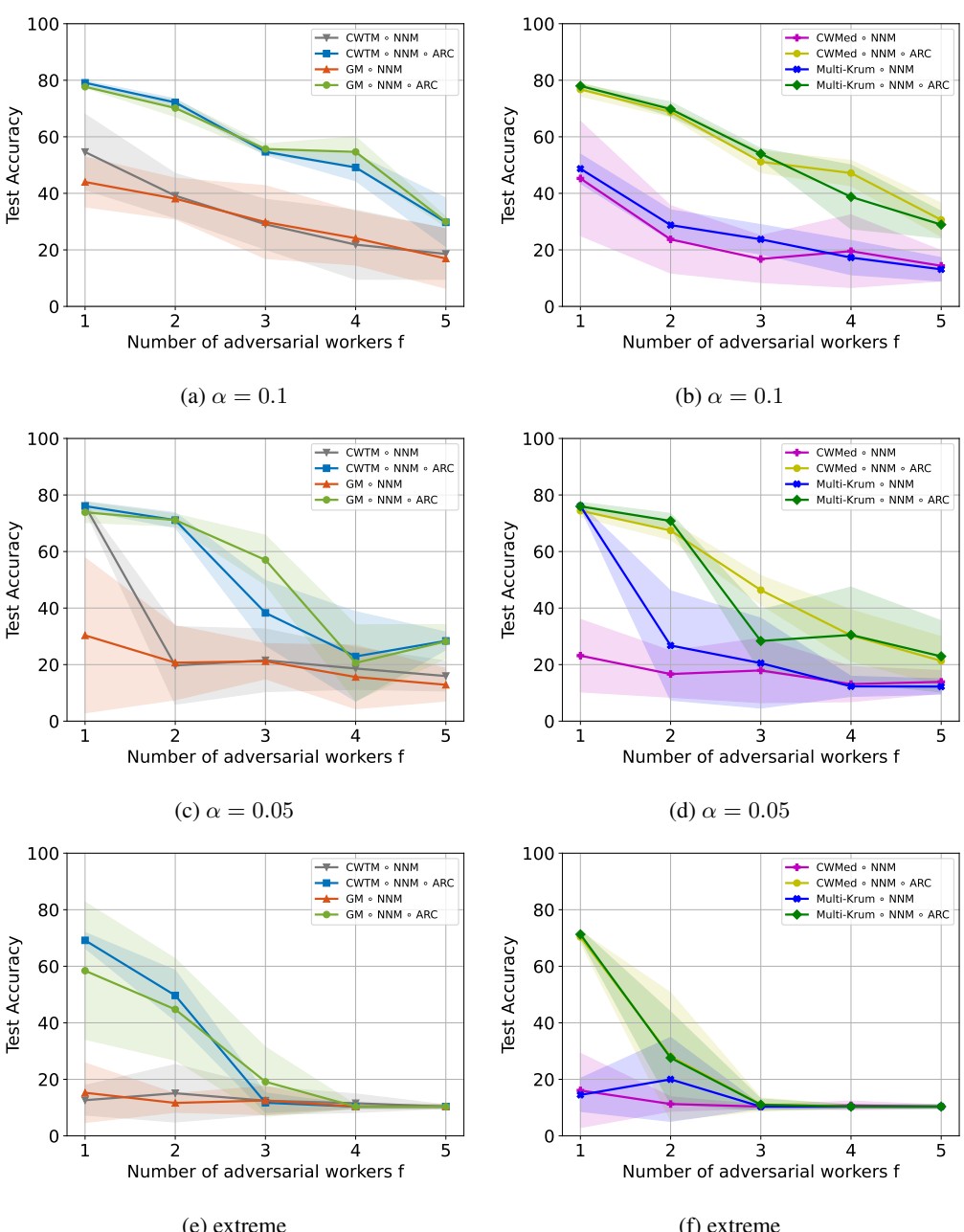

Figure 20: *Worst-case maximal accuracies* achieved by Robust-DSGD when using ARC compared to no clipping, on heterogeneously-distributed Fashion-MNIST with 10 honest workers. We fix the heterogeneity level and vary the number of adversarial workers $f$.

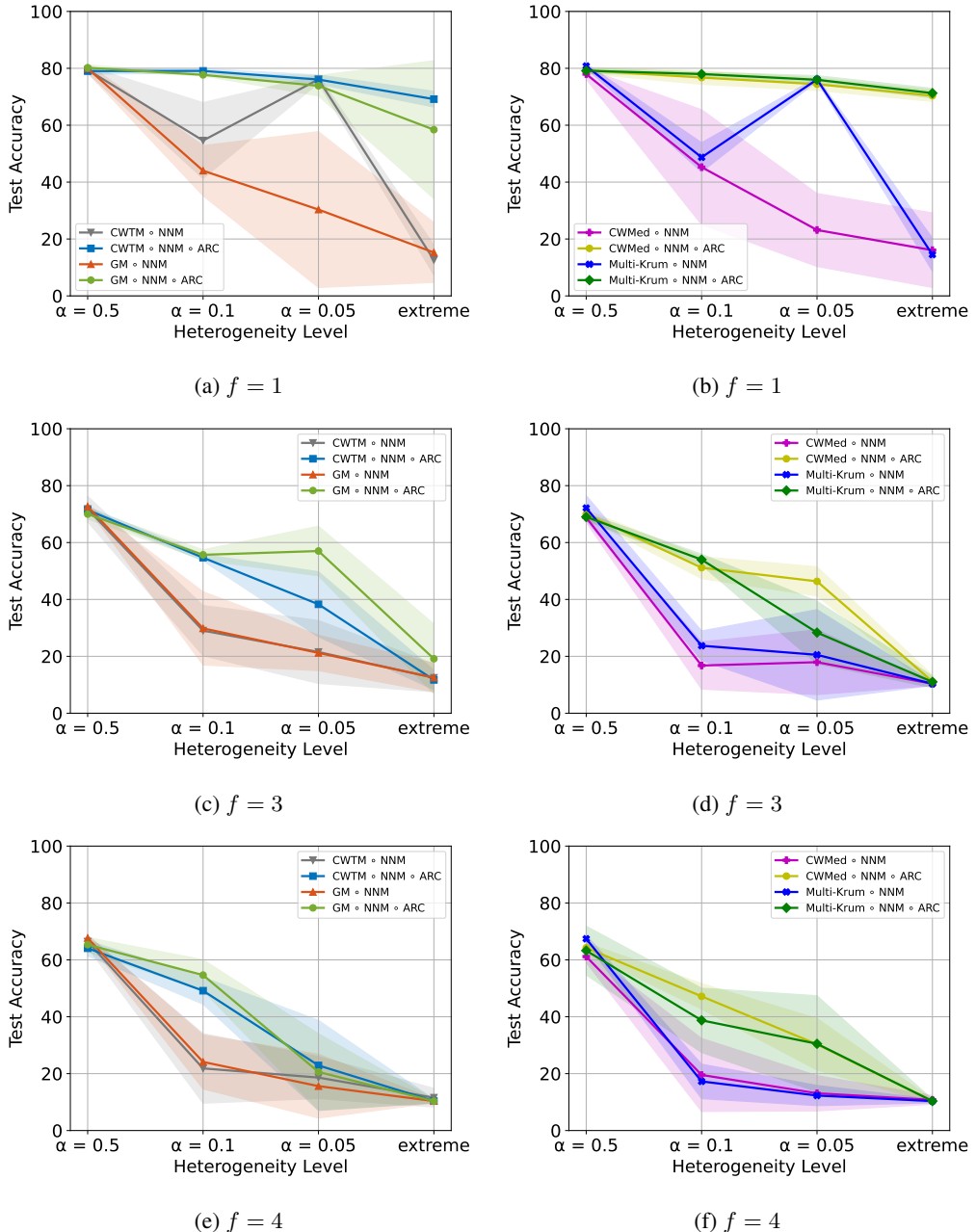

(a) $f = 1$

(b) $f = 1$

(c) $f = 3$

(d) $f = 3$

(e) $f = 4$

(f) $f = 4$

Figure 21: *Worst-case maximal accuracies* achieved by Robust-DSGD when using ARC compared to no clipping, on heterogeneously-distributed Fashion-MNIST with 10 honest workers. We fix the number of adversarial workers $f$ and vary the heterogeneity level.

### F.3 CIFAR-10

**Initial System (Main Paper): 16 Honest Workers**

| | $\alpha = 0.2$ | | $\alpha = 0.075$ | |
| Aggregation | No Clipping | ARC | No Clipping | ARC |
| --- | --- | --- | --- | --- |
| CWMed | $43.4 \pm 4.2$ | $\mathbf{69.4 \pm 0.8}$ | $13.7 \pm 12.1$ | $\mathbf{62.7 \pm 1.2}$ |
| MK | $50.9 \pm 5.1$ | $\mathbf{68.7 \pm 0.5}$ | $40.5 \pm 0.7$ | $\mathbf{59.9 \pm 1.9}$ |

Table 5: *Worst-case maximal accuracies* (%) achieved by Robust-DSGD on heterogeneously-distributed CIFAR-10 with ARC and without. There is $f = 1$ adversarial worker among $n = 17$. As a baseline, DSGD ($f = 0$) reaches 76.5% and 70% when $\alpha = 0.2$ and 0.075, respectively. This table complements Table 1 in the main paper.

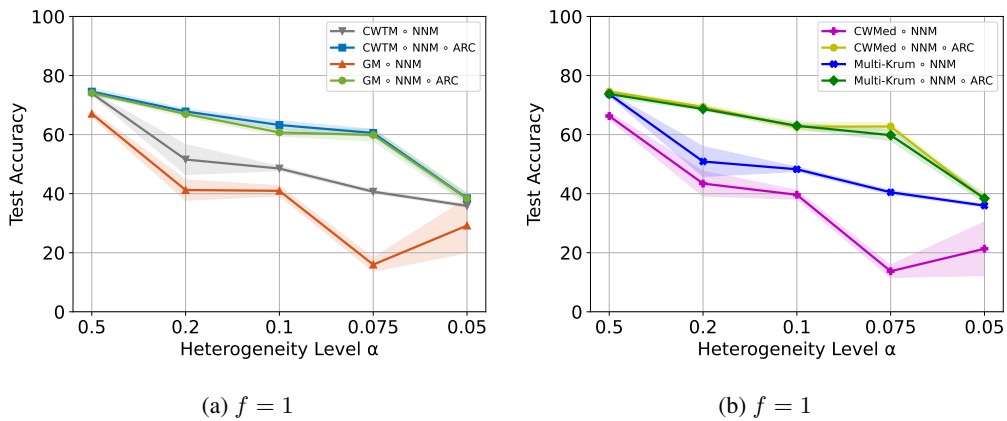

(a) $f = 1$             (b) $f = 1$

Figure 22: *Worst-case maximal accuracies* achieved by Robust-DSGD when using ARC compared to no clipping, on heterogeneously-distributed CIFAR-10 with 16 honest workers. We fix the number of adversarial workers $f = 1$ and vary the heterogeneity level.

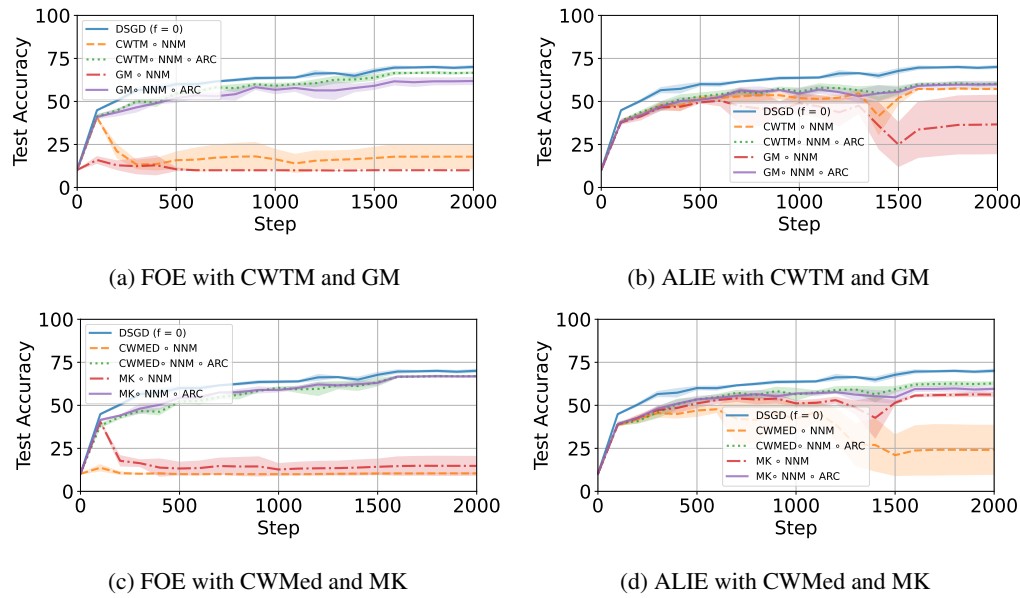

(a) FOE with CWTM and GM          (b) ALIE with CWTM and GM

(c) FOE with CWMed and MK          (d) ALIE with CWMed and MK

Figure 23: Performance achieved by Robust-DSGD when using ARC compared to no clipping, on heterogeneously-distributed CIFAR-10 ($\alpha = 0.075$) with 16 honest workers and $f = 1$ executing ALIE and FOE.

We also test the performance of ARC in more adversarial regimes, with $f \in \{2,3\}$ adversarial workers (along with $n - f = 16$ honest workers). We consider heterogeneity regimes of $\alpha = 0.2$ and 0.5, when $f = 2$ and 3, respectively. The results are presented in Table 6, and we show in Figure 24 the performance of ARC compared to no clipping under the FOE attack.

| Aggregation | $f = 2, \alpha = 0.2$ | | $f = 3, \alpha = 0.5$ | |
|---|---|---|---|---|
| | No Clipping | ARC | No Clipping | ARC |
| CWTM $\circ$ NNM | $44.3 \pm 5.2$ | $\mathbf{53.0 \pm 4.7}$ | $52.0 \pm 1.3$ | $\mathbf{55.4 \pm 1.1}$ |
| GM $\circ$ NNM | $33.8 \pm 6.7$ | $\mathbf{50.2 \pm 3.3}$ | $50.9 \pm 2.5$ | $\mathbf{49.6 \pm 1.4}$ |
| CWMed $\circ$ NNM | $32.6 \pm 9.8$ | $\mathbf{49.7 \pm 2.7}$ | $48.7 \pm 2.4$ | $\mathbf{62.7 \pm 0.9}$ |
| MK $\circ$ NNM | $45.0 \pm 4.2$ | $\mathbf{51.9 \pm 2.1}$ | $50.3 \pm 1.4$ | $\mathbf{50.2 \pm 2.4}$ |

Table 6: *Worst-case maximal accuracies* (%) achieved by Robust-DSGD on heterogeneously-distributed CIFAR-10 with ARC and without. There are $f \in \{2,3\}$ adversarial workers with $n - f = 16$ honest workers.

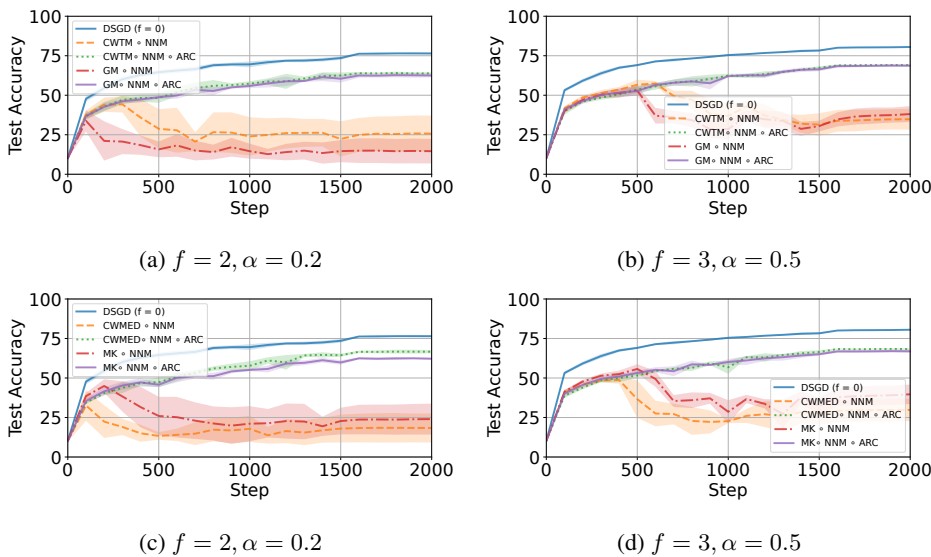

(a) $f = 2, \alpha = 0.2$      (b) $f = 3, \alpha = 0.5$

(c) $f = 2, \alpha = 0.2$      (d) $f = 3, \alpha = 0.5$

Figure 24: Performance of Robust-DSGD when using ARC and without clipping on CIFAR-10. There are 16 honest workers, and $f = 2$ (left) and $f = 3$ (right) adversarial workers executing the FOE attack. The aggregations used are CWTM $\circ$ NNM and GM $\circ$ NNM (top), and CWMed $\circ$ NNM and MK $\circ$ NNM (bottom).

**Larger System: 33 Honest Workers** We also run experiments on CIFAR-10 in a larger system comprised of $n - f = 33$ honest workers and $f \in \{2,3\}$ adversarial workers. We consider the high heterogeneity regime of $\alpha = 0.1$. The results are presented in Table 7, and we show in Figure 25 the performance of ARC compared to no clipping under the FOE attack.

| Aggregation | $f = 2$ | | $f = 3$ | |
|---|---|---|---|---|
| | No Clipping | ARC | No Clipping | ARC |
| CWTM $\circ$ NNM | $45.5 \pm 1.3$ | $\mathbf{54.2 \pm 1.0}$ | $41.3 \pm 0.5$ | $\mathbf{47.5 \pm 1.5}$ |
| GM $\circ$ NNM | $41.8 \pm 1.1$ | $\mathbf{55.2 \pm 2.2}$ | $40.0 \pm 1.2$ | $\mathbf{47.5 \pm 1.5}$ |
| CWMed $\circ$ NNM | $42.5 \pm 1.42$ | $\mathbf{54.7 \pm 1.2}$ | $35.7 \pm 8.7$ | $\mathbf{49.7 \pm 3.4}$ |
| MK $\circ$ NNM | $45.8 \pm 0.8$ | $\mathbf{54.8 \pm 2.5}$ | $41.4 \pm 0.6$ | $\mathbf{48.7 \pm 2.8}$ |

Table 7: *Worst-case maximal accuracies* (%) achieved by Robust-DSGD on heterogeneously-distributed CIFAR-10 with ARC and without. There are $f = 2$ (left) and $f = 3$ (right) adversarial workers, along with $n - f = 33$ honest workers. We consider the high heterogeneity regime $\alpha = 0.1$.

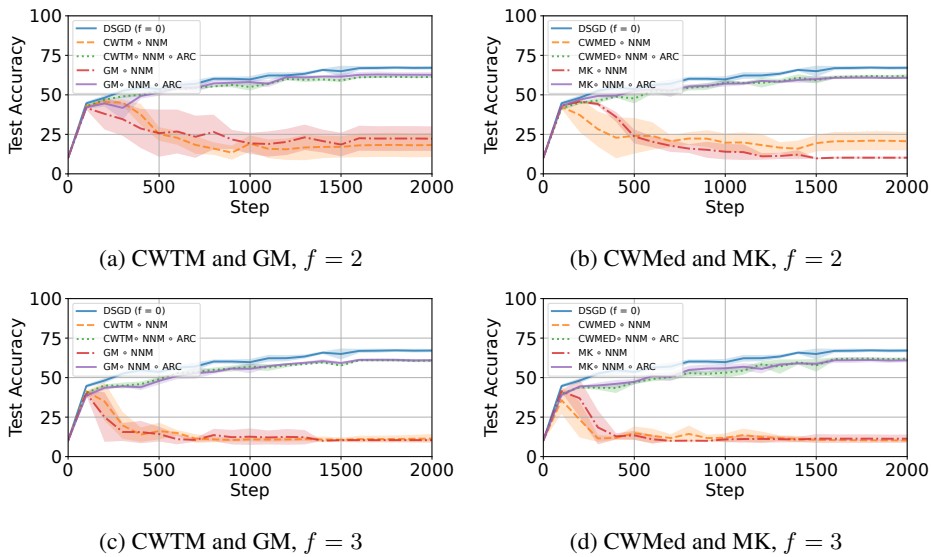

(a) CWTM and GM, $f = 2$        (b) CWMed and MK, $f = 2$

(c) CWTM and GM, $f = 3$        (d) CWMed and MK, $f = 3$

Figure 25: Performance of Robust-DSGD when using ARC and without clipping on CIFAR-10. There are 33 honest workers, and $f = 2$ (top) and 3 (bottom) adversarial workers executing the FOE attack. We consider the high heterogeneity regime $\alpha = 0.1$. The aggregations used are CWTM ∘ NNM and GM ∘ NNM (left), and CWMed ∘ NNM and MK ∘ NNM (right).

## F.4 RUNTIME COMPARISON: NNM (ALLOUAH ET AL., 2023A) VS. ARC

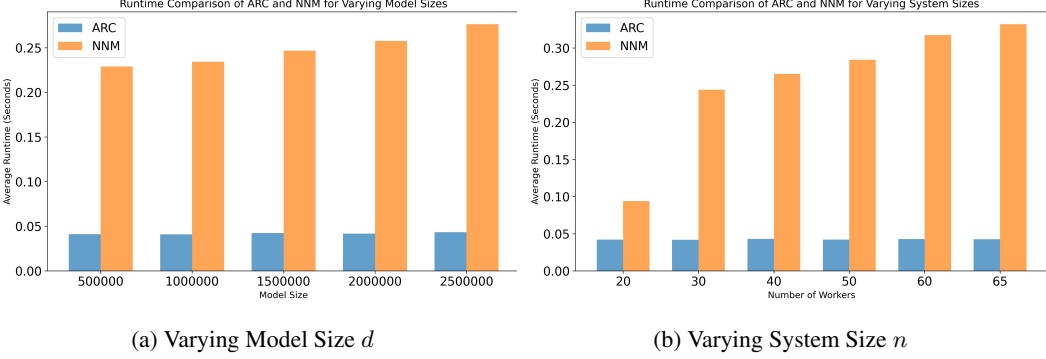

(a) Varying Model Size $d$        (b) Varying System Size $n$

Figure 26: Computational performance of ARC vs NNM in two different settings. In the left plot, we fix the total number of workers $n = 30$, the number of adversarial workers $f = 3$, and vary the model size $d$. In the right plot, we fix the model size $d = 1310922$ (the size of the model we used on CIFAR-10) and $f = 3$, and vary the number of workers $n$ in the distributed system.

