# OpenReview forum: "Adaptive Gradient Clipping for Robust Federated Learning"
_ICLR.cc/2025/Conference — ICLR 2025 Spotlight_

### Official Review · Reviewer_n1Gk · 2024-10-22

**Soundness:** 2
**Presentation:** 3
**Contribution:** 4
**Rating:** 8
**Confidence:** 5

**Summary:**

The authors introduce a variation on the static clipping technique used to overcome byzantine attacks in federated learning. Their algorithm makes the process dynamic by adapting the clipping value to the input gradients themselves; this algorithm, called ARC, or Adaptive Robust Clipping, is proved to be robust: (f, 3k)-robust. More importantly, the authors prove that static clipping breaks the standard (f,k)-robustness, which highlights the shortcomings of the empirical results demonstrated in the papers highlighted in paragraph five of the Introduction. These reasons motivate the need for a dynamic approach to gradient clipping. ARC was paired with and tested using the following techniques: coordinate-wise trimmed mean, geometric median, multi-Krum, and nearest-neighbor mixing. Various simulations were ran by the authors to show the utility of ARC, these include: simulations on varying degrees of data heterogeneity, simulations on varying f (the number of malicious agents), simulations showing the improved breakdown point. All of these simulations show how ARC can provide robustness.

A current problem point is that the authors perform simulations and demonstrate against not using gradient clipping. In paragraph 5 of the Introduction, the authors clearly state that static methods are a problem that their approach, ARC, solves. Then, the authors proceed to perform simulations and do not compare their results against static clipping, but compare against no clipping. It is known, and evidenced by the cited work, that not clipping is a problem that is overcome by using some form of clipping. Therefore, results compared against not clipping yields no additional information. While the authors have shown that ARC has obvious utility that could help overcome known issues, readers cannot determine the excess utility over using static clipping. While I believe the paper holds merit, as the empirical results show, I do not believe the paper can proceed without the authors running the experiments again and showing the results with static gradient clipping. The comparison between static and dynamic clipping is the fundamental point of the paper and not having a comparison of the two makes the paper unqualified to proceed. If the authors can show those results, so readers, such as myself, can see how much improvement is gained by using a dynamic choice for clipping, then I believe the paper will contain enough merit to be accepted and to receive a higher score.

As a final, syntactic, comment, I believe the authors should move the Related Works section to an earlier spot in the paper so readers can more easily understand the historical context and how the motivation for the novel work. This swap will increase flow of understanding for the reader who will have to exert less mental effort to juggle the chronological and intellectual pieces together.

**Strengths:**

The paper introduces dynamic gradient clipping as an improvement to its static counterpart. More importantly it proves that the static approach is not robust, in accordance with the definition of robustness given in the paper. Furthermore, the authors prove the robustness of their approach and provide empirical evidence of the utility of their algorithm.

**Weaknesses:**

The major weakness of the paper, which is a critical one, is the complete lack of comparison of ARC versus static clipping. The authors run experiments against not using clipping; this is rendered moot by prior work and therefore is not a necessary point of comparison. The authors must go back and run the same experiments they ran with with static gradient clipping and plot that against their dynamic approach. Without doing this, it is not possible to determine the benefit their work over prior work.

**Questions:**

1) Why were no experiments ran using static clipping?
2) What is the reason for only using a network with 10 agents? Typically, networks with more agents that test for heterogeneity have a harder time than those with networks because there is a wider gap on intra and inter-class datasets.
3) What was the reason for selecting 17 agents for the simulations in the section "Improved robustness on CIFAR-10"? Can you expand to more agents?

---

> ### Author Response · Authors · 2024-11-16
>
> We thank Reviewer n1Gk for the comments, which we discuss below.
>
> ### Comparison With Static Clipping
> We appreciate the reviewer’s insight into the importance of a direct comparison between ARC and static clipping methods. However, we believe there may be a minor misunderstanding, as we have indeed included an extensive comparison with static clipping in Appendix F (referenced in line 256, Section 3). In this section, we perform experiments using Robust-DSGD on MNIST, evaluating the performance of static clipping across different settings and comparing it with ARC. Specifically, we tested a range of clipping values $C \in $ {0.02, 0.2, 2, 20}, covering various heterogeneity levels.
>
> Our findings indicate that no single static clipping value consistently yields robust performance across diverse scenarios, largely due to the inherent sensitivity of static clipping to factors like data heterogeneity and the nature of Byzantine attacks. Since the server cannot directly access data or a priori determine the attack executed by Byzantine workers, tuning $C$ optimally for static clipping becomes infeasible. This highlights the necessity of a robust, adaptive clipping mechanism like ARC that naturally adjusts to these conditions. In contrast, ARC consistently adapts to different heterogeneity levels and adversarial settings, providing robust performance without the need for parameter tuning.
>
> In addition to the results presented on MNIST, we have also conducted further empirical tests on Fashion-MNIST and CIFAR-10, which support the same observations. Although these results were omitted from the paper, we would be happy to provide them in Appendix G of the revised paper, demonstrating that ARC consistently outperforms static clipping across datasets and conditions. Additionally, for the final version of the paper, we intend to move some key results from Appendix F to the main body to also showcase ARC’s advantages over static clipping.
>
> ### Large-scale experiments
> We acknowledge the reviewer’s point regarding network size and appreciate the suggestion to evaluate ARC’s performance with a greater number of agents. While we used 10 honest workers on MNIST to simulate extreme heterogeneity by assigning each worker a single label, we agree that expanding our study to larger networks would enhance the practical relevance of our results. Preliminary experiments with 30 honest workers on MNIST (tripling the initial system size) affirm ARC’s consistent empirical advantages even in larger setups. We are currently running additional experiments in this setting, and will make these results available shortly in Appendix G of the revised paper.
> Regarding CIFAR-10, we chose a network of 17 workers to create a system larger than what we had for MNIST, as suggested. However, we are constrained by the significant computational resources required for these experiments, which limits our ability to expand further.
> We hope the reviewer can appreciate this trade-off, as CIFAR-10’s computational demands are high, and we aimed to balance system size with experimental feasibility.

---

> > ### Comment · Reviewer_n1Gk · 2024-11-18
> >
> > Comment on "Comparison with Static Clipping"
> > I appreciate the comment from the authors regarding their extensive testing. I kindly remind the authors that reviewers are not under the obligation to review the Appendix. For this reason I gave the comment which I did. Considering a comparison with static clipping is the most important point in the paper, I strongly urge the authors to rearrange their writing to include the results of Appendix F.
> >
> > Comment on "Large Scale Computing"
> > In the paper, it is stated that ARC does not introduce significant overhead. How does that statement add up to the comment provided by the authors stating that they lack computational resources for an experiment with more than 17 workers? I believe it is important to add an appendix section and a comment in the paper itself referencing the computational power limitations of the framework or at least to make a statement on the machinery used to justify the limited resources.

---

> > > ### Author Response · Authors · 2024-11-19
> > >
> > > We thank Reviewer n1Gk for their prompt and constructive feedback.
> > >
> > > **Comparison with Static Clipping**
> > > We completely agree with the reviewer on the critical importance of comparing ARC with static clipping. To address this concern, we will summarize the extensive comparison to static clipping in Appendix F and move the key results to the main paper. This adjustment will ensure that readers can easily see that ARC not only outperforms unclipped strategies but also demonstrates clear advantages over static clipping methods. We believe this change will emphasize the core contributions of the paper.
> > >
> > > **Large-Scale Computing**
> > > We would like to point out to the reviewer that it is not ARC that constitutes the bottleneck in our CIFAR-10 experiments, but rather NNM. ARC’s computational complexity is $\mathcal{O}(nd + n\log(n))$, while NNM’s complexity is
> > > $\mathcal{O}(dn^2)$, making the latter significantly more resource-intensive, particularly as the number of workers scales up. To address this, we have included detailed runtime experiments in Appendix G.2, which explicitly illustrate the substantial overhead introduced by NNM compared to the comparatively much smaller computational cost associated with ARC. We encourage the reviewer to review this section for further clarification.
> > >
> > > As suggested by the reviewer, we will include a discussion in the main paper on the runtime of ARC compared to NNM. Additionally, we will describe the computational resources and machinery used for our experiments for full transparency.
> > >
> > > Lastly, we would like to share that we have initiated experiments on CIFAR-10 with 35 workers, effectively doubling the initial system size. While we aim to complete these experiments before the discussion period ends, we cannot guarantee their completion within this timeframe. Regardless, we plan to include these results in the final version of the paper to further strengthen our contributions.

---

> > > > ### Author Response · Authors · 2024-11-22
> > > >
> > > > We would like to inform Reviewer n1Gk that the requested larger-scale experiments on CIFAR-10 are completed. In these experiments, we considered a larger system consisting of $n−f=33$ honest workers with
> > > > $f=2$ and $f=3$ Byzantine workers.
> > > >
> > > > The results of these additional experiments have been included in Appendix G.3.2 of the revised paper. Notably, these experiments confirm and reinforce the observations presented in the main paper, demonstrating ARC’s ability to enhance performance even in larger systems. Given their significance, we plan to incorporate these results into the main paper to further highlight ARC’s scalability and effectiveness.
> > > >
> > > > We sincerely thank the reviewer for their constructive feedback and the insightful changes they proposed. We hope we have adequately addressed all concerns, and we respectfully anticipate that these revisions will reflect positively in the reviewer’s final assessment.

---

> > > > > ### Author Response · Authors · 2024-11-25
> > > > >
> > > > > We sincerely thank Reviewer n1Gk  for their time and dedication to improving the quality of the paper.
> > > > >
> > > > > As the reviewer suggested, we were able to scale the experiments on CIFAR-10 and have included them in the revised version of the paper. We also intend to move these results, along with the comparison with static clipping, to the main body of the final version of the paper to ensure these key findings are highlighted appropriately.
> > > > >
> > > > > Thank you once again for your valuable feedback, which has greatly contributed to strengthening the paper.

---

### Official Review · Reviewer_Ap5T · 2024-11-01

**Soundness:** 3
**Presentation:** 3
**Contribution:** 3
**Rating:** 8
**Confidence:** 3

**Summary:**

This paper explores enhancing the robustness of distributed machine learning in the presence of Byzantine clients. The authors propose  Adaptive Robust Clipping (ARC) that improves the robustness of Robust-DGD beyond traditional static clipping techniques. ARC dynamically adjusts clipping thresholds based on gradient magnitudes, allowing it to better counteract adversarial impacts.
The authors provide experiments to demonstrate that ARC improves model robustness against various attacks compared to static clipping methods as well as theory showing that ARC has a similar convergence rate as the classical ones known in the literature.

**Strengths:**

The main strengths are:

-The authors propose Adaptive Robust Clipping (ARC), a new mechanism to enhance robustness in adversarial settings.

-The authors show that ARC almost retains the theoretical robustness guarantees of existing Robust methods while enhancing their practical performance.

-The authors validate ARC through several experiments.

**Weaknesses:**

The main weaknesses are:

-Increased complexity produced by ARC in practical implementation

-ARC performance depends on good model initialization which may degrade the performance in the case of poor initialization.
Did you try some experiments to assess this?

-While ARC improves robustness by adaptively clipping gradients, its thresholding could risk clipping too aggressively in certain settings, potentially discarding useful gradient information.

**Questions:**

See section before.

In addition:

-In Figure 1, C=2 for static clipping (SC) is too small. I think that is why you have a very bad performance of SC. You need to test with bigger values of C as well for SC.

-Can you report the Adaptive C of ARC over steps in these plots?

- Line 94, can you justify why one needs to clip this exact k number gradients? what happens if one clip less or more than this number of gradients?

-The intuitive k is the number of potential malicious workers which is f?

-Line 238, you require \kappa B^2 < 1, (which means B should be small) can you give an example of loss where B is small?

-Line 264, I disagree with the comment that "ARC does not introduce a significant overhead" especially in the case of large models. It will be good to have some experiments with runtime on the x-axis

-ARC theory requires that the model is well-initialized, it will be good to assess numerically the impact of the initialization on the performance of ARC

---

> ### Author Response · Authors · 2024-11-16
>
> We thank Reviewer Ap5T for the comments, which we discuss below.
>
> ### Small Static Clipping Threshold
> We appreciate the reviewer’s recommendation to examine the impact of larger static clipping thresholds. In fact, we already tested a broad range of clipping values in our experiments to provide a robust comparison between ARC and static clipping methods. Appendix F presents the results of this comparison, which spans several static clipping parameters (from $0.2$ to $20$) across different levels of heterogeneity, numbers of Byzantine workers, and Byzantine attacks. As our results indicate, no static clipping threshold provides consistent robustness across various heterogeneity levels and Byzantine attack scenarios. By contrast, ARC’s adaptive mechanism enables it to consistently adjust to these changing conditions, delivering stable performance. For further details, please refer to our response to Reviewer n1Gk.
>
> Additionally, we highlight ARC’s adaptivity in Figure 19, where we plot the evolving adaptive clipping parameter over time under the same conditions as those in our static clipping experiments. This demonstrates ARC’s ability to dynamically adjust its threshold in response to ongoing training conditions.
>
> ### Influence of Model Initialization
> We believe the reviewer may have overlooked our analysis of model initialization on ARC's performance, which is presented in Figure 2b of the Introduction and Figure 6 in Section 5.2. In these experiments, we assess the impact of progressively worse initialization conditions on ARC-enhanced Robust-DSGD for MNIST with 10 honest workers, under extreme heterogeneity and at $\alpha = 0.1$ with  $f = 1$ adversarial worker. Beginning with well-chosen initial parameters, we scale them by a factor $\mu$
> where increasing $\mu$ corresponds to worse initialization ($\mu \in \{1, ..., 5\}$).
> Our findings indicate that ARC significantly improves performance under good initialization ($\mu=1$), resulting in substantial gains in accuracy. As initialization degrades, ARC’s advantage over vanilla Robust-DSGD gradually diminishes. However, even under poor initialization ($\mu=5$), ARC still matches the performance of unmodified Robust-DSGD. Importantly, ARC’s behavior aligns closely with Byzantine-free DSGD (which also degrades in performance under unfavorable initialization conditions), while plain Robust-DSGD struggles to leverage well-initialized models effectively, maintaining lower accuracy around 20\% in Figure 2b.
> This analysis underscores the positive influence of good initialization on ARC’s effectiveness and empirically supports our theoretical insights.
>
> ### Intuition for the Design of ARC
> We thank the reviewer for prompting a discussion on ARC’s design and clipping mechanism.
> Lemma B.1 in Appendix B shows that $\mathbf{F} \circ \mathbf{Clip}_C$ is $(f, \tilde \kappa)$-robust, provided that $\lvert S \setminus S_c\rvert \geq 1$ for all subsets $S$ of size $n - f$.
> Note that this condition is impossible to guarantee when using a fixed clipping threshold that does not depend on the given set of input vectors. In order to ensure that $\lvert S\setminus S_c\lvert \geq 1$ for all subsets $S$ of size $n-f$, the clipping threshold $C$ should be large enough such that less than $n - f$ input vectors are clipped.
> Accordingly, we propose to choose a clipping threshold such that the total number of clipped vectors is of the form $\lfloor \lambda (n-f) \rfloor$, where $\lambda < 1$. Note that it is natural to clip more vectors when the fraction of adversarial workers $\frac{f}{n}$ is large, to control the impact of Byzantine behavior.
> Therefore, we set $\lambda \coloneqq \zeta \frac{f}{n}$ where $0 \leq \zeta \leq 2$. Since $\frac{f}{n}<\frac{1}{2}$ and $\lambda < 1$, the number of clipped vectors $\lfloor \zeta \frac{f}{n} (n - f)\rfloor < n-f$ for all $\zeta \in [0, 2]$.
> This constitutes the underlying idea behind our adaptive clipping strategy ARC.
> Furthermore, note that our theory holds for all $\zeta \in [0, 2]$, but we observed empirically that $\zeta = 2$ provides ARC with the best performance.
> We thank the reviewer for this question. We will indeed include this discussion in the paper.

---

> ### Author Response · Authors · 2024-11-16
>
> ### On the Overhead of ARC
> ARC’s complexity is $\mathcal{O}(nd + n\log(n))$(see Appendix A), with
> $n$ as the number of workers and $d$ as the model dimension.
> While it is true that ARC incurs a linear dependence on
> $d$, it is worth noting that this is unavoidable, as all robust aggregation methods must process gradients across model dimensions. Thus, ARC’s linear dependence on $d$ is a fundamental requirement shared across robust aggregations.
> What is particularly critical to consider, however, is the dependence on
> $n$, the number of workers, as this truly distinguishes between efficient and costly aggregation methods, especially in the context of scaling to large language models.
> ARC’s complexity in terms of
> $n$ remains modest whereas more costly aggregation schemes like Nearest Neighbor Mixing (NNM), which is crucial in high heterogeneity, exhibit $\mathcal{O}(dn^2)$ complexity. In distributed ML scenarios where
> $n$ is large, this difference in $n$-scaling makes ARC a comparatively efficient choice.
> However, we recognize the value of further efficiency improvements and agree that developing a more computationally efficient adaptive clipping mechanism that retains ARC’s resilience could be an interesting avenue for future research.
> We will make this clear in the paper and we thank the reviewer for pointing this out.
>
> ### On ARC Clipping Aggressively
> We thank the reviewer for this interesting question. As explained above, the clipping threshold depends on the number of tolerated Byzantine workers $f$. By construction, therefore, the clipping applied by ARC depends on the threat under consideration. If the number of Byzantine workers is small, then ARC will clip only a small portion of the gradients; on the contrary, if the number of Byzantine workers $f$ is large, ARC clips more gradients to limit the impact of malicious vectors while ensuring that at least one honest gradient will remain unclipped (see Lemma B.1).
>
> ### Loss Example where $B$ is Small
> Indeed, there is no guarantee of convergence if $\kappa B^2 \geq 1$. In Figure 3 of *Robust Distributed Learning: Tight Error Bounds and Breakdown Point under Data Heterogeneity* (Allouah et al., page 9), the authors trained a logistic regression model on MNIST and showed that the associated error (taking into account $G$ and $B$) is small, while empirically guaranteeing that $\kappa B^2 < 1$. In this case, the loss function used was the Negative Log Likelihood loss.

---

> > ### Author Response · Authors · 2024-11-19
> >
> > We would like to draw Reviewer Ap5T’s attention to Appendix G.2, where, as requested, we have included a detailed runtime comparison of the computational performance of NNM and ARC across varying numbers of workers and model sizes. These experiments clearly demonstrate the significantly lower computational cost of ARC compared to NNM, particularly as the number of workers increases.
> >
> > We encourage the reviewer to review this new section, and thank them for their valuable feedback.

---

> > > ### Comment · Reviewer_Ap5T · 2024-11-24
> > >
> > > I appreciate the authors' clarification on my earlier questions. I would like to seek further clarification on the following:
> > >
> > > The computation of the adaptive threshold relies on knowing the current number of Byzantine clients in the system, $f$. However, estimating this parameter can be challenging due to the unpredictable nature of Byzantine faults. Could you provide practical guidelines for estimating $f$ when determining adaptive thresholds for clipping? Additionally, what are the potential impacts of overestimating or underestimating the number of attackers on the robustness and performance of ARC?

---

> > > > ### Author Response · Authors · 2024-11-25
> > > >
> > > > We thank Reviewer Ap5T for their thoughtful questions, which we address below.
> > > >
> > > > ### **On the Assumption of Knowing $f$.**
> > > > First, we would like to emphasize that the assumption of knowing the number of Byzantine workers ($f$) is standard in the literature on Byzantine-robust ML. Numerous prior works rely on this assumption, including (Farhadkhani et al., 2022; Allouah et al., 2023a; Karimireddy et al., 2022; Yin et al., 2018; Karimireddy et al., 2021). In practice, $f$ is typically treated as the maximum number of Byzantine workers the system is expected to tolerate, and it is set by practitioners based on their desired level of robustness. We will explicitly clarify this point in the paper to ensure this context is fully transparent.
> > > >
> > > > ### **On Practical Guidelines for Estimating $f$.**
> > > > In practical scenarios, estimating $f$ can indeed be challenging, especially since Byzantine behavior is often unpredictable. In such cases, we recommend that practitioners adopt a conservative approach, setting $f$ based on the worst-case scenario they anticipate for their system. This ensures that the algorithm remains robust under the expected threat level. We will include these practical guidelines in the revised paper to provide actionable advice to practitioners.
> > > >
> > > > ### **Impact of Overestimating or Underestimating $f$.**
> > > > In our framework, $f$ represents an upper bound on the number of Byzantine workers that may exist in the system. Importantly, our theoretical guarantees remain valid as long as the actual number of Byzantine workers does not exceed $f$. If the true number of Byzantine workers is greater than $f$, the guarantees may no longer hold. This distinction will also be explicitly clarified in the revised version of the paper.
> > > >
> > > > The impact of misestimating $f$ on the performance of ARC is discussed below. Note that similar observations also apply to other robust aggregation schemes that use $f$ in their design, such as NNM, coordinate-wise trimmed mean (CWTM), Multi-Krum (see the aforementioned papers).
> > > >
> > > > - **Overestimating $f$:** If $f$ is set higher than the actual number of Byzantine workers, ARC may unnecessary clip honest gradients, which could result in slower convergence.
> > > > - **Underestimating $f$:** If $f$ is underestimated (i.e., the actual number of Byzantine workers exceeds $f$), robustness guarantees may no longer hold, and the system could fail to prevent the influence of adversarial gradients. This is the more critical scenario, which underscores the importance of setting $f$ conservatively.
> > > >
> > > > We will expand the discussion in the paper to address these points, highlighting both the practical considerations for estimating $f$ and the potential impacts of misestimation.
> > > >
> > > > Once again, we thank the reviewer for raising these important points, which will help us further improve the clarity and practical value of the paper. We hope that this response addresses your concerns, and we welcome any additional feedback you may have.

---

### Official Review · Reviewer_zvxn · 2024-11-02

**Soundness:** 2
**Presentation:** 3
**Contribution:** 3
**Rating:** 6
**Confidence:** 3

**Summary:**

The paper introduces an adaptive gradient clipping method applied to worker outputs before passing them through a robust aggregator in heterogeneous synchronous Byzantine settings. The authors address a practical issue, as they observe that while fixed gradient clipping can enhance performance in some cases, it may also impair it in others. To ensure robustness while utilizing gradient clipping, they propose an adaptive strategy that adjusts the clipping threshold according to the magnitude of worker outputs, applying clipping selectively to only a subset of them. Experimental results across various Byzantine scenarios and robust aggregators, tested on MNIST and CIFAR-10 datasets, demonstrate the effectiveness of this adaptive approach when combined with the established NNM method. The authors further support their method with theoretical guarantees.

**Strengths:**

The paper proposes an adaptive method that maintains the robustness guarantees of the aggregators it employs while improving their practical performance, especially under high heterogeneity. The authors provide valuable insights into selecting the clipping threshold, demonstrating that a fixed threshold for all workers, commonly used in practice, may be inefficient in some cases and does not meet robust criteria. They also emphasize the gap between Byzantine theory and practical applications, highlighting that existing theory may not fully capture real-world performance.

**Weaknesses:**

Considering the critical role that numerical evaluation plays in supporting the paper’s claims,
* The paper introduces an adaptive clipping approach designed to work with any robust aggregator independently of NNM. However, the numerical results primarily showcase its effectiveness only when combined with the NNM aggregator (and it is unclear if NNM was also used in Figure 6; if so, this single example may be insufficient). Since NNM has a computational complexity of $O(dn^2)$, it would be valuable to assess the performance of this approach with other robust aggregators (without integrating NNM) to explore potentially lower computational costs. For instance, the CWTM ($O(dn \log{n})$) or the $\epsilon$-approximation GM ($O(dn + d\epsilon^{-2})$) might offer alternatives that may retain robustness in practice while reducing time complexity. Conducting such experiments could provide a more comprehensive evaluation and emphasize the approach’s practicality.
* The CIFAR-10 evaluation is somewhat limited, with only one Byzantine worker out of 17. Expanding the evaluation to include a higher proportion of Byzantine workers and testing on more complex datasets could better demonstrate the method’s effectiveness in more practical scenarios.

**Questions:**

* How does NNM contribute to achieving the guarantees outlined in Theorem 5.2? Is it possible to attain similar results on robust aggregators without incorporating NNM?
* Using a fixed clipping threshold can often be effective in homogeneous Byzantine settings. How does the adaptive approach perform compared to a fixed threshold in such cases?

---

> ### Author Response · Authors · 2024-11-16
>
> We thank Reviewer zvxn for the comments, which we discuss below.
>
> ### On Combining ARC with NNM
> We appreciate the reviewer’s suggestion to explore ARC’s performance independently of NNM. In the main paper, our experiments primarily highlight ARC’s improvement when paired with NNM, the state-of-the-art mixing algorithm for addressing heterogeneity in Byzantine ML. This choice is driven by prior findings showing that NNM significantly enhances the robustness of existing aggregations, especially under conditions of high data heterogeneity or a large proportion of Byzantine workers.
> Without NNM, robust aggregations such as CWTM or GM alone yield poor empirical results in high heterogeneity regimes, as demonstrated in prior work on NNM (see *Fixing by mixing: A recipe for optimal byzantine ml under heterogeneity* (Allouah et al.)).
> Furthermore, in homogeneous settings where NNM is not required, ARC results in comparable or better empirical performance compared to vanilla robust aggregations without ARC.
> Additionally, we confirm that NNM was indeed used in Figure 6, and we will clarify this in the final version.
>
> ### On CIFAR-10 and More Complex Datasets
> In our CIFAR-10 evaluation, we used only one Byzantine worker among $
> n=17$, aiming to demonstrate that even a single adversarial worker ($f=1$, or under 6\% of the system) can significantly degrade the learning when ARC is not used in Robust-DSGD. We agree that assessing ARC under a higher proportion of Byzantine workers would further underscore its practical value. We have already begun running additional experiments with more Byzantine workers, which so far confirm ARC’s robustness benefits under increased $f$ values. We will include these results in the revised version of the paper in Appendix G when they are finalized.
>
> Regarding dataset complexity, our current evaluation leverages standard benchmarks in Byzantine ML: MNIST, Fashion-MNIST, and CIFAR-10. Although these datasets are less challenging in conventional ML, they introduce considerable difficulty under Byzantine settings, as confirmed by prior research (see *Fixing by mixing: A recipe for optimal byzantine ml under heterogeneity* (Allouah et al.), *Byzantine machine learning made easy by resilient averaging of momentums* (Farhadkhani et al.),*Byzantine-Robust Learning on Heterogeneous Datasets via Bucketing* (Karimireddy et al.)). Extending Byzantine ML evaluations to more complex datasets is a promising future direction, and we agree that it could further substantiate ARC’s applicability in practical, large-scale scenarios.
>
> ### Performance of ARC in Homogeneous Settings Compared to Static Clipping
> Thank you for raising the question about ARC’s performance under homogeneous conditions compared to static clipping. As detailed in Appendix F, we compared ARC with various static clipping strategies across three heterogeneity levels. Notably, in moderate heterogeneity in Figure 17 (corresponding to sampling from a  Dirichlet distribution with parameter $\alpha = 1$) — the closest setting to homogeneity considered — ARC performs on par with the best static clipping strategies (e.g., $C = 2$ and $C = 20$) and outperforms others (e.g., $C = 0.2$ and $C = 0.02$). Additional experiments we conducted in completely homogeneous settings further support these observations, showing that ARC matches or exceeds the performance of the best static clipping strategies available. Thus, ARC remains effective even in low-heterogeneity environments, adapting well to homogeneous scenarios without compromising robustness.
>
> ### On NNM's Importance for the Guarantees of Theorem 5.2
> We thank the reviewer for this insightful question.
> NNM ensures that the robustness coefficient $\kappa \in \mathcal{O}(f/n)$ (more specifically, see Lemma 2.5), making it possible to induce an improvement when $f/n$ approaches the breakdown point. This improvement could not be demonstrated otherwise, i.e., if $\kappa$ is not proportional to $f/n$.

---

> > ### Comment · Reviewer_zvxn · 2024-11-24
> >
> > Thank you for your rebuttal. I will raise my score to 6.

---

> > > ### Author Response · Authors · 2024-11-25
> > >
> > > Thank you for your time and the constructive feedback that will improve the quality of the paper.

---

### Official Review · Reviewer_KAQT · 2024-11-04

**Soundness:** 3
**Presentation:** 3
**Contribution:** 4
**Rating:** 8
**Confidence:** 4

**Summary:**

This paper proposes a novel strategy called Adaptive Robust Clipping (ARC) for Byzantine-resilient distributed machine learning. Empirical results show that using ARC can significantly enhance Byzantine resilience compared to the methods without clipping. Theoretical analysis of convergence is also provided to show the effect of ARC.

**Strengths:**

1. This paper is generally well-written.
2. The idea of adaptive clipping intuitively makes sense and has an excellent empirical performance in the experiments of this work.
3. Byzantine resilience in distributed learning is an important and timely topic.

**Weaknesses:**

Although the proposed ARC strategy is generally not hard to implement and has a good empirical performance, there are major concerns about the theoretical analysis in this paper, which I specify point by point below.

1. The theoretical results in section 3 show that $F\circ ARC$ is $(f,3\kappa)$-robust when $F$ is $(f,\kappa)$-robust (Theorem 3.2). Although the property of $ARC$ is much better than trivial clipping (as shown in Lemma 3.1), the convergence guarantee obtained from Theorem 3.2 for $F\circ ARC$ is worse than that for $F$ (without $ARC$). In other words, the theoretical results in section 3 show that $ARC$ has a better property than trivial clipping, but do not show that using $ARC$ can improve the convergence guarantees.

2. The improvement of convergence guarantees for $ARC$ is mainly shown by Theorem 5.2. Theorem 5.2 says that when the maximum gradient norm of the initial point is not larger than $\zeta$, using $ARC\circ F$ can guarantee to find a point $\hat{\theta}$ such that the square norm of the gradient at $\hat{\theta}$ is not larger than $v \epsilon_0$ in expectation. However, $v \epsilon_0$ can be much larger than $\zeta^2$ (which is specified in the next paragraph). Briefly speaking, the result of Theorem 5.2 can be even weaker than the conditions, which makes the theorem meaningless.
- Since $\xi \leq \min(\frac{v}{\Phi(G,B,\rho)},\xi_0)$, it is obtained that $\xi \leq \frac{v}{\Phi(G,B,\rho)}$, and thus $v\geq \xi \cdot \Phi(G,B,\rho)=\xi\cdot 640(1+\frac{1}{B^2})^2(1+\frac{B^2\rho^2}{G^2}).$ Therefore,
$v\epsilon_0 \geq [\xi\cdot 640(1+\frac{1}{B^2})^2(1+\frac{B^2\rho^2}{G^2})]\cdot[\frac{1}{4}\cdot\frac{G^2(f/n)}{1-(2+B^2)(f/n)}].$ The term $\rho^2=\exp (2\frac{(2+B^2)\Delta_0}{(1-\xi_0)G^2}L)\zeta^2$ can be much larger than $\zeta^2$. Thus, $v\epsilon_0$ can be much larger than $\zeta^2$, which will make Theorem 5.2 meaningless.

Overall, the idea of ARC is interesting. The ARC method is easy to implement and has a good empirical performance. However, the theoretical analysis in the current version does not show the improvement of convergence guarantees, and can be misleading.

**Questions:**

Please focus on the weakness of Theorem 5.2 in the rebuttal. Specifically, please compare the value of $v\epsilon_0$ with $\zeta^2$ in Theorem 5.2 (or address the concern in some different ways). I am willing to raise the score if the concerns are properly addressed.

---

> ### Author Response · Authors · 2024-11-16
>
> We thank Reviewer KAQT for the comments, which we discuss below.
> We would like to clarify the purpose of this theorem and address the specific points raised.
>
> ### Purpose of Theorem 5.2
> Theorem 5.2 aims to demonstrate the improvement induced by ARC over existing methods in strong adversarial regimes, specifically when the corruption fraction $\frac{f}{n}$ approaches the breakdown point (BP) of the system. In this extreme setting, ARC ensures that the error is strictly better than the lower bound, provided the norms of honest gradients are bounded at model initialization.
>
> ### Addressing the Reviewer’s Concerns
> The reviewer’s calculations are correct, but they do not fully consider the role of $\xi$, which measures how close the corruption fraction $\frac{f}{n}$ is to the BP.
> In Theorem 5.2, $\xi$ explicitly influences the final error bound $\upsilon\varepsilon_o$.
> Therefore, the lower bound of $\upsilon\varepsilon_o$ computed by the reviewer can be made arbitrarily small since it is proportional to $\xi$, by considering $f/n$ to be arbitrarily close to the BP.
> While $\rho^2$ can indeed be much larger than $\zeta^2$, this does not render the theorem meaningless. The large value of $\rho^2$ reflects the difficulty of maintaining robustness in extreme adversarial regimes. However, the proportionality of $\upsilon\varepsilon_o$ to $\xi$ ensures that the bound remains relevant. Specifically, as $\xi$ becomes small, the influence of $\rho^2$ diminishes relative to the overall bound.
> We refer the reviewer to Appendix C, specifically Theorem C.4 and Corollary C.5, for additional details on the convergence of Robust DGD with ARC. We hope that these additional results will make things clearer.
>
> ### Proposed Revisions for Clarity
> To address potential confusion and enhance the clarity of Theorem 5.2, we will include specific values for $v$ and $\xi_0$ in the final version of the paper.
> Indeed, we will consider a special case of the theorem by choosing $\xi_o = 0.5$. In this case, $\rho \coloneqq \exp\left( \frac{2 (2 + B^2)\Delta_o L}{G^2} \right) \zeta$.
> Furthermore, let $\boldsymbol{\upsilon} \leq \xi_o = 0.5$. Since, in this particular case, $\frac{\boldsymbol{\upsilon}}{\Psi(G, B, \rho)} \leq \xi_o$ (because $\Psi(G, B, \rho) \geq 1$), Theorem 5.2 implies that **if** $0 < \xi \leq \frac{1}{2}\frac{\boldsymbol{1}}{\Psi(G, B, \rho)}$ **then** $\mathbb{E} {\lVert \nabla \mathcal{L}_{\mathcal{H}} \hat{\left ( \theta \right)} \rVert}^2 \leq \boldsymbol{\frac{1}{2}} ~ \varepsilon_o < \varepsilon_o$.
> This special case focuses on the regime of small $\xi$, explicitly illustrating how $\upsilon\varepsilon_0$ (where $\upsilon = \frac{1}{2}$) improves over existing methods under these conditions. Additionally, we will emphasize that the strict improvement shown in Theorem 5.2 is particularly significant when $\frac{f}{n}$ is close to the BP, aligning with the theorem's primary objective.
>
> ### Meaningfulness of Theorem 5.2.
> We believe Theorem 5.2 presents an important result in Byzantine-robust distributed learning. The theorem shows that if the fraction of adversarial workers $f/n$ is close to the breakdown point then the lower bound on the learning error under $(G, B)$-gradient dissimilarity can be circumvented, provided that the honest workers' gradients at model initialization are bounded. This result opens up a new line of research of considering pragmatic distributed learning settings that avoid the worst-case scenarios in the presence of adversarial workers, thereby improving robustness guarantees.
>
> ### Robustness Preservation by ARC
> Yes, we agree that Theorem 3.2, which shows that ARC preserves $(f, 3 \kappa)$-robustness, is *not sufficient* for proving an improvement. Indeed, we show that ARC has an additional property shown in Lemma 5.1 (in Section 5), which in conjunction with Theorem 3.2, yields an improvement in the learning error characterized in Theorem 5.2.
>
> We hope that the above properly addresses the reviewer's concerns. In which case, we expect the reviewer to raise the score of our paper.

---

> > ### Author Response · Authors · 2024-11-24
> >
> > Dear Reviewer KAQT,
> >
> > We sincerely appreciate your detailed feedback and the points you raised regarding Theorem 5.2.
> >
> > In our rebuttal, we have provided a detailed response addressing your concerns about the meaningfulness of Theorem 5.2. Specifically, we clarified how the parameter $\xi$ plays a critical role in ensuring it, especially in adversarial regimes where $\frac{f}{n}$ approaches the breakdown point. This explanation highlights how Theorem 5.2 demonstrates ARC’s strict improvement in such scenarios.
> >
> > We would be grateful if you could review our response and engage in a discussion with us to address any remaining doubts or concerns. We are committed to ensuring that all aspects of the paper are clear and rigorous, and we welcome the opportunity to provide further clarifications if needed.
> >
> > Thank you for your constructive review, and we look forward to hearing your thoughts during the discussion period.

---

> > > ### Comment · Reviewer_KAQT · 2024-11-24
> > >
> > > I thank the authors for the detailed response. I have understood the meaningfulness of Theorem 5.2. Meanwhile, I would like to present the remaining concern below.
> > >
> > > **Remaining concern**: Although I appreciate the authors' explanation about Theorem 5.2, I am wondering why $f/n$ can be arbitrarily close to BP. Integer $n$ is decided by the problem. Thus, the difference between $f/n$ and BP should be at least $1/n$ for any specific problem with $n$ workers. Thus, $\xi=\frac{BP-f/n}{BP}>2/n$ when there are $n$ workers since $BP<1/2$. Therefore, $v\geq\xi\cdot\Phi(G,B,\rho)>\frac{2\Phi(G,B,\rho)}{n}$. Noticing that $\Phi(G,B,\rho)>640$ according to the definition (probably much larger than $640$). Therefore, the theorem shows the improvement of ARC only when $n$ is very large and $f/n$ is extremely close to BP.
> > >
> > > Meanwhile, I would like to clarify that the concern above is just to point out the limitation in application scope of Theorem 5.2. Overall, I am inclined to raise my rating after reading the authors' explanation, and I would like to hear the authors' response to the remaining concern above before finally deciding my rating.

---

> > > > ### Author Response · Authors · 2024-11-25
> > > >
> > > > We sincerely thank the reviewer for their prompt response and willingness to engage in improving the paper. While we acknowledge the reviewer's point that the improvement guaranteed by Theorem 5.2 is more prominent when $n$ is large, the theorem is still applicable when $n$ is small as we explain below.
> > > >
> > > > ### **BP$-f/n$ can be smaller than $1/n$.**
> > > > We respectfully disagree with the reviewer's assertion that the difference between $f/n$ and BP must be at least $1/n$. This difference can, in fact, be significantly smaller than $1/n$, depending on the system configuration and heterogeneity parameter $B$. The BP (Breakdown Point) is defined as the maximum fraction of Byzantine workers that the system can tolerate while maintaining robustness. This value is a constant, independent of $f$ and $n$, and depends on the heterogeneity parameter $B$, as noted in line 419 of the paper. Specifically, BP is given by $\frac{1}{2 + B^2}$. Consequently, there exist configurations where $f/n$ can be extremely close to BP. For example, consider the case where $B^2 = 1$, giving BP $= 1/3$. Now imagine a system with a small number of workers, $n = 10$, among which $f = 3$ are Byzantine. In this case, $f/n = 0.3$, and $\text{BP} - f/n = 1/30 < 1/n$. The difference between $f/n$ and BP can thus be arbitrarily small. Therefore, the inequality on $\upsilon$ provided by the reviewer, which assumes a minimum difference of $1/n$, does not universally hold.
> > > >
> > > > ### **Practical scope of Theorem 5.2.**
> > > > That said, we appreciate the reviewer’s point regarding certain practical limitations of Theorem 5.2. Indeed, there are scenarios where $\text{BP} - f/n$ equals $1/n$ (or more). For instance, consider the same system with $n = 10$ workers but with $f = 2$. In this configuration, $\text{BP} - f/n \geq 0.1 = 1/n$. In this particular case, the inequality provided by the reviewer holds, and Theorem 5.2 does not guarantee an improvement. We acknowledge this practical limitation and will include a discussion in the paper to clearly outline these scenarios. As the reviewer correctly pointed out, when $n$ is large, the range of $f$ for which Theorem 5.2 guarantees an improvement is also large. To **summarize**:
> > > > - When $n$ is **small** relative to $\psi(G, B, \rho)$, Theorem 5.2 may guarantee an improvement for only one value of $f$ (i.e., the largest value of $f$ for which $f/n$ is smaller than BP.).
> > > > - When $n$ is **large** relative to $\psi(G, B, \rho)$, as the reviewer pointed out, Theorem 5.2 guarantees an improvement for a wide range of $f$.
> > > >
> > > > The primary purpose of Theorem 5.2 is to demonstrate that the established lower bound in Byzantine ML can be circumvented in strong adversarial regimes where $f/n$ is very close to the BP, provided the models of honest workers are bounded at initialization. Moreover, our extensive empirical results consistently show a strict improvement induced by ARC over Robust-DSGD across systems of varying sizes (from small to large) and for different numbers of Byzantine workers $f$.
> > > >
> > > > ### **Planned Revisions.**
> > > > We will incorporate the reviewer’s feedback into the final version of the paper by discussing the above practical limitations of Theorem 5.2 and their implications.
> > > >
> > > > Once again, we thank the reviewer for their thoughtful and constructive feedback, and we hope this clarification adequately addresses all remaining concerns.

---

> > > > > ### Comment · Reviewer_KAQT · 2024-11-25
> > > > >
> > > > > I appreciate the authors' follow-up explanations. My concerns are almost properly addressed. As I promised, I will increase my rating to 8.
> > > > >
> > > > > Meanwhile, I still politely disagree with the authors on the statement that 'BP can be arbitrarily small'. I understand that for any $\epsilon>0$, there exists a case such that $BP-f/n<\epsilon$. However, the total worker number $n$ and the breakdown point BP are determined by the problem and should not be considered as variables. Thus, the statement could be misleading. I hope that the authors could re-consider this statement in future versions.

---

> > > > > > ### Author Response · Authors · 2024-11-25
> > > > > >
> > > > > > We thank the reviewer for the for detailed feedback and constructive comments that will help improve the paper.
> > > > > >
> > > > > > We acknowledge the reviewer's point regarding BP - $f/n$ being arbitrarily small.
> > > > > > We agree that $n$ and BP are problem-specific constants rather than variables.
> > > > > > We will clarify this point in the revised version of the paper, and ensure that the limitation on the applicability of Theorem 5.2 when $n$ is small is properly addressed.

---

### Author Response · Authors · 2024-11-18

Dear Reviewers,

We are pleased to inform you that the additional experiments requested have now been completed. These experiments, which address the specific points raised in the reviews, have been incorporated into the revised version of the paper. For ease of reference, we have included these results in Appendix G, and highlighted all additions and revisions in blue.

The new results further reinforce the claims made in the paper and provide additional clarity on the robustness and adaptability of ARC in various settings. We encourage you to review these updates and welcome any further comments or suggestions you may have.

Thank you once again for your time and consideration.

---

### Author Response · Authors · 2024-11-27

We sincerely thank all the reviewers for their constructive comments and insightful suggestions. Your feedback has been invaluable in improving the quality and clarity of the paper.

---

### Meta-Review · Area_Chair_tXob · 2024-12-17

**Metareview:**

This paper proposes a new method for adaptive gradient clipping in Byzantine-resilient distributed setting. It provides both theoretical and empirical support for the method. The discussion during the rebuttal for this paper centered on theoretical rigor, experimental breadth, and practical applicability of Adaptive Robust Clipping (ARC) in Byzantine-resilient distributed learning. Reviewers raised concerns about the limitations of the theoretical guarantees, particularly in highly adversarial regimes, the dependence of ARC on robust initialization, and the lack of direct comparisons to static clipping in the main text. Authors addressed these points by providing detailed clarifications on theoretical underpinnings, incorporating additional experiments comparing ARC to static clipping, and demonstrating ARC's scalability through new larger-scale experiments. Finally, all reviewers were very positive and hence I recommend acceptance.

**Additional Comments On Reviewer Discussion:**

There was discussion about the strength of the paper's theoretical results, the performance with more Byzantine workers, etc. The authors have addressed these issues.

---

### Decision · Program_Chairs · 2025-01-22

Accept (Spotlight)